# Entropy Regularizing Activation: Boosting Continuous Control, Large Language Models, and Image Classification with Activation as Entropy Constraints

**Zilin Kang**[1,2*] **Chonghua Liao**[3*] **Tingqiang Xu**[3*] **Huazhe Xu**[1,3,4]
[1]Shanghai Qi Zhi Institute
[2]Department of Computer Science and Technology, Tsinghua University
[3]Institute for Interdisciplinary Information Sciences, Tsinghua University
[4]Shanghai Artificial Intelligence Laboratory
{kzl22,lch22,xtq23}@mails.tsinghua.edu.cn

## Abstract

We propose ERA, a new paradigm for entropy-constrained policy via output activation. It guarantees minimum sampling entropy by transforming the outputs of the last layer. Our approach demonstrates broad effectiveness across different domains: 1) for large language models (LLMs), boosting the average score across six benchmarks for Qwen2.5-Math-7B by 11.6%; 2) for continuous control reinforcement learning agents, improving performance by more than 30% over strong baselines such as SAC on the challenging HumanoidBench; 3) for image classification, enhancing ImageNet top-1 accuracy by 0.69% for ResNet-50. These gains are achieved with a computational overhead of less than 7%. Our work validates output activation as a powerful tool for entropy control, opening a new direction for designing simpler and more robust algorithms. Code available at: *https://nothingbutbut.github.io/era*

## 1 Introduction

Decision-making problems represent a broad class of challenges, from robotic control to Large Language Models alignment (Sutton et al., 1998; Ouyang et al., 2022; Kober et al., 2013). In these settings, encouraging exploration and maintaining policy stochasticity, often quantified by entropy, is critical (Ziebart et al., 2008; Schulman et al., 2017b). In reinforcement learning, the maximum entropy paradigm, exemplified by algorithms like Soft Actor-Critic (SAC) (Haarnoja et al., 2018), has become a prevailing approach in control tasks. However, these methods, which add an entropy bonus directly to the training objective, inevitably alter the optimization landscape and can interfere with the optimization of the primary objective.

The challenge becomes even more pronounced in LLM alignment. Policy gradient methods (Sutton et al., 1999) such as GRPO (Shao et al., 2024) frequently suffer from entropy collapse (Cui et al., 2025b), leading to reduced diversity and performance degradation. Directly incorporating entropy bonuses has been shown to be unstable or ineffective in this setting (Cui et al., 2025b). Moreover, prior works have explored methods that avoid direct modification of the loss function, including clip-higher (Yu et al., 2025) and training exclusively on the high-entropy tokens (Wang et al., 2025). While these methods provide useful insights, they remain ad hoc, lack a principled mechanism for entropy regulation, and are narrowly tailored to the LLM domain, limiting their applicability to broader settings such as continuous control and computer vision tasks.

These observations highlight a fundamental gap: existing approaches either distort the primary optimization objective, as in RL algorithms with entropy bonus terms, or provide heuristic, domain-specific fixes with no theoretical guarantees, as in LLM alignment. Therefore, there is a pressing need

---

*Equal Contribution

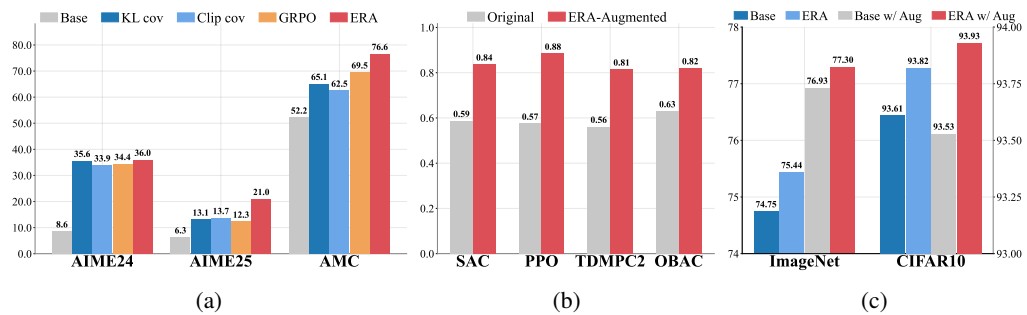

Figure 1: **ERA Boosts Large Language Models, Continuous Control and Image Classification.** (a) **Large Language Models:** ERA consistently enhances the performance of Qwen-2.5-Math-7B on AIME'24, AIME'25 and AMC datasets. (b) **Continuous Control:** ERA significantly improves multiple popular RL algorithms, including SAC, PPO, TD-MPC2 and OBAC. (c) **Image Classification:** ERA consistently boosts the performance of ResNet-50 on ImageNet and CIFAR-10 datasets.

for a new entropy-constraining paradigm that is universally applicable, non-invasive to the primary objective, and theoretically grounded.

In this work, we introduce **E**ntropy **R**egularizing **A**ctivation (ERA), a novel paradigm for entropy-constrained training. The key insight of ERA lies in realizing an entropy-constrained policy via output activation. Specifically, we impose the constraint through a class of well-designed activation functions applied to the model's final output. This approach completely decouples the optimization of the primary objective from the entropy constraint, allowing the loss function to focus solely on its original goal (e.g., maximizing rewards). We show that ERA not only provides provable entropy guarantees in theory, but in practice, it functions as a non-invasive module that can be seamlessly integrated with existing algorithms.

The generality and effectiveness of this paradigm are validated across diverse domains, including continuous control, image classification, and large language models. For example, on the DeepMind Control Suite (Tassa et al., 2018), ERA improves the performance of SAC on high-dimensional tasks like Humanoid and Dog by over 25%. Its versatility is also demonstrated in image classification, a domain where preventing model overconfidence via regularization is critical. Our approach complements established methods, boosting performance on top of strong data augmentation and label smoothing (Szegedy et al., 2016). In LLM RL, ERA enables a GRPO-trained Qwen-2.5-Math-7B (Yang et al., 2024b) to achieve a remarkable improvement of 4.7% and 70.7% on the AIME-24 and AIME-25 benchmarks, respectively.

Our main contributions are summarized as follows:

- We introduce **ERA, a novel entropy constraint paradigm** based on activation functions, and establish a theoretical framework with provable entropy guarantees.
- We design effective instantiations of ERA for both continuous (control) and discrete (image classification) domains. For large language models, we propose a **specialized, adaptive variant of ERA that addresses the unique challenges** within this domain.
- Our experiments of these instantiations **demonstrate significant performance improvements** over strong baselines across domains, and reveal their properties such as parameter sensitivity.

## 2 RELATED WORK

**Policy learning in control.** Entropy maximization is a crucial aspect of RL, significantly enhancing exploration and robustness (Ziebart, 2010; Haarnoja et al., 2017). Prior work has explored various methods to incorporate entropy maximization into RL algorithms (O'Donoghue et al., 2016; Nachum et al., 2017; Haarnoja et al., 2017). PPO (Schulman et al., 2017a) introduced an entropy bonus in its clipped surrogate objective. SAC (Haarnoja et al., 2018) later employed a maximum-entropy objective with a dynamically adjusted temperature parameter, but this can lead to suboptimal performance. More recent approaches have introduced alternative methodologies for implementing maximum entropy RL (Chao et al., 2024; Choe & Kim, 2024), while others have shifted the optimization focus directly to state entropy (Zhong et al., 2024). A different line of work avoids modifying the objective function. Akrour et al. (2019); Otto et al. (2021) pioneered this direction by projecting the policy

parameters to a constrained subspace. However, their instantiation relies on isotropic transformations (e.g., uniform mixing), which impose uniform regularization across all dimensions—a strategy that scales poorly to high-dimensional action spaces. In contrast, our work intervenes at the output layer with a non-linear activation that provides dimension-specific gradient guidance, enabling the network to learn structured exploration strategies rather than being forced into uniform stochasticity.

**RL for LLMs.** Recent breakthroughs in LLM reasoning, such as OpenAI-o1 (Jaech et al., 2024), DeepSeek-R1 (Guo et al., 2025), and Kimi-k1.5 (Team et al., 2025), have redirected attention from chain-of-thought prompting (Wei et al., 2022) and supervised fine-tuning (Li et al., 2024a; Yeo et al., 2025) toward RL. Within this paradigm, policy entropy collapse emerges as a fundamental obstacle: the decay of exploratory behavior often leads to performance plateaus. A prevalent approach is reward shaping (Cheng et al., 2025), which augments the reward or advantage with an entropy bonus to maintain a viable exploration–exploitation trade-off. Complementary strategies, including loss re-weighting (Wang et al., 2025; Cui et al., 2025b) and clip-higher regularization (Yu et al., 2025), mitigate the risk of entropy collapse. Unlike these approaches, our method is a general and concise paradigm, universally applicable across domains and endowed with rigorous theoretical guarantees.

## 3 PRELIMINARIES

**Markov Decision Process.** We consider a Markov Decision Process (MDP) (Bellman, 1957) defined by the tuple $(\mathcal{S}, \mathcal{A}, P, R)$, where $\mathcal{S}, \mathcal{A}$ are the state and action spaces, $P$ is the transition dynamics, and $R$ is the reward function. A policy $\pi_\theta(a_t|s_t)$ parameterized by $\theta$ aims to maximize the expected discounted return:

$$J(\pi_\theta) = \mathbb{E}_{\tau \sim \pi_\theta} \left[ \sum_{t=0}^{T} \gamma^t R(s_t, a_t) \right],$$ (1)

where $\gamma \in [0, 1)$ is the discount factor, $t$ is the timestep, $s_t \in \mathcal{S}$ and $a_t \in \mathcal{A}$ are the state and action at timestep $t$, and $\tau = (s_0, a_0, s_1, a_1, \dots)$ represents a full trajectory sampled by following the policy $\pi_\theta$.

**Policy optimization.** Policy gradient (PG) methods optimize $J(\pi_\theta)$ via gradient ascent. In the context of large language model (LLM) alignment, this MDP formalism is adapted: the **state** $s_t$ represents the initial prompt $x$ combined with the sequence of tokens generated so far ($y_{<t}$), and the **action** $a_t$ is the next token $y_t$ sampled from the policy $\pi_\theta(y_t|s_t)$. The full trajectory $\tau$ thus corresponds to the complete generated response, denoted as $y = (y_1, \dots, y_T)$. The reward is typically sparse, with a single score $r(y)$ (from a reward model) assigned to the entire sequence $y$ at the final timestep.

Proximal Policy Optimization (PPO) (Schulman et al., 2017b) is commonly used for this optimization. The GRPO variant estimates the advantage $A(y)$ for a single, complete response $y$. This advantage is normalized using a set of $K$ responses, $y^{1:K} = \{y^1, \dots, y^K\}$, sampled from the policy for the *same initial prompt*:

$$A(y) = \frac{r(y) - \text{mean}(r(y^{1:K}))}{\text{std}(r(y^{1:K}))}.$$ (2)

The policy is then updated using the clipped surrogate objective, which operates at the token level:

$$\mathcal{L}^{\text{CLIP}}(\theta) = \mathbb{E}_t \left[ \min \left( r_t(\theta) A_t, \text{clip}(r_t(\theta), 1 - \epsilon, 1 + \epsilon) A_t \right) \right],$$ (3)

where $r_t(\theta) = \frac{\pi_\theta(a_t|s_t)}{\pi_{\theta_{\text{old}}}(a_t|s_t)}$ is the probability ratio, and the per-timestep advantage $A_t$ is the trajectory-level advantage $A(y)$ from Eq. 2 propagated back to timestep $t$.

**Policy entropy.** Policy entropy, $\mathcal{H}(\pi(\cdot|s))$, measures the policy's stochasticity. For discrete action spaces, the token-level entropy is given by Eq. 4. For continuous policies, there are several common ways to ensure actions remain within a bounded space. A popular method is to use a squashed Gaussian policy, which outputs a bounded action $a = \tanh(u)$ by sampling $u$ from a Gaussian distribution $\pi_\theta(\cdot|s) = \mathcal{N}(\mu_\theta(s), \Sigma_\theta(s))$ parameterized by the policy network. The entropy of this policy is given by Eq. 5. Alternatively, another common approach is to directly sample actions from a Truncated Gaussian distribution $\pi_\theta(\cdot|s) = \text{TN}(\mu_\theta(s), \Sigma_\theta(s), -1, 1)$ over the bounded hypercube

$[-1,1]^D$. Assuming the dimensions are independent, its entropy is given by Eq. 6.

$$\mathcal{H}(\pi_\theta) = -\mathbb{E}_{x \sim \rho_\pi, y \sim \pi_\theta(x)} \left[ \frac{1}{|y|} \sum_{t=1}^{|y|} \log \pi_\theta(y_t|y_{<t}, x) \right], \tag{4}$$

$$\mathcal{H}(\pi_\theta) = \mathbb{E}_{s \sim \rho_\pi, u \sim \mathcal{N}(\mu_\theta(s), \Sigma_\theta(s))} \left[ -\log \mathcal{N}(u|\mu_\theta(s), \Sigma_\theta(s)) + \sum_{i=1}^{D} \log(1 - \tanh(u_i)^2) \right], \tag{5}$$

$$\mathcal{H}(\pi_\theta) = \mathbb{E}_{s \sim \rho_\pi} \left[ \sum_{i=1}^{D} \left( \log(\sigma_{\theta,i}(s) Z_i(s) \sqrt{2\pi e}) - \frac{\beta_i(s)\phi(\beta_i(s)) - \alpha_i(s)\phi(\alpha_i(s))}{2 Z_i(s)} \right) \right] \tag{6}$$

where for the truncated Gaussian entropy in Eq. 6, $\phi$ and $\Phi$ are the PDF and CDF of the standard normal distribution, respectively. We define the standardized bounds $\alpha_i(s) = (-1 - \mu_{\theta,i}(s))/\sigma_{\theta,i}(s)$, $\beta_i(s) = (1 - \mu_{\theta,i}(s))/\sigma_{\theta,i}(s)$, and the normalization constant $Z_i(s) = \Phi(\beta_i(s)) - \Phi(\alpha_i(s))$.

**Maximum entropy reinforcement learning.** Building upon policy entropy, the maximum entropy RL framework aims to maximize the standard reward objective subject to a minimum entropy constraint $\mathcal{H}_0$:

$$\max_\theta J(\pi_\theta) \quad \text{s.t.} \quad \mathbb{E}_{s \sim \rho_\pi}[\mathcal{H}(\pi_\theta(\cdot|s))] \geq \mathcal{H}_0. \tag{7}$$

Practical algorithms like Soft Actor-Critic (SAC) (Haarnoja et al., 2018) solve the Lagrangian dual of this problem. SAC is an off-policy actor-critic algorithm that updates a soft Q-function $Q_\phi$ and a policy $\pi_\theta$. The Q-function is updated by minimizing the soft Bellman residual $J_Q(\phi)$:

$$J_Q(\phi) = \mathbb{E}_{(s_t, a_t, s_{t+1}) \sim \mathcal{D}} \left[ \frac{1}{2} \left( Q_\phi(s_t, a_t) - y \right)^2 \right] \tag{8}$$

$$y = R(s_t, a_t) + \gamma \mathbb{E}_{a_{t+1} \sim \pi_\theta(\cdot|s_{t+1})} \left[ Q_{\phi'}(s_{t+1}, a_{t+1}) - \alpha \log \pi_\theta(a_{t+1}|s_{t+1}) \right] \tag{9}$$

with the target $y$ computed using a target Q-network $Q_{\phi'}$. The target network parameters $\phi'$ are updated via an exponential moving average (EMA): $\phi' \leftarrow \tau\phi + (1 - \tau)\phi'$.

$$J_\pi(\theta) = \mathbb{E}_{s_t \sim \mathcal{D}, a_t \sim \pi_\theta} \left[ Q_\phi(s_t, a_t) - \alpha \log \pi_\theta(a_t|s_t) \right]. \tag{10}$$

The policy is then updated by maximizing the objective in Eq. 10.

## 4 THE ENTROPY REGULARIZING ACTIVATION

### 4.1 THE CORE IDEA: ENTROPY-CONSTRAINED POLICY VIA OUTPUT ACTIVATION

The core of Entropy Regularizing Activation is to enforce maximum entropy reinforcement learning on the policy, not through a loss penalty, but via integrating the constraint into the network's architecture via a special activation function at the output layer.

Let a parameterized policy $f_\theta(s)$ produce distribution parameters $z = f_\theta(s)$, where $z$ belongs to a parameter space $\mathcal{Z}$. The policy corresponding to these parameters is $\pi_z(\cdot|s)$. We introduce an activation function $g : \mathcal{Z} \to \mathcal{Z}$, which transforms the initial parameters $z$ to a new set $z' = g(z)$. The final policy, which we denote as $\pi_\theta$, is thus given by $\pi_\theta(\cdot|s) = \pi_{g(f_\theta(s))}(\cdot|s)$. The function $g(.)$ is designed to ensure that the policy $\pi_\theta$ satisfies a constraint on its expected entropy, for a given target entropy $\mathcal{H}_0$:

$$\mathbb{E}_{s \sim \rho_\pi}[\mathcal{H}_{\pi_\theta(\cdot|s)}] \geq \mathcal{H}_0$$

This formulation enables the policy to satisfy the expected entropy condition while leaving the training objective for $\theta$ free of an explicit entropy term, as shown in Eq. 7. This architectural perspective unifies prior projection methods: for instance, the method in Akrour et al. (2019) can be viewed as a specific, linear instantiation of $g(\cdot)$. ERA generalizes this to a class of non-linear activations that strictly satisfy the bound while **modulating gradients in a dimension-aware manner**, avoiding the suboptimal uniform regularization of prior linear methods. For the facts of this argument, we refer the reader to the derivations in Appendix B.1 and the experimental results in C.1, where detailed theoretical analysis and experimental demonstrations are provided.

## 4.2 Instantiations for Continuous and Discrete Spaces

To ground the general framework presented in section 4.1, we now instantiate the entropy regularizing activation $g(.)$ for two canonical policy classes: policies based on a bounded Gaussian distribution, such as the Tanh-squashed Gaussian (Haarnoja et al., 2018) or the clipped Gaussian (Fujimoto et al., 2018), commonly used in continuous control; and the softmax policy prevalent in discrete spaces.

### 4.2.1 Continuous Control with Bounded Gaussian Policies

In continuous control, policies often sample actions from a Gaussian distribution and then apply a bounding function (e.g., a $\texttt{tanh}$ squash or clipping) to ensure outputs lie within a valid range. This bounding operation complicates direct entropy maximization, as it introduces a state-dependent bias term. Prior methods typically address this by adding an entropy bonus to the learning objective. Our insight is that the entropy of the final bounded policy, $\mathcal{H}_\pi$, can be seen as the entropy of the original unbounded Gaussian, $\mathcal{H}_{\text{Gaussian}}$, minus a non-negative bias term introduced by the bounding operation, i.e., $\mathcal{H}_\pi = \mathcal{H}_{\text{Gaussian}} - \mathbb{E}[\text{bias}]$. Consequently, a minimum entropy constraint on the final policy can be satisfied by constraining the underlying Gaussian's entropy to a corresponding, higher value. This is achieved by adjusting the Gaussian's standard deviation, $\sigma$. The entropy of a $D$-dimensional Gaussian with a diagonal covariance matrix is:

$$\mathcal{H}_{\text{Gaussian}}(s) = \frac{1}{2}\sum_{i=1}^{D}\log(2\pi e\sigma_i(s)^2) \tag{11}$$

To maintain training stability, the standard deviation must also be kept within a predefined range $[\sigma_{\min}, \sigma_{\max}]$. Our activation function $g(.)$ simultaneously satisfies both constraints. Given network outputs (before tanh squash or truncation) for the mean $\mu$ and a pre-activation standard deviation $\hat{\sigma}$, the function $g(\mu, \hat{\sigma})$ produces the final parameters $(\mu', \sigma')$ where:

$$\mu' = \mu, \quad \sigma' = \exp\left[\max\left(\log\sigma_{\max} + (\mathcal{H}_0' - D\log\sqrt{2\pi e} - D\log\sigma_{\max})\frac{e^{\hat{\sigma}_i}}{\sum_{j=1}^{D}e^{\hat{\sigma}_j}}, \log\sigma_{\min}\right)\right] \tag{12}$$

Here, $\mathcal{H}_0'$ is the target entropy for the final policy $\mathcal{H}_0$ plus a compensation term $\delta = -\mathbb{E}[\text{bias}]$ to account for the bounding bias. We use a parameter $\hat{\delta}$ to estimate $\delta$. In practice, $\hat{\delta}$ can either be set a constant or automatically tuned by learning with the loss in Eq. 13.

$$\mathcal{L}(\hat{\delta}) = \mathbb{E}_{s\sim\mathcal{D}}\left[\hat{\delta}(\mathcal{H}[\pi(\cdot|s)] - \mathcal{H}_0)\right] \tag{13}$$

We refer the reader to Appendix A.1 for implementation details and Appendix B.1 for a proof of the entropy bound.

By satisfying the entropy constraint architecturally, our method obviates the need for an explicit entropy term in the objective function. Hence, target of the critic and the actor loss of SAC in Eq. 9 and Eq. 10 can be simplified to the form in Eq. 14 and Eq 15

$$y = R(s_t, a_t) + \gamma\mathbb{E}_{a_{t+1}\sim\pi_\theta(\cdot|s_{t+1})}\left[Q_{\phi'}(s_{t+1}, a_{t+1})\cancel{-\alpha\log\pi_\theta(a_{t+1}|s_{t+1})}\right] \tag{14}$$

$$J_\pi(\theta) = \mathbb{E}_{s_t\sim\mathcal{D}, a_t\sim\pi_\theta}\left[Q_\phi(s_t, a_t)\cancel{-\alpha\log\pi_\theta(a_t|s_t)}\right] \tag{15}$$

### 4.2.2 Discrete Classification with Softmax Policies

In discrete classification, regularizing the predictive entropy is crucial for preventing the overconfidence that leads to overfitting. ERA provides architectural regularization by enforcing a minimum entropy level, analogous to how techniques like label smoothing improve generalization by smoothing the output distribution. For a softmax policy, we enforce this constraint by transforming the pre-activation logits $z$ into $z'$ such that the resulting policy's entropy is at least $\mathcal{H}_0$:

$$z' = h^{-1}\left[\max\left(\frac{\log\tau}{\tau} + \left(C_{\mathcal{H}_0} - n\frac{\log\tau}{\tau}\right)\frac{1}{D-1}\left(1 - \frac{e^{z_i}}{\sum_{j=1}^{D}e^{z_j}}\right), 0\right)\right] \tag{16}$$

Here, $h^{-1}$ denotes the inverse of $-xe^x$ on $(-\infty, -1]$, approximated by $\hat{h}^{-1}(x) = -\frac{1}{4} - \sqrt{2(-1 - \ln(x))} + \frac{3}{4}\ln x$. We also define $C_{\mathcal{H}_0} = \exp(\mathcal{H}_0 - 1)$, where $\tau \geq e$ is a fixed hyperparameter (e.g., $\tau = 4$). A formal proof is provided in Appendix B.2.

Figure 2: **Main Results of ERA in Continuous Control.** Aggregate normalized performance on HumanoidBench (6 tasks, with SAC), DMC (Humanoid & Dog) (6 tasks, with TD-MPC2), HumanoidBench (8 tasks, with FastSAC) and Mujoco Gym (4 tasks, with PPO). ERA consistently accelerates learning and achieves superior asymptotic performance.

In contrast to label smoothing, which applies a fixed and uniform regularization, ERA offers greater flexibility. It allows the model to learn a structured, input-dependent uncertainty distribution, tailoring the regularization to each sample and thus offering greater expressive capacity and potential for improved performance.

### 4.3 INSTANTIATIONS FOR RL IN LARGE LANGUAGE MODELS

In reinforcement learning for LLMs, each token is treated as a discrete action, with the policy defined by a canonical softmax distribution. Prior approaches to addressing entropy collapse in LLMs—such as the traditional entropy bonus, clip-higher , KL-Cov, and Clip-Cov —do not provide a provable entropy lower bound, and are incompatible with the on-policy setting, as they often need the importance sampling ratio or the KL loss term that arises only in off-policy training. In contrast, our method introduces ERA, a simple and non-invasive activation function that offers a theoretical guarantee of a minimum entropy level, effectively resolving entropy collapse in on-policy reinforcement learning.

In contrast to standard RL settings, the action space is extremely large. In the previous ERA instantiation, each token has a lower entropy bound. However, due to the intrinsic structure of natural language, most tokens are nearly deterministic; therefore, directly enforcing high entropy across all tokens is impractical: it will lead to unintended tokens and can corrupt the entire response. Furthermore, modifying the internal structure of the model also introduces instability in different training environments, leading to unpredictable behavior.

To address these challenges, we propose a new instantiation of ERA that is applied *after* the sampling process. Specifically, responses are first generated using the original model output $z$, and the advantages are computed following the GRPO rule. Then, during model updates, the probabilities of the sampled tokens are reinterpreted as $z'$, obtained by applying our entropy-regularized activation. This design leaves the sampling policy unchanged while still ensuring effective entropy regularization.

Formally, when updating model parameters, we apply an activation layer to the logits $z$ to obtain a transformed set $z'$, defined as:

$$z' = \begin{cases} kz & H_{\text{resp}} < \omega_{\text{low}},\ A_t > 0, \\ z & (\omega_{\text{low}} \le H_{\text{resp}} \le \omega_{\text{high}},\ A_t < 0)\ \text{or}\ A_t > 0, \\ \frac{1}{k}z & H_{\text{resp}} > \omega_{\text{high}},\ A_t > 0, \end{cases} \tag{17}$$

where $k > 1$, and $\omega_{\text{low}}, \omega_{\text{high}}$ are algorithm-specific constants. Here, $A_t$ denotes the advantage of the token, and $H_{\text{resp}}$ is the average entropy of the top $20\%$ of tokens with the highest entropy in the response. To balance the gradient between modified tokens and unmodified tokens (details are shown in Appendix B.3), we add another scaling factor on the advantages of modified tokens:

$$A'_t = \begin{cases} \frac{1}{k}A_t & H_{\text{resp}} < \omega_{\text{low}},\ A_t > 0, \\ A_t & (\omega_{\text{low}} \le H_{\text{resp}} \le \omega_{\text{high}},\ A_t < 0)\ \text{or}\ A_t > 0, \\ kA_t & H_{\text{resp}} > \omega_{\text{high}},\ A_t > 0, \end{cases} \tag{18}$$

The on-policy GRPO objective becomes:

$$J(\theta) = \mathbb{E}_t[\mathbb{E}_{a_t \sim \pi_\theta(\cdot|s_t)} \log \pi'_\theta(a_t|s_t) A'_t] \tag{19}$$

where $\pi_\theta$ is the original policy from $z$ (representing that the inference still follows the original policy), and $\pi'_\theta$ is the ERA-adjusted policy from $z'$ (representing that the model update relies on the new policy). Intuitively, this activation layer adjusts all positively advantaged responses: when entropy is too low, it sharpens the probability distribution; when entropy is too high, it flattens it. Unlike our instantiation for control tasks, increasing policy entropy here requires *sharpening* the distribution. The rationale is that sampling has already occurred, and by treating the samples as if they were drawn from a sharpened policy, the model perceives itself as overexploiting, thus encouraging additional exploration. The choice of the top $20\%$ tokens is based on the fact that, in natural language, these tokens are considered forking tokens, whose entropy is the target of regularization, and the remaining tokens are allowed to have almost zero entropy (Wang et al., 2025).

We show that, under reasonable assumptions, this ERA instantiation ensures that the policy entropy remains above a fixed constant $\mathcal{H}_0$. We refer the reader to Appendix B.3 for a formal proof.

## 5 RESULTS AND ANALYSIS

### 5.1 EXPERIMENTS ON CONTINUOUS CONTROL

We conduct extensive experiments to validate the effectiveness of ERA in continuous control tasks. We demonstrate the broad applicability and performance gains by integrating ERA into five distinct algorithms—SAC, OBAC (Luo et al., 2024), TD-MPC2, PPO, and FastSAC (Seo et al., 2025). The evaluation is performed on a wide range of challenging benchmarks, including the DeepMind Control Suite (Humanoid & Dog), HumanoidBench (Sferrazza et al., 2024), and MuJoCo Gym (Todorov et al., 2012). Implementation details, environment specifics, and hyperparameter settings are available in Appendix A.1. Comprehensive results for all tasks can be found in the Appendix C.

**Main results.** We present our main results in continuous control in Figure 2. Integrating ERA consistently yields significant improvements in both sample efficiency and final performance across diverse algorithms and benchmarks.

**ERA consistently improves performance across various entropy targets.** We evaluate the performance of SAC and SAC-ERA under varying entropy targets. The results in Figure 3a, tested on four DMC tasks (*dog-run, dog-trot, humanoid-run, humanoid-walk*) with 5 seeds on each environment, show that SAC-ERA consistently outperforms original SAC across the entire tested spectrum of entropy values. By bypassing the entropy constraint within the learning objective, ERA allows the policy to focus more directly on reward maximization. While simply removing the entropy term from SAC can also avoid this constraint, its performance is inferior to the ERA-enhanced version due to insufficient exploration. This consistent outperformance suggests that ERA can achieve strong results without precise tuning of the entropy hyperparameter, offering a significant practical advantage.

### 5.2 EXPERIMENTS ON IMAGE CLASSIFICATION

Table 1: Top-1 and Top-5 accuracy (%) on ImageNet and CIFAR-10. We compare ERA against the original ResNet-50 baseline. $\Delta$ denotes the absolute improvement of ERA. All models are trained for 200 epochs.

| Dataset | Method | Without Data Augmentation | | | | With Data Augmentation | | | |
|---|---|---|---|---|---|---|---|---|---|
| | | Top-1 Acc. | $\Delta$ | Top-5 Acc. | $\Delta$ | Top-1 Acc. | $\Delta$ | Top-5 Acc. | $\Delta$ |
| ImageNet | Original | $74.75 \pm 0.38$ | - | $92.04 \pm 0.23$ | - | $76.93 \pm 0.36$ | - | $93.37 \pm 0.21$ | - |
| | ERA | $\mathbf{75.44 \pm 0.37}$ | +0.69 | $\mathbf{92.15 \pm 0.23}$ | +0.11 | $\mathbf{77.30 \pm 0.36}$ | +0.37 | $\mathbf{93.39 \pm 0.21}$ | +0.02 |
| CIFAR-10 | Original | $93.61 \pm 0.14$ | - | $99.69 \pm 0.08$ | - | $93.53 \pm 0.03$ | - | $99.84 \pm 0.02$ | - |
| | ERA | $\mathbf{93.82 \pm 0.08}$ | +0.21 | $\mathbf{99.82 \pm 0.03}$ | +0.13 | $\mathbf{93.93 \pm 0.12}$ | +0.4 | $\mathbf{99.86 \pm 0.01}$ | +0.02 |

We evaluate our method on the ImageNet (Russakovsky et al., 2015) and CIFAR-10 datasets (Krizhevsky et al., 2009). Our implementation utilizes the ResNet-50 architecture from the PyTorch Image Models (`timm`) library (Wightman, 2019). To ensure a fair comparison, both our method and the baseline were trained for 200 epochs, with all other hyperparameters held constant. Notably, we retain key default settings from `timm` for all experiments, including a label smoothing factor of 0.1. This demonstrate ERA's complementarity with existing regularizations.

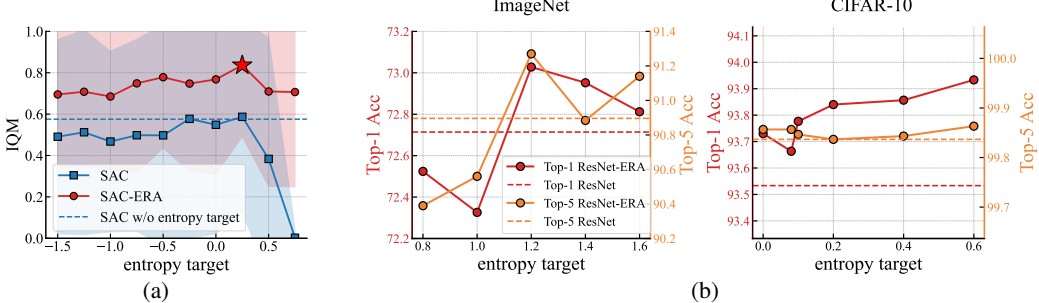

Figure 3: **Sensitivity of ERA to the Minimum Entropy.** (a) **1M Steps Performance on DMC Tasks.** Comparison between SAC-ERA and the baseline SAC on Humanoid and Dogs environments under various minimum entropy constraints. Our method achieves superior performance across all settings. (b) **Accuracy on ImageNet and CIFAR-10.** ResNet-ERA maintains stable Top-1 and Top-5 accuracy across a range of minimum entropy values, indicating its robustness to the choice of this hyperparameter.

**Main results.** Table 1 summarizes the primary classification results, comparing ERA against the standard ResNet-50 baseline. For these results, we use a minimal entropy of 1.2 for ImageNet and 0.6 for CIFAR-10. The comparison is conducted under two settings: with and without the standard data augmentation provided by the timm library. The results show that ERA consistently outperforms the baseline across both datasets and settings.

**Ablation study on minimal entropy.** We study our method's robustness to the minimal entropy hyperparameter on ImageNet and CIFAR-10, using checkpoints from the 100th and 200th epochs, respectively, for efficiency. As shown in Figure 3b, our method exhibits low sensitivity to this parameter. Rather than fine-tuning for peak performance, our intent is to show that competitive accuracy is maintained across a reasonable range of values. This demonstrates strong performance is achievable without extensive tuning.

## 5.3 RESULTS AND ANALYSIS ON LARGE LANGUAGE MODELS

We first present the results of ERA in §5.3.1 Main Results and §5.3.2 Extension to More Models and Algorithms. We then use §5.3.3 Analysis on Entropy and Reasoning Capacity Boundary and §5.3.4 Out-of-Distribution Generalization to illustrate the role of encouraging exploration. Additional ablation studies on method design are provided in the Appendix C.3.

### 5.3.1 MAIN RESULTS

We evaluate ERA on Qwen2.5-Math-7B, trained with the DAPO-Math-17K (Yu et al., 2025) dataset using codebase adopted from verl (Sheng et al., 2025). We set $\omega_{\text{low}} = 0.45$, $\omega_{\text{high}} = 3.0$, and $k = 2$.

We then evaluate the final resulting model on six standard mathematical reasoning tasks: AIME'24, AIME'25, AMC'23 (Li et al., 2024b), MATH500 (Hendrycks et al., 2021), Minerva (Lewkowycz et al., 2022), and OlympiadBench (He et al., 2024). Table 2 presents comparisons against base models, classical RL methods, and recent entropy-control approaches. AIME'24, AIME'25, and AMC'23 are conducted with a decoding temperature of $0.7$, and reported as the average accuracy over 16 sampled responses. MATH500, Minerva, and OlympiadBench are conducted with greedy sampling. The evaluation process is sampled on the original policy $z$ (before ERA). Full implementation details and hyperparameter settings are provided in Appendix A.3. The results show that ERA consistently achieves the best results on most of the benchmarks. Notably, it outperforms strong entropy-based baselines such as KL-Cov and Clip-Cov by significant margins.

### 5.3.2 EXTENSION TO MORE MODELS AND ALGORITHMS

To demonstrate ERA's effectiveness across different model sizes and algorithms, we extend it to the weaker Qwen2.5-Math-1.5B model and also apply ERA to other algorithms such as GSPO (Zheng et al., 2025) on Qwen2.5-Math-7B, showing that ERA is a generic approach not tied to any specific model or algorithm. As reported in Table 3, ERA yields significant gains on both the smaller model and GSPO. For instance, on Qwen2.5-Math-1.5B it achieves an average improvement of 14.1%.

Table 2: Main results (%) on five competition-level reasoning benchmarks based on Qwen2.5-Math-7B. For AIME and AMC, the results are avg.@16. The best results on each benchmark are highlighted in **bold**.

| Model | AIME24 ↑ | AIME25 ↑ | AMC ↑ | MATH500 ↑ | Minerva ↑ | Olympiad ↑ | Avg. ↑ |
|---|---|---|---|---|---|---|---|
| *Base Models* | | | | | | | |
| Qwen2.5-Math Yang et al. (2024a) | 8.6 | 6.3 | 52.2 | 50.8 | 12.1 | 17.2 | 24.5 |
| Qwen2.5-Math-Instruct Yang et al. (2024a) | 13.3 | 10.0 | 57.1 | 81.0 | 32.7 | 38.8 | 38.8 |
| *Classical Methods* | | | | | | | |
| SimpleRL-Zero Zeng et al. (2025) | 26.7 | 9.3 | 60.0 | 74.6 | 27.6 | 35.8 | 39.0 |
| OpenReasoner-Zero Hu et al. (2025) | 15.4 | 13.4 | 56.5 | 81.0 | 32.7 | 43.2 | 40.4 |
| PRIME-Zero Cui et al. (2025a) | 18.9 | 11.7 | 57.7 | 79.0 | 36.4 | 40.6 | 40.7 |
| Oat-Zero Liu et al. (2025) | 28.8 | 10.8 | 65.2 | 79.6 | 34.2 | 39.9 | 43.1 |
| *Entropy Control Methods* | | | | | | | |
| GRPO + Entropy Loss | 32.5 | 14.0 | 66.9 | 80.8 | 36.0 | 42.5 | 45.5 |
| GRPO w/ 20% Forking Tokens (Wang et al., 2025) | 29.0 | 17.7 | 63.6 | 81.8 | 39.7 | 44.6 | 46.1 |
| KL-Cov (Cui et al., 2025b) | 35.6 | 13.1 | 65.1 | 81.0 | **40.4** | 44.1 | 46.6 |
| Clip-Cov (Cui et al., 2025b) | 33.9 | 13.7 | 62.5 | 78.4 | 35.6 | 40.3 | 44.1 |
| GRPO (Shao et al., 2024) | 34.4 | 12.3 | 69.5 | 80.6 | 36.8 | 40.6 | 45.7 |
| ERA | **36.0** | **21.0** | **76.6** | **85.4** | 40.1 | **46.8** | **51.0** |
| △ (↑) | +4.7% | +70.7% | +10.4% | +6.0% | +9.0% | +15.3% | +11.6% |

Table 3: Accuracy (%) results of different LLMs and different algorithms across six benchmarks. The best results in each box are highlighted in **bold**.

| Method | AIME24 ↑ | AIME25 ↑ | AMC ↑ | MATH500 ↑ | Minerva ↑ | Olympiad ↑ | Avg. ↑ |
|---|---|---|---|---|---|---|---|
| Qwen2.5-Math-1.5B Yang et al. (2024a) | | | | | | | |
| CoT | 4.3 | 2.3 | 26.4 | 59.0 | 24.3 | 27.6 | 24.0 |
| GRPO | 11.1 | 6.0 | 40.2 | 66.4 | 25.0 | 30.1 | 29.8 |
| ERA | **12.1** | **6.8** | **49.5** | **70.6** | **30.5** | **34.7** | **34.0** |
| △ (↑) | +9.0% | +13.3% | +23.1% | +6.3% | +22.0% | +15.3% | +14.1% |
| Qwen2.5-Math-7B Yang et al. (2024a) | | | | | | | |
| CoT | 8.6 | 6.3 | 52.2 | 50.8 | 12.1 | 17.2 | 24.5 |
| GSPO | 29.8 | 13.7 | 61.2 | **85.1** | 37.1 | 35.1 | 43.7 |
| GSPO + ERA | **33.3** | **15.2** | **63.8** | 84.3 | **40.8** | **42.7** | **46.7** |
| △ (↑) | +11.7% | +10.9% | +4.2% | -0.9% | +10.0% | +21.7% | +6.9% |

### 5.3.3 ANALYSIS ON ENTROPY AND REASONING CAPACITY BOUNDARY

To better understand the effect of our approach on exploration and reasoning, we examine both the entropy dynamics of the learned policies and their downstream reasoning performance. Figure 4 compares the entropy trajectories of our method (first stage) with the GRPO baseline. While GRPO suffers from entropy collapse, our method maintains a stable entropy level throughout training. This stability indicates the existence of a *non-trivial entropy lower bound*, as we desired by the definition of ERA, which prevents premature policy concentration and preserves the model's ability to explore diverse reasoning paths.

The presence of this entropy floor correlates with improved reasoning performance. To reduce the effect of stochastic fluctuations, we report the pass@$k$ score as the highest value achieved across all evaluation checkpoints (evaluated every 10 steps). As shown in Figure 4, ERA consistently attains higher pass@$k$ scores than GRPO on AIME'24 and AIME'25. These results indicate that preventing entropy collapse is not merely a statistical artifact, but directly translates into enhanced reasoning capability. In particular, maintaining sufficient entropy enables the model to preserve diverse candidate reasoning trajectories, thereby increasing the probability of producing correct solutions under pass@$k$ evaluation.

### 5.3.4 OUT-OF-DISTRIBUTION GENERALIZATION

Models trained in a specific domain often struggle when applied to other domains (Yuan et al., 2023; Wang et al., 2024a). Since ERA uses entropy constraints to encourage exploration, we hope it can learn *more general skills*. Therefore we want to see if ERA will also do better on out-of-distribution (OOD) data than standard GRPO. To test this, we evaluate ERA on three hard OOD benchmarks: ARC-C (Clark et al., 2018), GPQA-Diamond (Rein et al., 2024), and MMLU-Pro (Wang et al.,

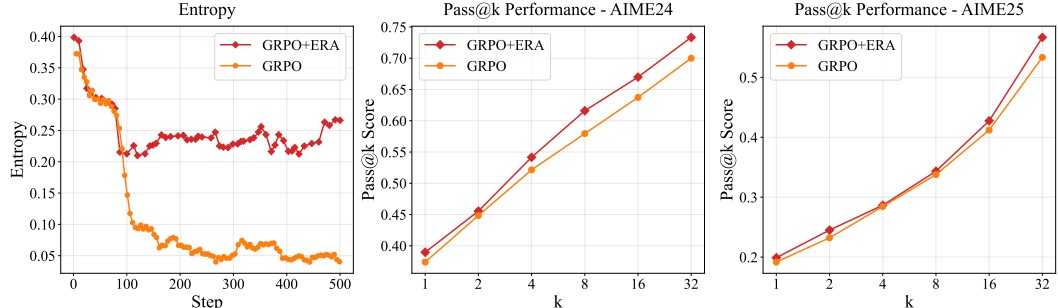

Figure 4: **Entropy comparison and pass@$k$ results for GRPO with ERA (ours) versus GRPO alone.** The entropy curves demonstrate that ERA mitigates entropy collapse and establishes a clear lower bound. The pass@$k$ results further indicate that ERA enhances exploration and strengthens the model's reasoning ability.

2024b). As shown in Figure 5, ERA outperforms GRPO by 16.9% on average. This confirms our hypothesis that ERA can also enable models to learn more generalizable abilities.

## 6 LIMITATIONS AND FUTURE WORK

Our work is centered within the maximum entropy reinforcement learning (MaxEnt RL) framework, with the primary objective of imposing effective entropy constraints to enhance exploration. We have demonstrated its effectiveness across diverse tasks, including continuous locomotion, discrete-space image classification, and the reinforcement learning post-training of large language models.

However, this reliance on the MaxEnt objective constitutes a potential limitation. The goal of maximizing entropy is not universally beneficial and can lead to suboptimal policies in certain task scenarios, as highlighted by Zhang et al. (2025). Therefore, the broader applicability of our method in domains where maximum entropy may not be the desired objective requires further investigation.

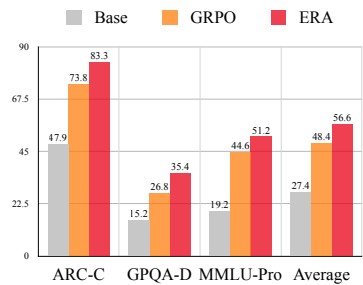

Figure 5: Results on three OOD benchmarks (Qwen2.5-Math-7B).

A promising direction for future research is to adapt and apply our method to a wider range of domains. This includes areas such as diffusion and flow-based generative models, or other tasks that could benefit from structured policy diversity and efficient exploration, even outside the strict MaxEnt RL paradigm.

## 7 CONCLUSIONS

In this work, we introduced ERA, a novel entropy-constrained paradigm built upon the unique principle of treating output activations as a direct medium for entropy regularization. Our theoretical analysis is substantiated by strong empirical results across diverse and challenging domains. In these settings, ERA consistently surpasses prominent baselines without incurring significant computational overhead. Ultimately, this work offers a new perspective on entropy regularization for both supervised and unsupervised decision-making, opening a promising research avenue for developing more robust and efficient learning agents.

## REPRODUCIBILITY STATEMENT

We are strongly committed to the reproducibility of our work. To this end, we provide detailed derivations and proofs for all theoretical claims in the appendix. The appendix also contains comprehensive experimental details, including hyperparameters, environment setups, and additional results, which are crucial for replicating our findings. Furthermore, the core source code for our proposed method, ERA, instantiated across all domains, is included in the appendix. As our implementations are built upon publicly available codebases and frameworks, we believe the provided key source code is sufficient for a straightforward reproduction of our results. Our official implementations can be found at *https://nothingbutbut.github.io/era*

## ACKNOWLEDGEMENTS

This work was supported by the Tsinghua University Initiative Scientific Research Program No. 20257020004. We would also like to express our gratitude to Kaizhe Hu, Ruizhe Shi, and Huanyu Li from the Tsinghua Embodied AI Lab for their invaluable discussions and insightful feedback, which have significantly contributed to this work.

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

# A IMPLEMENTATION DETAILS

## A.1 IMPLEMENTATION DETAILS OF CONTINUOUS CONTROL TASKS

### A.1.1 CODE IMPLEMENTATION OF ERA IN CONTINUOUS CONTROL

Listing 1: Original Implementation

```
# Original implementation from the
    jaxrl codebase, suggested by
    Ilya
# log_std_min, log_std_max: bounds
    for log standard deviation
# action_dim: dimension of the
    action space
# pre_stds: direct output from the
    actor network
log_stds = log_std_min + (
    log_std_max - log_std_min) *
    0.5 * (1 + nn.tanh(pre_stds))
```

Listing 2: ERA Implementation

```
# h_0: target entropy, can be a
    fixed value or a learnable
    parameter
# action_dim: dimension of the
    action space
k = - self.action_dim * (
    log_std_max + h_0 + jnp.log(jnp
    .sqrt(2 * jnp.pi * jnp.e)))
log_stds = k * nn.softmax(pre_stds,
    axis = -1) + log_std_max
log_stds = jax.clip(log_stds, self.
    log_std_min, self.log_std_max)
```

Figure 6: Comparison of the activation function at the actor's output.

We provide the following JAX implementation snippet of ERA for the reader's reference, where h_0 is the target entropy ($\mathcal{H}_0'$ in Eq. 12), which can be a constant (e.g., -action_dim/2) or a learnable parameter. The terms log_std_min and log_std_max represent the lower and upper bounds of the log standard deviation, respectively; action_dim is the dimension of the action space; and pre_stds refers to the raw output of the actor network.

### A.1.2 ENVIRONMENTS

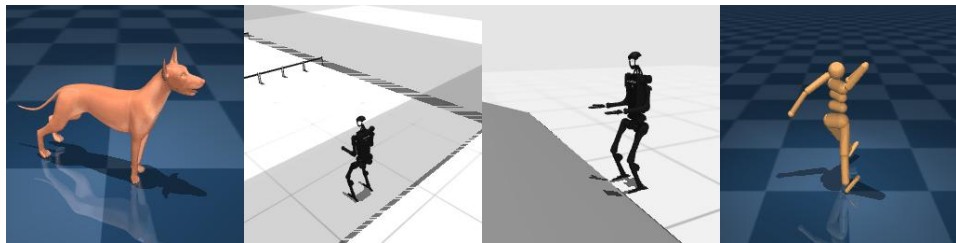

Figure 7: **Visualization of some continuous control environments used in our experiments.** From left to right: dog-run (DMC), h1-hurdle-v0 (HumanoidBench), h1hand-slide-v0 (HumanoidBench), humanoid-walk (DMC)

Our evaluation of ERA spans a diverse set of continuous control tasks from three established benchmarks: Mujoco Gym (Todorov et al., 2012), DeepMind Control Suite (DMC) (Tassa et al., 2018), and HumanoidBench (Sferrazza et al., 2024). For the Mujoco Gym and DMC environments, we utilized their standard, unmodified configurations. For HumanoidBench, we introduced specific modifications for certain agents.

For experiments involving SAC and OBAC on HumanoidBench, we implemented an action repeat of 2 and disabled episode termination. These adjustments were necessary because the standard tasks proved exceedingly challenging for a baseline SAC agent, as demonstrated in Figure 8. Conversely, for the FastSAC agent, which is capable of solving the original tasks, we used the standard HumanoidBench environments without these modifications.

For our comparison against TD-MPC2 on DMC environments, we used the performance data reported in the original manuscript. We therefore adhered to their experimental setup, which includes an action repeat of 2.

For main results and training curves, we report results over 10 random seeds for SAC, OBAC, and FastSAC, 5 seeds for PPO, and 3 seeds for TD-MPC2, matching the number provided in its original publication.

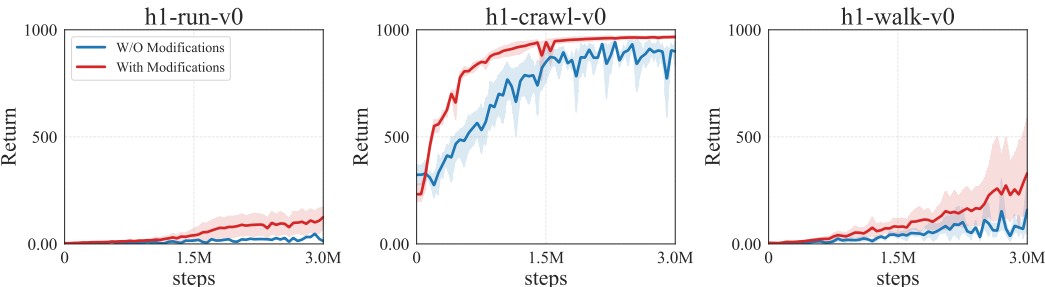

Figure 8: **Ablation of Environment Modifications for HumanoidBench.** Performance comparison of a standard SAC agent on three challenging HumanoidBench tasks with and without our modified settings (action repeat of 2 and disabled termination). The significant performance gap justifies using these modified settings for our main SAC-based experiments.

The action, observation spaces and maximal episode length of the respective environments are shown in Table 4 and Table 5.

Table 4: List of tasks from DeepMind Control and MetaWorld on which the agents were ablated. The table also contains the dimensions of action, observation space and maximal episode length.

| Task | Observation dimension | Action dimension | Max episode length |
|---|---|---|---|
| DEEPMIND CONTROL | | | |
| Dog-Trot | 223 | 38 | 1000 |
| Dog-Walk | 223 | 38 | 1000 |
| Dog-Run | 223 | 38 | 1000 |
| Humanoid-Run | 67 | 24 | 1000 |
| Humanoid-Walk | 67 | 24 | 1000 |
| Humanoid-Stand | 67 | 24 | 1000 |
| MUJOCO GYM | | | |
| HalfCheetah-v4 | 17 | 6 | 1000 |
| Ant-v4 | 27 | 8 | 1000 |
| Hopper-v4 | 11 | 3 | 1000 |
| Walker2d-v4 | 17 | 6 | 1000 |

### A.1.3 PSEUDO CODE OF SAC-ERA

To better illustrate the role of our method within the algorithmic framework, we present the pseudocode for a representative example, the Soft Actor-Critic (SAC) algorithm, adapted with ERA in Algorithm 1.

### A.1.4 HYPERPARAMETERS

We present the hyperparameters used in our experiments with SAC and PPO in Table 6

Our implementations of SAC and OBAC are heavily inspired by the official `jaxrl` repository (Kostrikov, 2021). For the network design, we follow the insights from Nauman et al. (2024) and incorporate LayerNorm (Ba et al., 2016) into the neural networks.

Our OBAC implementation is built upon the codebase provided by Kang et al. (2025). It shares the same fundamental hyperparameters as our SAC implementation, with the behavior cloning weight set to $1 \times 10^{-3}$.

Table 5: List of tasks from HumanoidBench on which the agents were ablated. The table also contains the dimensions of action, observation space and maximal episode length.

| Task | Observation dimension | Action dimension | Max episode length |
|---|---|---|---|
| h1-walk-v0 | 51 | 19 | 500 |
| h1-run-v0 | 51 | 19 | 500 |
| h1-stand-v0 | 51 | 19 | 500 |
| h1-hurdle-v0 | 51 | 19 | 500 |
| h1-stair-v0 | 51 | 19 | 500 |
| h1-crawl-v0 | 51 | 19 | 500 |
| h1hand-balance_simple-v0 | 164 | 61 | 1000 |
| h1hand-hurdle-v0 | 151 | 61 | 1000 |
| h1hand-pole-v0 | 151 | 61 | 1000 |
| h1hand-push-v0 | 163 | 61 | 1000 |
| h1hand-stair-v0 | 151 | 61 | 1000 |
| h1hand-slide-v0 | 151 | 61 | 1000 |
| h1hand-walk-v0 | 151 | 61 | 1000 |
| h1hand-run-v0 | 151 | 61 | 1000 |

Table 6: Comparison of hyperparameters for SAC and PPO.

| Hyperparameter | SAC | PPO |
|---|---|---|
| *Optimizer Settings* | | |
| Actor optimizer | | Adam |
| Actor learning rate | | $3 \times 10^{-4}$ |
| Critic optimizer | AdamW | Adam |
| Critic learning rate | | $3 \times 10^{-4}$ |
| Temperature learning rate | $3 \times 10^{-4}$ | — |
| Adam epsilon | — | $1 \times 10^{-5}$ |
| Gradient clipping | — | 0.5 |
| *Network Architecture* | | |
| Actor/Critic network | | 3-layer MLP |
| Hidden layer dimensions | (512, 512) | (64, 64) |
| Activation function | ReLU | Tanh |
| LayerNorm | True | False |
| *Algorithm Hyperparameters* | | |
| Discount factor ($\gamma$) | | 0.99 |
| Replay buffer size | $1 \times 10^{6}$ | — |
| Polyak averaging coefficient ($\tau$) | 0.005 | — |
| Initial temperature ($\alpha$) | 1.0 | — |
| Target entropy ($\mathcal{H}_0$) | $-\dim(\mathcal{A})/2$ | — |
| Gradient steps per env. step | 2 | — |
| Random exploration steps | 5,000 | — |
| GAE parameter ($\lambda$) | — | 0.95 |
| PPO clip ratio | — | 0.2 |
| Entropy coefficient | — | 0.01 |
| Batch size | 256 | 2048 |
| Mini-batch size | — | 64 |
| Log std Interval $[\sigma_{\min}, \sigma_{\max}]$ | [-8,0] for ERA, [-10,2] for baseline | |

For the PPO and PPO-ERA experiments, our implementation is based on the publicly available codebase of Li (2022). We use target entropy of $-0.3\mathcal{A}$ for main experiments on PPO-ERA.

For the TD-MPC2 baseline, we utilize the official implementation provided by the original authors. The results for comparison are also directly sourced from those reported in the official repository. We use target entropy of $-\mathcal{A}$ for main experiments on TD-MPC2-ERA.

Similarly, our implementations of FastTD3 and FastSAC are based on the official codebases provided by their respective authors. We note that our construction of FastSAC-ERA differs from the method described in the original paper; these differences are detailed in Section A.1.5.

### A.1.5  FASTSAC-ERA

The FastTD3 (Seo et al., 2025) framework demonstrated the potential of applying off-policy RL methods to massively parallel RL scenarios, achieving excellent performance on HumanoidBench.

Authors of FastTD3 also provided a FastSAC implementation, which replaced the mixed noise mechanism in FastTD3 with the standard entropy maximization objective from Soft Actor-Critic (SAC). However, they noted that this approach yielded unstable results, and hypothesized that maximizing action entropy in high-dimensional action spaces might be inherently challenging.

To address this issue, we investigated a solution based on minimal modification to the original FastTD3. Our approach, named FastSAC-ERA, is derived from FastTD3 by retaining its noise mechanism while removing the Delayed Policy Updates and incorporating an entropy constraint via ERA implementation. This method achieved performance superior to that of FastTD3.

In practice, our implementation was built directly upon the official FastTD3 codebase. The only modifications were the removal of Delayed Policy Updates and the addition of the ERA implementation at the actor's output. All other hyperparameters and implementation details were kept identical to the original FastTD3 configuration.

### A.1.6  IMPLEMENTATION DETAILS: NORMALIZED SCORE COMPUTATION

In this work, we use normalized scores to evaluate and compare algorithm performance across multiple environments. The rationale for this is that when aggregating results across a benchmark, raw scores can allow environments with disparate score ranges to have a disproportionate influence on the final result. Normalized scores mitigate this by mapping all results onto a uniform scale, enabling a more equitable comparison and aggregation.

When calculating the normalized score, we uniformly use the minimum and maximum scores achieved **among all tested algorithms** in that specific environment as the normalization bounds, rather than relying on the environment's theoretical minimum or maximum scores. This approach avoids distortions that can arise from theoretical score ranges being exceptionally large or small, thereby providing a more accurate reflection of the algorithms' relative performance in practice.

Specifically, the normalized score is calculated using the following formula:

$$\text{Normalized Score} = \frac{\text{Algorithm Score} - \text{Min Score}}{\text{Max Score} - \text{Min Score}}$$

Where Algorithm Score is the score of a given algorithm in a specific environment at a particular time, and Min Score and Max Score are the minimum and maximum scores, respectively, achieved among all participating algorithms in that same environment.

To compute an aggregate normalized score across multiple environments, we first calculate the normalized score for each algorithm within each environment. We then average these scores. This method ensures that each environment contributes equally to the final metric, providing a more comprehensive and fair assessment of overall algorithm performance.

### A.1.7  IMPLEMENTATION DETAILS: SHADING AREAS IN PLOTS

For aggregated performance plots in 2, we use 25% and 75% percentiles to create shaded areas around the mean performance curves. This choice corresponds to common practices in RL community.

For training curves of individual environments in the appendix, we use 95% confidence intervals to create shaded areas around the mean performance curves. This choice provides a clearer depiction of variability in individual environment results, which can be more pronounced than in aggregated plots.

### A.2  IMPLEMENTATION DETAILS OF IMAGE CLASSIFICATION

### A.2.1  CODE IMPLEMENTATION OF ERA IN IMAGE CLASSIFICATION

---

**Algorithm 1** Soft Actor-Critic (SAC) with ERA

1: **Initialize:** actor parameters $\theta$, critic parameters $\phi_1, \phi_2$.
2: **Initialize:** target network parameters $\phi_1' \leftarrow \phi_1, \phi_2' \leftarrow \phi_2$.
3: **Initialize:** replay buffer $\mathcal{D}$.
4: **Hyperparameters:** learning rates $\lambda_\pi, \lambda_Q, \lambda_\alpha$, target entropy $\mathcal{H}_0$, Polyak coefficient $\tau$.
5: **for** each training step **do**
6:     Sample action from the policy: $a_t \sim \pi_\theta(\cdot|s_t)$.
7:     Execute action $a_t$, observe reward $r_t$ and next state $s_{t+1}$.
8:     Store transition $(s_t, a_t, r_t, s_{t+1})$ in replay buffer $\mathcal{D}$.
9:     Sample a random minibatch of transitions $B = \{(s, a, r, s')\}$ from $\mathcal{D}$.
10:     *// Update the Q-functions (critics)*
11:     Sample next actions: $a' \sim \pi_\theta(\cdot|s')$.
12:     Compute the target Q-value by taking the minimum of the two target critics:

$$Q'_{\text{target}}(s', a') \leftarrow \min_{i=1,2} Q_{\phi_i'}(s', a')$$

13:     Compute the Q-target $y$ (matches Eq. 14):

$$y \leftarrow r + \gamma Q'_{\text{target}}(s', a')$$

14:     Update both critics by one step of gradient descent using the loss from Eq. 8:

$$\nabla_{\phi_i} \frac{1}{|B|} \sum_{(s,a,y)\in B} \frac{1}{2}\left(Q_{\phi_i}(s,a) - y\right)^2 \quad \text{for } i = 1, 2$$

15:     *// Update the policy (actor)*
16:     Sample new actions for the policy update (using reparameterization trick): $\tilde{a} \sim \pi_\theta(\cdot|s)$.
17:     Compute Q-values for the new actions using the minimum of the two critics:

$$Q_{\min}(s, \tilde{a}) \leftarrow \min_{i=1,2} Q_{\phi_i}(s, \tilde{a})$$

18:     Update the policy by one step of gradient ascent to maximize the objective from Eq. 15:

$$\nabla_\theta \frac{1}{|B|} \sum_{s \in B} Q_{\min}(s, \tilde{a})$$

19:     *// Update target networks using Polyak averaging*
20:     $\phi_i' \leftarrow \tau\phi_i + (1-\tau)\phi_i' \quad \text{for } i = 1, 2$
21: **end for**

---

We provide the implementation of ERA for image classification tasks in Listing 3. In the code, C_H corresponds to $C_{\mathcal{H}_0}$ defined in Eq. 16, and n_dims denotes the number of classes. We set $\tau = 4$ in our implementation without performing any tuning for this parameter.

### A.2.2   TRAINING SETUP

Our training for ImageNet was completed on 4 A100 GPUs, and we report the 95% confidence interval calculated from the dataset. For CIFAR-10, which requires less computation, we trained three separate runs on 3 machines, each with 4 A40 GPUs, and report the confidence interval computed from these three results to ensure maximum reproducibility.

### A.2.3   COMMANDS USED FOR EXPERIMENTS

We provide two main commands used for training in image classification. The two commands delineate the training procedures for our models under two distinct settings: one incorporating data augmentation and the other without it. The training commands were sourced directly from the reference ImageNet training script within the timm library. We employed this identical set of

Listing 3: ERA Implementation in Image Classification

```python
class ERA(nn.Module):
    def __init__(self, C_H: float, n_dims: int):
        super().__init__()
        self._tau = 4.
        self.C_H = C_H
        self.n_dims = n_dims

        self.upper_bound = math.log(self._tau) / self._tau
        assert C_H >= self.upper_bound
        self.slope = (self.upper_bound - C_H / n_dims) / (1 - 1 / n_dims)
        self.b = (C_H - self.slope) / n_dims

    def forward(self, x: torch.Tensor) -> torch.Tensor:
        """
        x: logits before softmax, shape (..., n_dims)
        return: adjusted logits before softmax, shape (..., n_dims)
        """
        h = self.slope * x.softmax(dim=-1) + self.b
        u = -1 - torch.log(h)
        new_logits = (-1 - torch.sqrt(2 * u) - 3/4 * u).to(x.dtype)

        max_values = torch.max(x, dim=-1, keepdim=True).values.detach()
        x = x - max_values
        min_values = torch.min(new_logits, dim=-1, keepdim=True).values.
            detach()
        new_logits = new_logits - min_values

        return new_logits
```

commands for training on both the ImageNet and CIFAR-10 datasets without any dataset-specific hyperparameter tuning to ensure a consistent experimental setup.

Listing 4: Command to launch training with data augmentation.

```
./distributed_train.sh 4 --data-dir ../data --dataset torch/cifar10 --
    ↪ dataset-download -b 64 --model resnet50 --sched cosine --epochs 200
    ↪  --lr 0.05 --amp --remode pixel --reprob 0.6 --aug-splits 3 --aa
    ↪ rand-m9-mstd0.5-inc1 --resplit --split-bn --jsd --dist-bn reduce
```

Listing 5: Command to launch training without data augmentation (baseline).

```
./distributed_train.sh 4 --data-dir ../data --dataset torch/cifar10 --
    ↪ dataset-download -b 64 --model resnet50 --sched cosine --epochs 200
    ↪  --lr 0.05 --amp --dist-bn reduce
```

## A.3 IMPLEMENTATION DETAILS OF LLM TRAINING

### A.3.1 CODE IMPLEMENTATION OF ERA IN LLM

We provide the core implementation of ERA in LLM in Listing 6. In the code, era_lb, era_ub and era_k corresponds to $\omega_{\text{low}}, \omega_{\text{high}}, k$ defined in Eq. 17, respectively. In the first training stage, we further apply a top-$k$ filter (retaining the 20 largest logits) within the logprobs_from_logits function to enhance training stability. In addition, we provide advantage scaling as an option, as removing such scaling would reduce the update to a pure logit shift. The impact of advantage scaling is discussed in C.3.1.

Listing 6: ERA Implementation in LLM

```
length = response_mask.sum(dim=-1)
k_per_sample = (0.2 * length).long().clamp(min=1)

mean_top_entropy = []
masked_entropy = entropy.masked_fill(~response_mask.bool(), float("-inf")
    )
for b in range(entropy.size(0)):
    k = k_per_sample[b].item()
    top_entropy_b, _ = torch.topk(masked_entropy[b], k)
    mean_top_entropy.append(top_entropy_b.mean())

mean_top_entropy = torch.stack(mean_top_entropy).unsqueeze(-1)
cond_A = (mean_top_entropy < era_lb) & (advantages > 0)
cond_B = (mean_top_entropy > era_ub) & (advantages > 0)

logits[cond_A] = logits[cond_A] * era_k
logits[cond_B] = logits[cond_B] / era_k

if adv_scale:
    advantages[cond_A] = advantages[cond_A] / era_k
    advantages[cond_B] = advantages[cond_B] * era_k

log_prob = logprobs_from_logits(logits)
```

### A.3.2 HYPERPARAMETERS

For GRPO, GRPO w/ 20% Forking Tokens, ERA, we use a training batch size of 256 and a mini batch size of 256 in the verl configuration, which results in a on-policy setting. For KL-Cov and Clip-Cov, we use a training batch size of 256 and a mini batch size of 32, and other hyperparameters are consistent with their original paper. GRPO + Entropy Loss uses an entropy regularization term with coefficient $0.002$. The learning rate is $10^{-6}$ and no learning rate warm-up or scheduling is applied. We also utilize dynamic sampling to enhance training efficiency. Since our setting is on-policy, the clip ratio is irrelevant. The maximum response length is $8192$ with no overlong reward shaping. For Qwen2.5-Math-1.5B, we use MATH problems of levels 3–5 as the training set in this experiment since DAPO-Math-17K is too difficult.

The hyperparameters of ERA are fixed to $\omega_{\text{low}} = 0.45$, $\omega_{\text{high}} = 3.0$, and $k = 2$ across all settings, without any tuning. These values are chosen with reference to the initial entropy of the model, $H_{\text{resp}} \approx 1.5$, such that $\omega_{\text{low}}$ and $\omega_{\text{high}}$ lie below and above this value, respectively. We believe the results are not sensitive to these ERA hyperparameters.

## B  PROOFS AND DERIVATIONS

### B.1  PROOF OF ENTROPY BOUND IN CONTINUOUS SPACE

In this section, we provide a rigorous analysis of the Entropy Regularizing Activation (ERA) for continuous control. We proceed in three steps:

1. **Static Guarantee:** We prove that the ERA functional form structurally guarantees the entropy lower bound, provided the parameter $\hat{\delta}$ is sufficiently large.

2. **Dynamic Convergence:** We prove that the learnable parameter $\hat{\delta}$ converges to the required value under coupled policy updates, using two-timescale stochastic approximation theory.

3. **Non-negativity of Bias:** We prove that the entropy compensation term $\delta(s)$ is non-negative for both Squashed and Truncated Gaussian distributions.

### B.1.1 SETTING AND DEFINITIONS

Recall the continuous form of ERA. For a state $s$ and network outputs $(\mu(s;\theta), \hat{\sigma}(s;\theta))$, the activation maps to the final standard deviation $\sigma'$:

$$\sigma_i'(\theta, \hat{\delta}; s) = \exp\left[\max\left(\log\sigma_{\max} + (\mathcal{H}_0 + \hat{\delta} - C)\frac{e^{\hat{\sigma}_i(s;\theta)}}{\sum_{j=1}^{D} e^{\hat{\sigma}_j(s;\theta)}}, \log\sigma_{\min}\right)\right], \quad (20)$$

where $C = D\log\sqrt{2\pi e} + D\log\sigma_{\max}$. Here, $\mathcal{H}_0$ is the target entropy, and $\hat{\delta}$ is a learnable parameter intended to compensate for the entropy bias $\delta_{\text{bias}}(s)$ induced by the bounding function (e.g., Tanh or Truncation).

The actual entropy of the final policy $\pi_{\theta,\hat{\delta}}$ is given by:

$$\mathcal{H}(\pi_{\theta,\hat{\delta}}(\cdot|s)) = \mathcal{H}_{\text{Gaussian}}(\mu(s;\theta), \text{diag}(\sigma'(\theta, \hat{\delta}; s))) - \delta_{\text{bias}}(s), \quad (21)$$

where $\mathcal{H}_{\text{Gaussian}} = \frac{D}{2}\log(2\pi e) + \sum_{i=1}^{D}\log\sigma_i'$.

### B.1.2 STATIC ENTROPY BOUND

**Proposition 1.** *Given a target entropy $\mathcal{H}_0$ and a residual entropy parameter $\hat{\delta} \geq \delta_{bias}(s)$, the policy defined by Eq. equation 20 satisfies $\mathcal{H}(\pi) \geq \mathcal{H}_0$, and $\sigma'$ is strictly bounded within $[\sigma_{\min}, \sigma_{\max}]$.*

*Proof.* The entropy constraint $\mathcal{H}(\pi) \geq \mathcal{H}_0$ is equivalent to $\mathcal{H}_{\text{Gaussian}} - \delta_{\text{bias}}(s) \geq \mathcal{H}_0$. Substituting the Gaussian entropy formula, we require:

$$\sum_{i=1}^{D}\log\sigma_i' \geq \mathcal{H}_0 + \delta_{\text{bias}}(s) - \frac{D}{2}\log(2\pi e). \quad (22)$$

From Eq. equation 20, noting that the $\max$ operator ensures $\sigma_i' \geq \sigma_{\min}$, we consider the term inside the exponent:

$$\sum_{i=1}^{D}\log\sigma_i' \geq \sum_{i=1}^{D}\left[\log\sigma_{\max} + (\mathcal{H}_0 + \hat{\delta} - C)\frac{e^{\hat{\sigma}_i}}{\sum_{j=1}^{D} e^{\hat{\sigma}_j}}\right] \quad (23)$$

$$= D\log\sigma_{\max} + (\mathcal{H}_0 + \hat{\delta} - C)\underbrace{\sum_{i=1}^{D}\frac{e^{\hat{\sigma}_i}}{\sum_{j=1}^{D} e^{\hat{\sigma}_j}}}_{1} \quad (24)$$

$$= D\log\sigma_{\max} + \mathcal{H}_0 + \hat{\delta} - (D\log\sqrt{2\pi e} + D\log\sigma_{\max}) \quad (25)$$

$$= \mathcal{H}_0 + \hat{\delta} - \frac{D}{2}\log(2\pi e). \quad (26)$$

Comparing this result with Eq. equation 22, we see that if $\hat{\delta} \geq \delta_{\text{bias}}(s)$, the condition is satisfied. Furthermore, the functional form explicitly constrains outputs via $\max(\cdot, \log\sigma_{\min})$ and $\log\sigma_{\max}$ (in the softmax upper bound), ensuring $\sigma' \in [\sigma_{\min}, \sigma_{\max}]$. $\qquad\square$

### B.1.3 CONVERGENCE UNDER COUPLED UPDATES

We now prove that $\hat{\delta}$ automatically converges to the necessary value to satisfy the constraint, even when the policy parameters $\theta$ are updating simultaneously. We utilize the framework of two-timescale stochastic approximation (Borkar, 1997).

**Update Rule.** The parameter $\hat{\delta}$ is updated to minimize the loss $\mathcal{L}(\hat{\delta}) = \hat{\delta}(\mathcal{H}(\pi) - \mathcal{H}_0)$, leading to the gradient update:

$$\hat{\delta}_{t+1} \leftarrow \hat{\delta}_t + \beta_t(\mathcal{H}_0 - \mathcal{H}(\pi_{\theta_t, \hat{\delta}_t})). \quad (27)$$

**Assumptions.**

(A1) **Regularity:** The mappings $\mu(\theta)$ and $\hat{\sigma}(\theta)$ are continuously differentiable with bounded gradients.

(A2) **Non-saturation:** The optimization operates in a regime where the ERA activation is not fully saturated at the lower bound $\sigma_{\min}$ for all dimensions. This ensures $\frac{\partial \mathcal{H}}{\partial \hat{\delta}} > 0$.

(A3) **Timescale Separation:** Let $\{\alpha_t\}$ and $\{\beta_t\}$ be the step sizes for $\theta$ and $\hat{\delta}$ respectively. We assume $\hat{\delta}$ updates on a faster timescale: $\lim_{t \to \infty} \frac{\alpha_t}{\beta_t} = 0$, alongside standard Robbins-Monro conditions ($\sum \alpha_t = \infty$, $\sum \alpha_t^2 < \infty$, etc.).

**Lemma 1** (Monotonicity). *Under (A2), for fixed $\theta$, $\mathcal{H}(\pi_{\theta,\hat{\delta}})$ is strictly monotonically increasing with respect to $\hat{\delta}$.*

*Proof.* $\frac{\partial \log \sigma_i'}{\partial \hat{\delta}} = \frac{e^{\hat{\sigma}_i}}{\sum e^{\hat{\sigma}_j}} > 0$. Since $\mathcal{H} \propto \sum \log \sigma_i'$, it follows that $\frac{\partial \mathcal{H}}{\partial \hat{\delta}} > 0$. $\qquad\square$

**Proposition 2** (Global Asymptotic Stability). *Under the stated assumptions, the coupled iteration $(\theta_t, \hat{\delta}_t)$ converges such that $\hat{\delta}_t$ asymptotically tracks the equilibrium $\delta^*(\theta_t)$ satisfying $\mathcal{H}(\pi_{\theta_t,\delta^*}) = \mathcal{H}_0$.*

*Proof.* We analyze the system dynamics in two timescales:

**1. Fast Timescale ($\hat{\delta}$-update):** Since $\alpha_t/\beta_t \to 0$, $\theta$ is viewed as quasi-static. The dynamics of $\hat{\delta}$ follow the ODE: $\dot{\hat{\delta}}(t) = \mathcal{H}_0 - \mathcal{H}(\pi_{\theta,\hat{\delta}(t)})$. Define the Lyapunov function $V(\hat{\delta}) = \frac{1}{2}(\hat{\delta} - \delta^*(\theta))^2$, where $\delta^*(\theta)$ is the unique root of $\mathcal{H}(\pi_{\theta,\delta}) = \mathcal{H}_0$. The time derivative is $\dot{V} = (\hat{\delta} - \delta^*)(\mathcal{H}_0 - \mathcal{H}(\hat{\delta}))$. By monotonicity, if $\hat{\delta} > \delta^*$, then $\mathcal{H} > \mathcal{H}_0$, implying $\dot{V} < 0$. Thus, $\hat{\delta}$ converges globally to $\delta^*(\theta)$.

**2. Slow Timescale ($\theta$-update):** By the theory of two-timescale stochastic approximation, $\hat{\delta}_t$ tracks $\delta^*(\theta_t)$ almost surely. The policy update $\theta_t$ effectively proceeds along the manifold $\mathcal{M} = \{(\theta, \hat{\delta}) \mid \mathcal{H}(\pi_{\theta,\hat{\delta}}) \approx \mathcal{H}_0\}$, solving the constrained optimization problem.

**3. Robustness (Finite Step Sizes):** In practice, if $\alpha_t/\beta_t$ is bounded but non-zero, the system is Input-to-State Stable (ISS). The policy update $\dot{\theta}$ acts as a bounded disturbance. The entropy error is bounded by the ratio of the disturbance magnitude to the controller gain:

$$\limsup_{t \to \infty} |\mathcal{H}(\pi_t) - \mathcal{H}_0| \leq C \cdot \sup_t \alpha_t.$$

This guarantees that the entropy remains bounded within a small neighborhood of $\mathcal{H}_0$. $\qquad\square$

### B.1.4 NON-NEGATIVITY OF THE BIAS TERM

Finally, we show that the bias term $\delta_{\text{bias}}(s)$ in Eq. equation 20 is non-negative, justifying the form of our compensation.

**Case 1: Tanh-squashed Gaussian.** The bias is given by $\delta_{\tanh} = -\mathbb{E}[\sum \log(1 - \tanh^2(u_i))]$. Since $1 - \tanh^2(u) \in (0, 1]$, its logarithm is non-positive. Therefore, the negative expectation is non-negative: $\delta_{\tanh} \geq 0$.

**Case 2: Truncated Gaussian (TN).** Let $\pi_{\text{orig}} = \mathcal{N}(\mu, \sigma^2)$ be the original Gaussian distribution and $\pi_{\text{TN}}$ be the truncated distribution restricted to the interval $[-1, 1]$. The bias is defined as the entropy difference: $\delta_{\text{TN}} = h(\pi_{\text{orig}}) - h(\pi_{\text{TN}})$.

To rigorously prove $\delta_{\text{TN}} \geq 0$, we introduce a *moment-matched Gaussian* distribution $\bar{\pi} = \mathcal{N}(m_{\text{TN}}, v_{\text{TN}})$, where $m_{\text{TN}}$ and $v_{\text{TN}}$ denote the true mean and variance of the truncated distribution $\pi_{\text{TN}}$. The proof proceeds in two steps:

1. **Maximum Entropy of Gaussians ($h(\bar{\pi}) \geq h(\pi_{\text{TN}})$):** Among all continuous probability distributions with a fixed variance, the Gaussian distribution maximizes differential entropy. Since the constructed distribution $\bar{\pi}$ is Gaussian and shares the exact same variance $v_{\text{TN}}$ as $\pi_{\text{TN}}$, its entropy must be greater than or equal to that of $\pi_{\text{TN}}$:

$$h(\bar{\pi}) \geq h(\pi_{\text{TN}}). \tag{28}$$

2. **Variance Reduction by Truncation** ($h(\pi_{\mathbf{orig}}) \geq h(\bar{\pi})$)**:** The entropy of a Gaussian distribution is monotonically increasing with respect to its variance, given by $h(\mathcal{N}(\cdot, \sigma^2)) = \frac{1}{2}\log(2\pi e \sigma^2)$. Therefore, showing $h(\pi_{\text{orig}}) \geq h(\bar{\pi})$ is equivalent to proving that truncation reduces variance, i.e., $v_{\text{TN}} \leq \sigma^2$.

   We prove this inequality analytically by examining the sensitivity of the truncated mean $m_{\text{TN}}$ with respect to the original location parameter $\mu$.

   First, it is a known result in truncated statistics that the derivative of the truncated mean with respect to the location parameter $\mu$ is exactly the ratio of the truncated variance to the original variance:

   $$\frac{\partial m_{\text{TN}}}{\partial \mu} = \frac{v_{\text{TN}}}{\sigma^2}. \tag{29}$$

   Second, we bound this derivative using properties of log-concave functions. The truncated mean can be expressed in terms of the normalization constant $Z(\mu) = \Phi(\beta) - \Phi(\alpha)$ as:

   $$m_{\text{TN}} = \mu + \sigma^2 \frac{\partial \ln Z(\mu)}{\partial \mu}. \tag{30}$$

   Differentiating this expression with respect to $\mu$ yields:

   $$\frac{\partial m_{\text{TN}}}{\partial \mu} = 1 + \sigma^2 \frac{\partial^2 \ln Z(\mu)}{\partial \mu^2}. \tag{31}$$

   The normalization term $Z(\mu)$ can be viewed as the convolution of the standard normal PDF $\phi(\cdot)$ and the indicator function of the interval $[-1, 1]$. Since both the Gaussian PDF and the indicator function of a convex set are log-concave functions, and the convolution of log-concave functions preserves log-concavity (Boyd & Vandenberghe, 2004), $Z(\mu)$ is log-concave in $\mu$.

   By definition, the second derivative of the logarithm of a concave function is non-positive. Thus:

   $$\frac{\partial^2 \ln Z(\mu)}{\partial \mu^2} \leq 0. \tag{32}$$

   Substituting this inequality back into Eq. equation 31, we obtain the upper bound:

   $$\frac{\partial m_{\text{TN}}}{\partial \mu} \leq 1. \tag{33}$$

   Finally, combining this bound with Eq. equation 29, we arrive at:

   $$\frac{v_{\text{TN}}}{\sigma^2} \leq 1 \implies v_{\text{TN}} \leq \sigma^2. \tag{34}$$

   This strictly implies $h(\pi_{\text{orig}}) \geq h(\bar{\pi})$.

**Conclusion:** Summing the inequalities established in steps 1 and 2, we have:

$$h(\pi_{\text{orig}}) \geq h(\bar{\pi}) \geq h(\pi_{\text{TN}}). \tag{35}$$

Consequently, the bias term $\delta_{\text{TN}} = h(\pi_{\text{orig}}) - h(\pi_{\text{TN}})$ is guaranteed to be non-negative.

## B.2 PROOF OF ENTROPY BOUND IN DISCRETE SPACE

Recall the discrete form of ERA:

$$z' = h^{-1}\left[\max\left(\frac{\log \tau}{\tau} + \left(C_{\mathcal{H}_0} - n\frac{\log \tau}{\tau}\right)\frac{1}{D-1}\left(1 - \frac{e^{z_i}}{\sum_{j=1}^{D} e^{z_j}}\right), 0\right)\right]$$

Before we delve into the proof of its entropy bound, we first provide some insights into the design of ERA in the context of vision tasks. To adapt the entropy constraint function from continuous spaces for discrete domains, our initial idea was to have the network output the entropy of individual components rather than their logits. However, this direct approach is problematic because the function $H(p) = -p \ln p$ is non-monotonic over the interval $[0, 1]$. This ambiguity means a given entropy

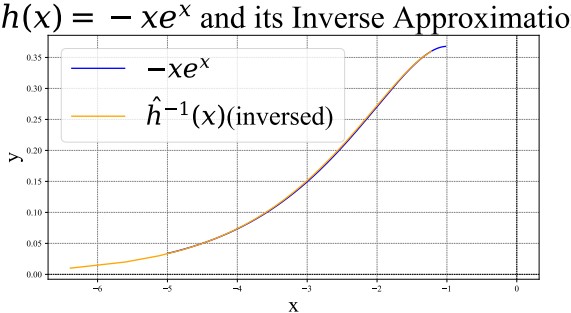

Figure 9: Plot of $h(x) = -xe^x$ and its Inverse Approximation $\hat{h}^{-1}(x)$. We reverse the x and y values for $\hat{h}^{-1}$ to visualize the inverse relationship(The more two curves overlap, the closer the approximation). We can see that the approximation is very close to the true inverse function.

value cannot be uniquely mapped back to its corresponding probability; for instance, an entropy of 0 could correspond to a probability of either 0 or 1.

To resolve this ambiguity, we introduce a scaling factor $\tau > e$ and consider a "$\tau$-divided distribution," where each probability is scaled down by $\tau$ (We note that the $\tau$-divided distribution is not actually a valid probability distribution, but a tool for analysis). By selecting $\tau > e$, we ensure that the function $-p \ln p$ is strictly monotonically increasing on the interval $[0, 1/\tau]$. This establishes a one-to-one mapping, allowing for the unique recovery of a probability value from its entropy within this restricted range. Therefore, our network is designed to output the entropy of this $\tau$-divided distribution. We then map these entropy values back to logits using an inverse function, $h^{-1}$. Note that an entropy value is $\mathcal{H} = -p \ln p$ for some $p \in [0, 1/\tau]$. From logits to entropy, we have the following mapping function:

$$h(x) = -x \ln x \circ \exp(x) = -\exp(x) \cdot x \tag{36}$$

Therefore, the inverse function $h^{-1}$ maps entropy values back to logits is exactly the inverse of $-x \exp(x)$, we have $h^{-1}(x) = W(-x)$. $W$ is known as the Lambert W function (Corless et al., 1996). Since there is no closed-form solution for the Lambert W function, we utilize a numerical approximation $\hat{h}^{-1}(x) = -\frac{1}{4} - \sqrt{2(-1 - \ln(x))} + \frac{3}{4} \ln x$. We derive this approximation from $(1 + x + \ln(-x))^{-1} \approx -1 - \sqrt{2x} - \frac{3}{4}x$. Here "-1" denotes the inverse function. A final normalization step is required because the resulting probabilities from this inverse mapping do not inherently sum to one.

Crucially, we have proven that the entropy loss during this normalization process is bounded. By leveraging the continuous-space entropy constraint function to ensure the initial output entropy is above a threshold $C_{\mathcal{H}_0}$, we can guarantee that the entropy of the final discrete distribution will also exceed $C_{\mathcal{H}_0}$. This constitutes the core mechanism behind the implementation of ERA in discrete spaces.

**Proposition 3.** *Given a target entropy $\mathcal{H}_0$ and a hyperparameter $\tau \geq e$, the policy defined by Eq. 16 has entropy $\mathcal{H}(\pi) \geq \mathcal{H}_0$.*

*Proof.* We denote $\kappa = \max(\frac{\log \tau}{\tau} + (C_{\mathcal{H}_0} - n\frac{\log \tau}{\tau})\frac{1}{D-1}(1 - \frac{e^{z_i}}{\sum_{j=1}^{D} e^{z_j}}), 0)$. Similar to the continuous case, we have $\kappa$ bounded within $[0, \frac{\log \tau}{\tau}]$ and $\sum_{i=1}^{D} \kappa_i \geq C_{\mathcal{H}_0}$. We denote the probability of the final softmax policy as $p = \text{softmax}(z') = \frac{e^{z'}}{\sum_{j=1}^{D} e^{z'_j}}$. Then we have:

$$\mathcal{H}(\pi) = -\sum_{i=1}^{D} p_i \log p_i$$

$$= -\frac{\sum_{i=1}^{D} e^{h^{-1}(\kappa_i)} h^{-1}(\kappa_i)}{\sum_{j=1}^{D} e^{h^{-1}(\kappa_j)}} + \log(\sum_{j=1}^{D} e^{h^{-1}(\kappa_j)})$$

$$\geq 1 + \log(-\sum_{i=1}^{D} e^{h^{-1}(\kappa_i)} h^{-1}(\kappa_i)) \tag{37}$$

Recall that $h = -x \ln x \circ e^x$, so $h^{-1} = \ln \circ (-x \ln x)^{-1}$. Hence we have:

$$\mathcal{H}(\pi) \geq 1 + \log(-\sum_{i=1}^{D} e^{h^{-1}(\kappa_i)} h^{-1}(\kappa_i))$$

$$= 1 + \log(\sum_{i=1}^{D} \kappa_i) \geq 1 + \log(C_{\mathcal{H}_0}) = \mathcal{H}_0 \qquad (38)$$

$\square$

### B.3 PROOF OF ENTROPY BOUND IN LLMS

Recall the definition of the ERA instantiation for LLMs:

$$z' = \begin{cases} kz & H_{\text{resp}} < \omega_{\text{low}}, A_t > 0, \\ z & (\omega_{\text{low}} \leq H_{\text{resp}} \leq \omega_{\text{high}}, A_t < 0) \text{ or } A_t > 0, \\ \frac{1}{k}z & H_{\text{resp}} > \omega_{\text{high}}, A_t > 0, \end{cases}$$

and

$$A'_t = \begin{cases} \frac{1}{k}A_t & H_{\text{resp}} < \omega_{\text{low}}, A_t > 0, \\ A_t & (\omega_{\text{low}} \leq H_{\text{resp}} \leq \omega_{\text{high}}, A_t < 0) \text{ or } A_t > 0, \\ kA_t & H_{\text{resp}} > \omega_{\text{high}}, A_t > 0, \end{cases}$$

where $z$ are the logits, $A_t$ the advantages, and $H_{\text{resp}}$ is the average entropy of the top $20\%$ of tokens with the highest entropy in the response.

These transformations are applied after sampling. The modified policy-gradient objective is therefore

$$J(\theta) = \mathbb{E}_t[\mathbb{E}_{a_t \sim \pi_\theta(\cdot|s_t)} \log \pi'_\theta(a_t|s_t)A'_t]$$

Intuitively, when the entropy is too low, ERA sharpens the policy; when it is too high, ERA flattens it. By rescaling the advantages of modified tokens, we show below that ERA is equivalent to augmenting the vanilla policy-gradient objective with an adaptive KL regularizer. This KL term guarantees that the entropy of responses remains in the interval $[\omega_{\text{low}}, \omega_{\text{high}}]$, preventing entropy collapse. Under mild assumptions, we derive a positive entropy lower bound.

Fixing the state $s_t$, denote $\pi_a = \pi_\theta(a|s_t)$, $\pi'_a = \pi'_\theta(a|s_t)$, and $A_a$ the advantage of action $a$. The entropy is $H = -\sum_a \pi_a \log \pi_a$. We first derive the gradient of the entropy.

**Lemma 2.**

$$\frac{\partial H}{\partial z_a} = \sum_{a'} -\frac{\partial \log \pi_{a'}}{\partial z_a}(\pi_{a'} \log \pi_{a'} + \pi_{a'})$$

$$= \sum_{a'} -([a = a'] - \pi_a)(\pi_{a'} \log \pi_{a'} + \pi_{a'})$$

$$= -\pi_a(\log \pi_a + H). \qquad (39)$$

We also define the $\pi$-weighted covariance that will be used later:

**Definition 1.** *Define the $\pi$-weighted covariance for two vectors $x = (x_a)$, $y = (y_a)$ by*

$$\text{Cov}_\pi(x, y) = \sum_a \pi_a x_a y_a - \Big(\sum_a \pi_a x_a\Big)\Big(\sum_a \pi_a y_a\Big).$$

Now we show our main result:

**Proposition 4.** *Let $\pi_\theta$ be the base policy and $\pi'_\theta$ the ERA-adjusted policy from Eq. equation 17. Suppose that:*

(i) *(Logit approximation) The change in entropy can be approximated by treating logits $z$ as the effective policy parameters and using first-order (infinitesimal) sensitivity of entropy w.r.t. $z$.*

(ii) *(Positive advantage mass) The aggregated positive advantage restricted to the tokens considered in $H_{resp}$,*

$$C(s_t) = \sum_{a, A_a > 0} \pi_a A_a,$$

*satisfies $C(s_t) \geq \gamma$ for some $\gamma > 0$.*

(iii) *(Bounded response entropy) In some intermediate point of the training process, $H_{resp}$ has a lower bound $H_{min}$ and an upper bound $\omega_{high}$.*

(iv) *(Bounded PG-induced entropy decrease) We assume the vanilla policy-gradient term's expected effect on entropy is bounded as*

$$\mathbb{E}[\mathrm{Cov}_\pi(\pi_a A_a, \log \pi_a)] \leq \alpha H,$$

*for some $\alpha \geq 0$ and any fixed $H$, where $H$ denotes the entropy of the current policy $\pi$.*

(v) *(Bounded KL-induced entropy decrease) We assume there exists a constant $B_k > 0$ (that depends on $k$ and $H_{min}$) such that*

$$\mathrm{Cov}_\pi(\pi_a' - \pi_a, \log \pi_a) \geq B_k H,$$

*If $\gamma B_k - \alpha > \beta$ for $\beta > 0$, then there exists a constant $\mathcal{H}_0 > 0$ such that the response entropy satisfies*

$$\mathbb{E}[H_{resp}] \geq \mathcal{H}_0$$

*under ERA updates using a gradient flow approximization.*

*Proof.* When $H_{\mathrm{resp}} < \omega_{\mathrm{low}}$, ERA sharpens positively advantaged actions. Following the derivation, the ERA-adjusted gradient satisfies

$$
\begin{aligned}
&\frac{\partial}{\partial z_a} \mathbb{E}_{a' \sim \pi} \log \pi_{a'}' A_{a'}' \\
&= \frac{\partial}{\partial z_a} \mathbb{E}_{a' \sim \pi} \left( [A_{a'} > 0] \log \pi_{a'}' \frac{1}{k} A_{a'} + [A_{a'} < 0] \log \pi_{a'} A_{a'} \right) \\
&= \mathbb{E}_{a' \sim \pi} \left( [A_{a'} > 0] \frac{\partial \log \pi_{a'}'}{\partial z_a'} \frac{\partial z_a'}{\partial z_a} \frac{1}{k} A_{a'} + [A_{a'} < 0] \frac{\partial \log \pi_{a'}}{\partial z_a} A_{a'} \right) \\
&= \mathbb{E}_{a' \sim \pi} \left( [A_{a'} > 0]([a' = a] - \pi_{a'}') A_{a'} + [A_{a'} < 0]([a' = a] - \pi_{a'}) A_{a'} \right) \\
&= \pi_a A_a - \pi_a' \sum_{a', A_{a'} > 0} \pi_{a'} A_{a'} - \pi_a \sum_{a', A_{a'} < 0} \pi_{a'} A_{a'},
\end{aligned}
\tag{40}
$$

Since the expectation of advantage is zero, and we have defined $C(s_t) = \sum_{a', A_{a'} > 0} \pi_{a'} A_{a'}$, yielding

$$\frac{\partial}{\partial z_a} \mathbb{E}_{a' \sim \pi} \log \pi_{a'}' A_{a'}' = \pi_a A_a - C(s_t)(\pi_a' - \pi_a). \tag{41}$$

For the vanilla policy-gradient loss, this reduces to

$$\frac{\partial}{\partial z_a} \mathbb{E}_{a' \sim \pi} \log \pi_{a'} A_{a'} = \pi_a A_a \tag{42}$$

Meanwhile, by a similar derivation, the gradient of the KL divergence is

$$\frac{\partial}{\partial z_a} \mathrm{KL}[\pi' \| \pi] = -\frac{\partial}{\partial z_a} \mathbb{E}_{a' \sim \pi'} \log \pi_{a'} = \pi_a - \pi_a'. \tag{43}$$

Thus, by combining equation 41, equation 42 and equation 43, the ERA-adjusted objective can be written as

$$J'(\theta) = \mathbb{E}_t[\mathbb{E}_{a_t \sim \pi_\theta(\cdot|s_t)} \underbrace{\log \pi_\theta(a_t|s_t)A_t}_{J_{\text{PG}}} + \text{sg}(C(s_t)) \underbrace{\text{KL}[\pi'_\theta(\cdot|s_t), \pi_\theta(\cdot|s_t)]}_{J_{\text{KL}}}], \tag{44}$$

where the $\text{sg}(\cdot)$ denotes the stop gradient operator. For the other case $\omega_{\text{low}} \leq$ (we have assumed that $H_{\text{resp}} \leq \omega_{\text{high}}$, the same structure holds; only the definition of $\pi'_\theta$ changes. Hence, ERA is equivalent to a policy gradient objective augmented with an adaptive KL regularizer that sharpens or flattens the distribution depending on $H_{\text{resp}}$ and also the value of $C(s_t)$.

We will evaluate the instantaneous directional derivative of entropy along these gradient directions (this corresponds to the first-order change in entropy under an infinitesimal step in the indicated direction).

Using equation 39, the first-order change of entropy caused by $J_{\text{PG}}$ is

$$\begin{aligned}
\Delta H_{\text{PG}} &= \sum_a \frac{\partial H}{\partial z_a} \cdot \pi_a A_a \\
&= \sum_a -\pi_a(\log \pi_a + H) \cdot \pi_a A_a \\
&= -\sum_a \pi_a^2 A_a(\log \pi_a + H) \\
&= -\text{Cov}_\pi(\pi_a A_a, \log \pi_a). \tag{45}
\end{aligned}$$

By assumption (iv) this term is bounded below by $-\alpha H$:

$$\mathbb{E}[\Delta H_{\text{PG}}] \geq -\alpha H.$$

Thus the vanilla policy-gradient component can decrease entropy, but by no more than $\alpha H$ in magnitude.

Similarly, the KL-term directional derivative is

$$\begin{aligned}
\Delta H_{\text{KL}} &= \sum_a \frac{\partial H}{\partial z_a} \cdot (\pi_a - \pi'_a) \\
&= \sum_a -\pi_a(\log \pi_a + H) \cdot (\pi_a - \pi'_a) \\
&= \sum_a \pi_a(\pi'_a - \pi_a)(\log \pi_a + H) \\
&= \text{Cov}_\pi(\pi'_a - \pi_a, \log \pi_a) \tag{46}
\end{aligned}$$

By assumption (v) we have $\text{Cov}_\pi(\pi'_a - \pi_a, \log \pi_a) \geq B_k H$. Using assumption (ii) $C(s_t) \geq \gamma$ therefore yields

$$C(s_t)\Delta H_{\text{KL}} \geq \gamma B_k H.$$

Combining the two contributions,

$$\mathbb{E}[\Delta H] = \mathbb{E}[\Delta H_{\text{PG}} + C(s_t)\Delta H_{\text{KL}}] \geq -\alpha H + \gamma B_k H = (\gamma B_k - \alpha)H.$$

By the hypothesis $\gamma B_k - \alpha > \beta$ we have $\Delta H > \beta H$ whenever $H > 0$ and $H$ is in the sharpening regime. Thus, if $H_{\text{resp}}$ drops below $\omega_{\text{low}}$, the ERA-induced update produces a positive first-order increase in entropy proportional to $H_{\text{resp}}$. Consequently the dynamics push $H_{\text{resp}}$ upward until it leaves the sharpening regime (i.e., until $H_{\text{resp}} \geq \omega_{\text{low}}$ or the KL-term no longer sharpens).

Formally, when $H_{\text{resp}} < \omega_{\text{low}}$ we have $\mathbb{E}[\Delta H] \geq \beta H_{\text{resp}}$, and when $H_{\text{resp}} \geq \omega_{\text{low}}$ we have $\mathbb{E}[\Delta H] \geq -\alpha H_{\text{resp}}$. Therefore, the overall expected change in entropy is at least

$$\beta \mathbb{E}_{H_{\text{resp}} < \omega_{\text{low}}}[H_{\text{resp}}] - \alpha \mathbb{E}_{H_{\text{resp}} \geq \omega_{\text{low}}}[H_{\text{resp}}] \tag{47}$$

Applying Markov's inequality gives $\Pr(H_{\text{resp}} \geq \omega_{\text{low}}) \leq \mu/\omega_{\text{low}}$, where $\mu = \mathbb{E}[H_{\text{resp}}]$. Further, by assumption (iii): $H_{\min} \leq H_{\text{resp}} \leq \omega_{\text{high}}$, we obtain the sufficient condition to make the expected entropy change non-negative:

$$\beta \geq \alpha \cdot \frac{\mu\omega_{\text{high}}}{(\omega_{\text{low}} - \mu)H_{\min}}.$$

The entropy is expected to increase ($\mathbb{E}[\Delta H] \geq 0$) whenever the term in this inequality holds. Solving for $\mu$, we find the condition:

$$\mu \leq \frac{\beta\omega_{\text{low}}H_{\min}}{\alpha\omega_{\text{high}} + \beta H_{\min}}.$$

Then we set $\mathcal{H}_0$ as

$$\mathcal{H}_0 = \frac{\beta\omega_{\text{low}}H_{\min}}{\alpha\omega_{\text{high}} + \beta H_{\min}}.$$

Under the gradient-flow approximation, we have

$$\frac{d}{dt}\mathbb{E}[H_{\text{resp}}] \geq 0 \quad \text{whenever} \quad \mathbb{E}[H_{\text{resp}}] \leq \mathcal{H}_0.$$

By assumption (iii), there exists a time $t_0$ such that $\mathbb{E}[H_{\text{resp}}] \geq \mathcal{H}_0$ at $t_0$. Then, by the principle of differential inequalities, the ERA objective ensures that $\mathbb{E}[H_{\text{resp}}]$ stays above this threshold for all $t \geq t_0$.

$\square$

We now justify the assumptions made in Proposition 4.

(i) The first assumption, namely approximating entropy differences by treating logits as policy parameters, is standard and also adopted by (Cui et al., 2025b). This simplification is essential for analytical tractability; without it, the theoretical analysis of the model's behavior becomes prohibitively complex.

(ii) Recall that $C(s_t) = \sum_{a,A_a>0} \pi_a A_a$ measures the aggregated positive advantage, which reflects the "importance" of a token. Intuitively, $C(s_t)$ indicates whether a token should remain explorative and thus be subject to entropy regularization. We assume that for important tokens, $C(s_t)$ is uniformly bounded below by some constant $\gamma > 0$.

(iii) Empirically, our training curves show that responses with $H_{\text{resp}} > \omega_{\text{high}}$ vanish rapidly, and such cases contribute negligibly to the average entropy. This supports the assumption $H_{\text{resp}} \leq \omega_{\text{high}}$. Moreover, in the early stage of training, the highest entropy tokens (top 20%) contain a lot of exploratory tokens, exhibiting a large average entropy, motivating the assumption of a positive lower bound $H_{\text{resp}} \geq H_{\min}$.

(iv) It is provable that

$$\text{Cov}_\pi(\pi_a A_a, \log \pi_a) \leq H,$$

where $H$ denotes the entropy. In practice this upper bound is rarely tight, and we assume instead a looser bound with a small constant $\alpha \in (0, 1)$.

(v) In our regime, the entropy is low enough that the token with the largest probability dominates (with probability $\geq 0.6$). In this setting, the covariance is large enough and is proportional to the entropy $H$.

In practice, the observed entropy lower bound is higher than the theoretical bound derived in Proposition 4, owing both to the looseness of the Markov inequality used in the derivation and to the fact that the tokens outside $H_{\text{resp}}$ (bottom 80%) also get an entropy boost.

## C  ADDITIONAL RESULTS

### C.1  ADDITIONAL RESULTS ON CONTINUOUS CONTROL TASKS

In this subsection, we provide additional experimental results on continuous control tasks to further validate the effectiveness of our proposed method, ERA, and to find more insights regarding entropy regularization in reinforcement learning.

### C.1.1 TRUNCATED GAUSSIAN IS MORE STABLE THAN TANH GAUSSIAN

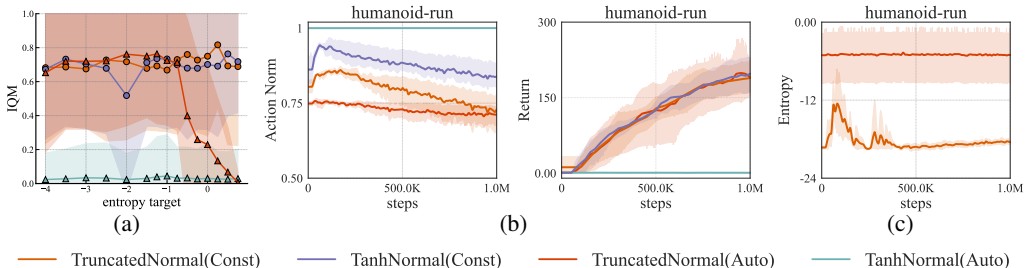

Figure 10: **Analysis of Policy Distributions.** Comparison of Truncated and Tanh Gaussian policies with varying $\delta$ on DMC tasks. Target entropy represents the desired average entropy per action dimension. (a) The Truncated Gaussian exhibits greater stability across four DMC tasks. (b) For the Tanh Gaussian with a learned $\delta$, instability arises as action norms approach the boundary, causing training to collapse. (c) The Truncated Normal distribution's entropy remains stable and well-controlled in both modes, shown here for a target entropy of -0.75.

We study the choice of policy distribution and the handling of its standard deviation, $\delta$. We compare a Truncated Gaussian against a Tanh-squashed Gaussian, each with a constant $\delta$ (set to 0 in our experiments) and a learned $\delta$, using SAC on four hardest tasks from the DMC Dog & Humanoid suites(*dog-run, dog-trot, humanoid-run, humanoid-walk*) with 5 seeds and 1M environmental steps. As shown in Figure 10, the Truncated Gaussian is significantly more stable. The Tanh Gaussian experiences catastrophic training failures when $\delta$ is learned. Our analysis reveals that with the Tanh Gaussian, the action norm often approaches the distribution's boundaries. This causes the learned $\delta$ to grow explosively, creating a vicious cycle of instability as the policy attempts to output actions near the boundary while satisfying the entropy objective. This issue is absent in the Truncated Gaussian, which yields stable $\delta$ values. Given that the performance difference between a learned and a constant $\delta$ is minimal under the Truncated Gaussian, we adopt the truncated gaussian distribution with constant $\delta$ of 0 setting for its simplicity in main results.

### C.1.2 BATCH-LEVEL ENTROPY REGULARIZATION V.S. STATE-LEVEL ENTROPY REGULARIZATION

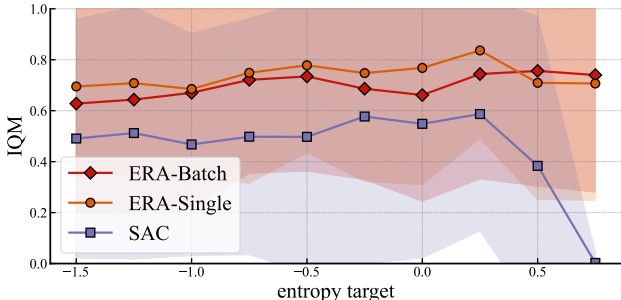

Figure 11: **Comparison between state-level and batch-level entropy regularization methods on DMC Dog & Humanoid suites.** Both methods outperform the SAC baseline.

In addition to the state-level entropy regularization method presented in the main paper, we also investigate a batch-level entropy regularization method, which directly constrains the expected entropy of the action distribution over $\rho_\pi$. Specifically, we modify the activation form of ERA in Eq. 12 to the form in Eq. 48.

$$\mu' = \mu, \quad \sigma' = \exp\left[\max\left(\log\sigma_{\max} + \left(\frac{\mathcal{H}'_0}{D} - \log\sqrt{2\pi e} - \log\sigma_{\max}\right)\frac{e^{\hat{\sigma}_i}}{\bar{e}^{\hat{\sigma}}}, \log\sigma_{\min}\right)\right] \quad (48)$$

Where $\bar{e}^{\hat{\sigma}} = \frac{1}{N}\sum_{i=1}^{N} e^{\hat{\sigma}_i}$ is the average of $e^{\hat{\sigma}}$ over the batch. During training, we can calculate $\bar{e}^{\hat{\sigma}}$ over the sampled batch, and during evaluation, we can use a running average of $\bar{e}^{\hat{\sigma}}$ over the training

process, which is similar to the running statistics in BatchNorm (Ioffe & Szegedy, 2015). We conduct an ablation study to compare the performance of state-level and batch-level entropy regularization methods on DMC Dog & Humanoid suites(*dog-run, dog-trot, humanoid-run, humanoid-walk*). As shown in Figure 11, both methods achieve similar performance, outperforming the SAC baseline. This indicates that in locomotion-dominated control tasks, which require high exploration due to the need for randomness but do not demand high precision, the difference between state-level and batch-level entropy regularization is minimal.

### C.1.3 SAC-ERA ON MUJOCO GYM ENVIRONMENTS

We also evaluate the performance of SAC-ERA on the classic Mujoco Gym environments, including *HalfCheetah-v4, Hopper-v4, Walker2d-v4, Ant-v4, Humanoid-v4, Swimmer-v4*, and compare it with the SAC baseline. Figure 12 shows the learning curves of SAC-ERA and SAC on these environments. Despite their massive performance gap on HumanoidBench, SAC-ERA demonstrates only slight advantages over SAC on Mujoco Gym environments. This may be due to the relatively low action space dimensionality in Mujoco environments, which reduces the impact of different constraint schemes. This finding suggests that modern algorithm design should shift focus from considering Mujoco to higher-dimensional action spaces, which can better evaluate algorithm performance in complex environments.

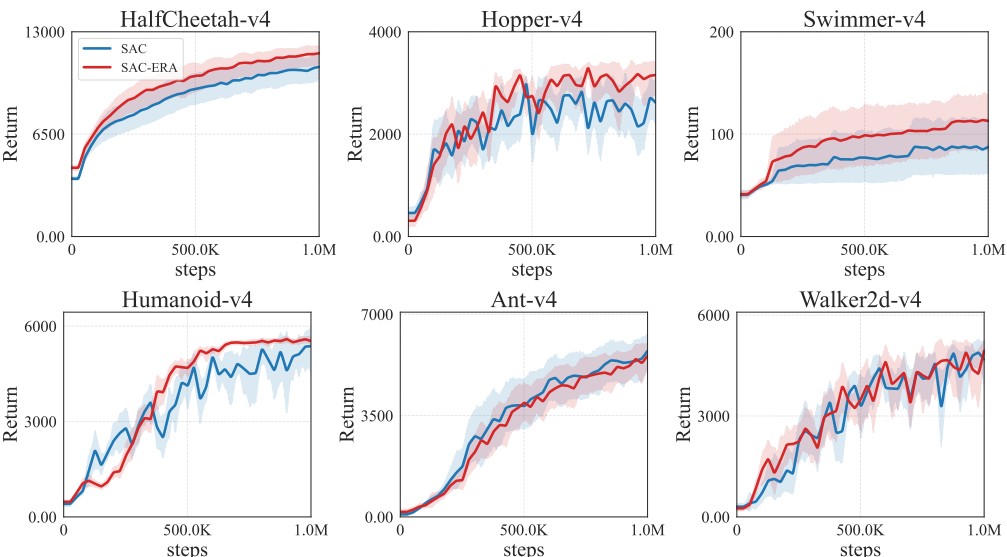

Figure 12: **Learning curves of SAC-ERA and SAC on Mujoco Gym environments.** SAC-ERA demonstrates very slight advantages over SAC.

### C.1.4 APPLICABILITY OF LLM RL TECHNIQUES TO CONTINUOUS CONTROL

We investigated the applicability of two recent techniques from Reinforcement Learning for Large Language Models (LLM RL), designed to prevent entropy collapse, to the domain of continuous control. Specifically, we trained a PPO agent on the HalfCheetah-v4 benchmark for 10 random seeds, incorporating two distinct methods: Selective High-Entropy Training, which trains the agent only on a certain proportion of high-entropy samples, and Clip-Higher, which applies a larger clip ratio for advantages greater than one. Recognizing the significant disparities between LLM RL and continuous control tasks, we evaluated a range of parameters for each technique to ensure that any ineffectiveness was not due to improper parameter selection.

The results, presented in Figure 13, show that these techniques struggle to provide higher policy entropy compared to the standard PPO algorithm in the control task. Furthermore, they yield no significant or only marginal performance improvements; we suspect such minor gains may not even stem from better entropy regularization. Consequently, the performance of these methods is not comparable to our proposed approach, ERA. These findings lead to two main conclusions. First, they highlight the substantial differences between LLM RL and continuous control, demonstrating

that techniques effective in one domain do not necessarily transfer to the other, even when using the same algorithmic framework. Second, they underscore the superior performance of our proposed ERA method.

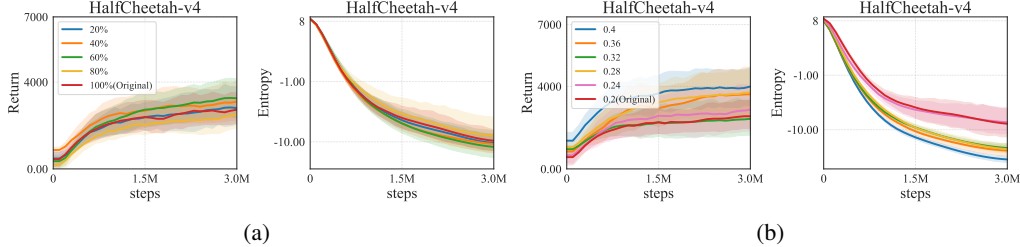

Figure 13: **Results of Selective High-Entropy Training and a Clip-Higher Strategy in Continuous Control.** (a) Performance when training the agent exclusively on a top percentage of high-entropy samples. (b) Performance of the clip-higher strategy with varying clipping ratios.

### C.1.5 COMPARING ERA WITH OTHER MAXIMUM ENTROPY RL APPROACHES

A key baseline for our entropy regularization approach (ERA) is the projection-based method from Akrour et al. (2019), which we term **Scale Std**. This method scales the standard deviation of a Gaussian policy by a factor $> 1$ if its entropy falls below a threshold, conceptually similar to rejection sampling. While this appears similar to our mapping concept, the mechanism is fundamentally different. Scale Std merely *translates* the policy's output log standard deviations by a uniform factor. This does not incentivize the policy to learn an optimal *allocation* of entropy across different action dimensions, as the constraint is borne uniformly.

This difference is evident in the gradient signal. For the Scale Std method, the mapped standard deviation $\vec{\sigma'}$ is calculated as:

$$\vec{\sigma'} = \vec{\sigma} \cdot \exp\left(\mathcal{H}_{\text{target}} - \mathcal{H}(\vec{\sigma})\right)/d \tag{49}$$

where $d$ is the action dimension, $\mathcal{H}_{\text{target}}$ is the target entropy, and $\mathcal{H}(\vec{\sigma}) = \frac{d}{2}\ln(2\pi e) + \sum_{i=1}^{d} \ln \sigma_i$ is the current policy entropy. The action is then sampled as $a \sim \mathcal{N}(\mu, \text{diag}(\vec{\sigma'}^2))$. The resulting gradient with respect to the policy's original log standard deviation $\ln \sigma_i$ (which the network outputs) is:

$$\frac{\partial a}{\partial \ln \sigma_i} = \epsilon_i \cdot \underbrace{\exp\left(\mathcal{H}_{\text{target}} - \mathcal{H}(\vec{\sigma})\right)/d}_{\text{Uniform Scalar } C} \cdot \left(\sigma_i - \frac{2}{d}\right) \tag{50}$$

For comparison, the gradient for SAC without an entropy penalty is simply:

$$\frac{\partial a}{\partial \ln \sigma_i} = \epsilon_i \cdot \sigma_i \tag{51}$$

Thus, the Scale Std gradient is merely the standard SAC gradient scaled by a uniform constant $C$ and offset by another uniform constant $(C \cdot 2/d)$. This adjustment provides no differential signal to incentivize entropy *allocation* between dimensions. **This post-processing of the policy output does not truly make the policy learn to allocate entropy among dimensions.**

In contrast, while the gradient with respect to ERA's *final* log std is also $\epsilon_i \cdot \sigma_i$, this gradient is backpropagated through the ERA activation to the policy's original output $\hat{\sigma}$(which is not actually a standard deviation). This process multiplies the gradient by the derivative of the ERA function, which is *dimension-specific* due to the softmax mechanism. This provides the necessary differential signal, compelling the policy to learn an optimal entropy allocation, which is the fundamental reason for its success.

We validated this theoretical analysis by comparing SAC-ERA and SAC (Scale Std) on the DMC *dog-trot* task. We used a target entropy of $-A$ (where $A = 38$ is the action dimension) and a compensated truncated distribution for both to ensure fair comparison. To visualize the learned exploration strategy, we generated density heatmaps of the policy's log standard deviations over training (10 seeds), shown in Figure 14. For Scale Std, we plot the pre-translation log stds (as a

uniform translation only alters the distribution's location, not its shape), and for ERA, we plot the final, post-mapping log stds.

The results are stark. SAC (Scale Std) exhibits a highly uneven distribution: most dimensions collapse to the lower bound -8, while a few saturate at the upper bound 0 (using the default range [-8, 0]). The mean log std was around -7, indicating that most dimensions cease exploration, while a few to explore excessively. Conversely, SAC-ERA shows a clear diffusion from a uniform start, as the policy learns to allocate entropy across dimensions in a targeted manner. The final distribution is well-spread, not clustered at the bounds, indicating all dimensions participate meaningfully in exploration.

This strategic difference directly impacts performance, as shown in Figure 15. We tested on four complex tasks: DMC *dog-trot*, *humanoid-walk*, and HumanoidBench *h1-walk* and *h1-run*. SAC (Scale Std) shows a mild improvement on *dog-trot* and is significantly *worse* than the baseline SAC on the other three, suggesting its naive exploration strategy hinders learning. In contrast, **SAC-ERA significantly outperforms both SAC and SAC (Scale Std) in all environments**, confirming that ERA effectively guides the policy to rationally allocate entropy across dimensions, a failure point for the Scale Std method.

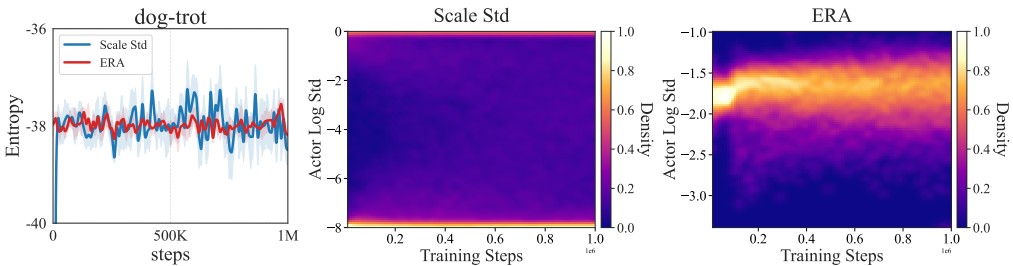

Figure 14: Entropy curves (left), evolution of log standard deviation distributions for SAC (Scale Std) (middle) and SAC-ERA (right) on the *dog-trot* task. Scale Std leads to a polarized, uneven distribution, while ERA learns a balanced, diffusive allocation.

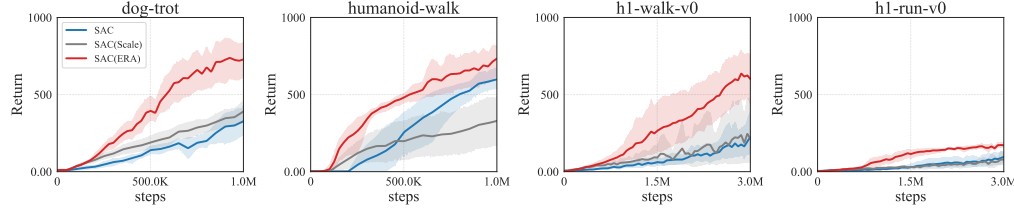

Figure 15: Performance comparison of SAC-ERA against SAC (Scale Std) and baseline SAC on complex locomotion tasks (DMC *dog-trot*, *humanoid-walk*, and HumanoidBench *h1-walk*, *h1-run*).

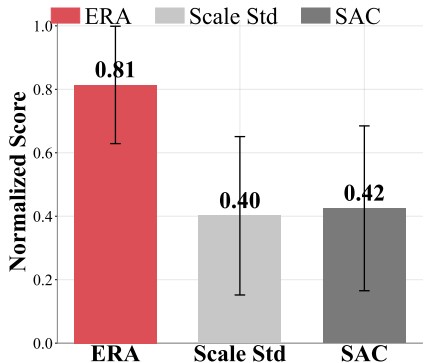

Figure 16: Normalized scores across 4 tasks. SAC-ERA significantly outperforms both SAC and SAC (Scale Std) in all environments

In addition to the projection-based method, several other approaches have been explored to implement maximum entropy reinforcement learning, including recent diffusion-based and flow-based methods (Celik et al., 2025; Chao et al., 2024; Ma et al., 2025). However, these methods often require significantly more computational resources; for instance, the MEow algorithm (Chao et al., 2024) requires at least 2.3 times the training time of SAC. We therefore focus our comparison on two recent methods that also adopt Gaussian policies:

- **EAPO** (Choe & Kim, 2024): The core innovation of Entropy Advantage Policy Optimisation (EAPO) is decomposing the max-entropy objective into cumulative reward and trajectory entropy, then independently estimating advantage functions for each. It introduces a dedicated "entropy critic" to separately learn the value of future uncertainty, combining it with the traditional value of future rewards.

- **MNSE** (Zhong et al., 2024): The Maximum Next-State Entropy (MNSE) paper argues for the direct maximization of next-state entropy, positing that this more directly measures the diversity of states induced by the policy and can lead to more efficient exploration.

Since no public code repositories were available, we compare against the curves reported in the original papers. The experimental setups are as follows:

- EAPO utilizes the PPO algorithm as its base and was trained for 4 million timesteps (more than the 3 million timesteps used in PPO-ERA).

- MNSE is built upon the SAC algorithm and was trained for 1 million timesteps (the same as SAC-ERA).

We compare PPO-ERA with EAPO, and SAC-ERA with MNSE on the Mujoco Gym benchmark. The results are presented in Figure 17 and Figure 18. As shown, ERA demonstrates superior performance over EAPO when both are built on PPO, and it also outperforms MNSE when SAC is used as the base algorithm. Although Mujoco Gym is a relatively low-difficulty benchmark, we are limited to it as neither of the other papers presented results in more complex environments like DMC Suite or HumanoidBench. These findings suggest that ERA is a more effective implementation of maximum entropy reinforcement learning.

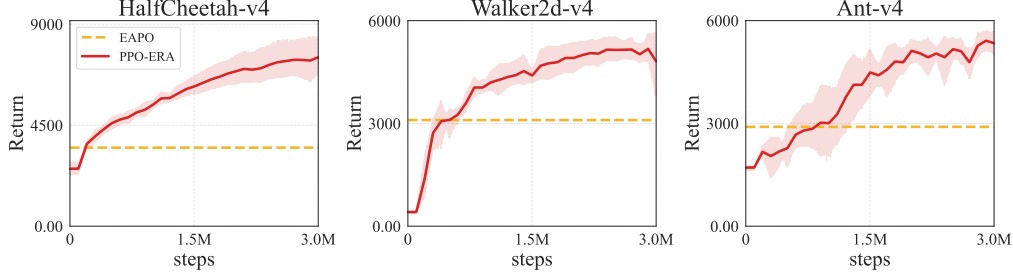

Figure 17: Performance comparison of PPO-ERA against EAPO on MuJoCo benchmark tasks.

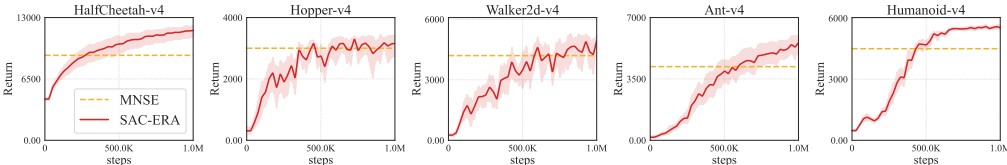

Figure 18: Performance comparison of SAC-ERA against MNSE on MuJoCo benchmark tasks.

Furthermore, both EAPO and MNSE require additional network architectures and computational resources. EAPO necessitates an extra entropy critic network, while MNSE requires an additional inverse dynamics model network. In contrast, ERA does not require any additional networks, leading to a negligible increase in computational overhead. This makes ERA a more advantageous choice for practical applications.

### C.1.6 SENSITIVITY ANALYSIS ON THE $\sigma$ INTERVAL

The hyperparameters $\sigma_{\max}$ and $\sigma_{\min}$ are frequently employed in algorithms such as SAC. Standard settings for $\sigma_{\min}$ typically include -20, -10, and -8, whereas $\sigma_{\max}$ is commonly set to 0 or 2. We evaluated the performance of SAC-ERA on the *dog-run* and *humanoid-walk* environments using three distinct sets of these values, as illustrated in Fig. 19.

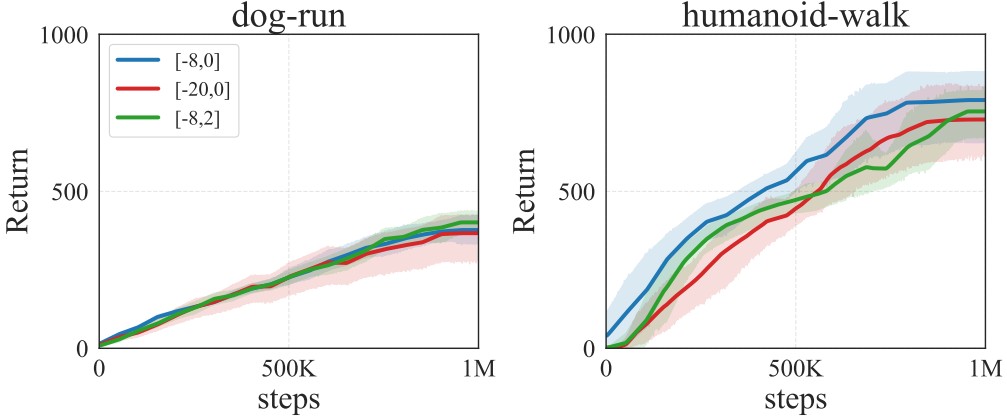

Figure 19: Ablation study on the $\sigma$ interval $[\sigma_{\min}, \sigma_{\max}]$ for SAC-ERA in dog-run and humanoid-walk, with 5 seeds. We compare performance across three different interval settings derived from $\sigma_{\min} \in \{-20, -8\}$ and $\sigma_{\max} \in \{0, 2\}$. The results show that the choice of these bounds has no significant impact on performance, highlighting the robustness of our method.

The experimental results demonstrate that all three settings exhibit nearly identical learning curves on the dog-run environment. On the humanoid-walk environment, the performance differences are also not significant, although the $[-8, 0]$ setting yields slightly better performance compared to the other two configurations. Overall, our method exhibits strong robustness to the choice of the interval. In practice, we recommend prioritizing the $[-8, 0]$ interval, which we use as the default in all our experiments, and considering other settings only when further fine-tuning is required.

### C.1.7 ON THE CHOICE OF THE ENTROPY TARGET

The selection of the entropy target is a key hyperparameter when employing ERA. As discussed in the main paper and prior appendices, the optimal value for SAC-ERA depends on the use of a compensation factor $\hat{\delta}$. For the truncated normal policy, we recommend a higher target (e.g., $0.25\mathcal{A}$) if the compensation factor is set to zero. If the compensation factor is used, we recommend a target of $-\mathcal{A}$, which aligns with the empirical values used in standard SAC implementations (e.g., in stablebaselines and other prior work).

For PPO-ERA, we conducted an ablation study on the entropy target value in the HalfCheetah-v4 and Ant-v4 MuJoCo environments. The results are presented in Figure 20. Overall, these results indicate that PPO-ERA is not highly sensitive to the choice of the entropy target in these environments. It outperforms the PPO baseline by a significant margin across a broad range of target values, with optimal performance observed around a target of $-0.25\mathcal{A}$.

Our experiments also involved TD-MPC2. Due to the extensive training time required for this algorithm, we only tested and reported the results for a target of $-\mathcal{A}$. This value was selected based on the empirical standard commonly adopted in SAC implementations.

### C.1.8 COMPARISON WITH SMALL INITIAL TEMPERATURE SAC

Recent studies (Lee et al., 2025) based on SAC have adopted a smaller initial temperature (e.g., 0.006) to mitigate the impact of fluctuations in the entropy constraint term during training. We compared the performance of SAC initialized with a small temperature (0.006) against the baseline SAC used in this work (initialized at 1.0) on 4 tasks, including the DMC *dog-run*, *humanoid-walk* and HumanoidBench *h1-walk* and *h1-run*. The results are presented in Fig. 21.

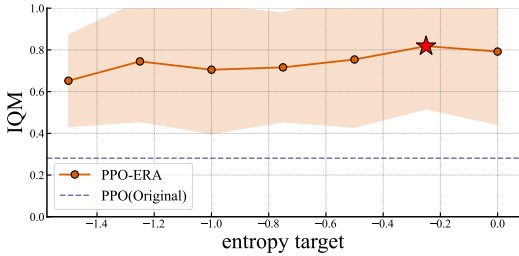

Figure 20: Ablation study on the entropy target for PPO-ERA in HalfCheetah-v4 and Ant-v4 environments.

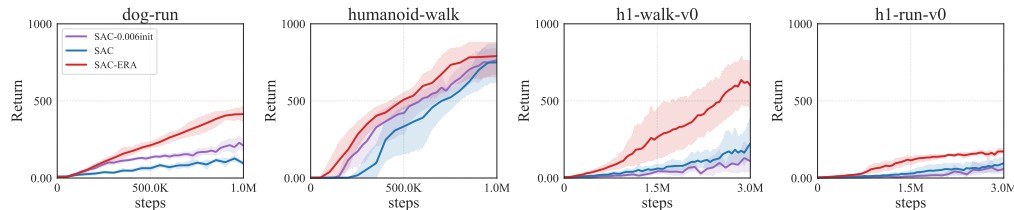

Figure 21: Performance comparison between SAC with a small initial temperature (0.006), the baseline SAC (initial temperature 1.0), and SAC-ERA on 4 tasks. SAC-ERA outperforms both baselines, demonstrating its superiority in complex control environments.

The results indicate that SAC with a small initial temperature outperforms the baseline SAC (initial temperature 1.0) in two of the four tested environments, while performing comparably or slightly worse in the other two. This suggests that using a small initial temperature may mitigate the impact of entropy constraint fluctuations in certain scenarios, but it is not effective in all environments, and its efficacy likely depends on specific environmental characteristics. Fundamentally, this approach does not resolve the underlying issue: while a small initial temperature can partially mitigate the fluctuations caused by the entropy constraint, the continued presence of the entropy term in the loss function may still hinder the optimization of cumulative returns, particularly when environmental rewards are sparse. Moreover, SAC-ERA significantly outperforms SAC with a small initial temperature across all four environments, further demonstrating the superiority of ERA in complex control environments.

Furthermore, many existing SAC implementations widely adopted by the community, such as stablebaselines3 (Raffin et al., 2021) and jaxrl (Kostrikov, 2021), still default to an initial temperature of 1.0. We argue that our use of this more common 1.0 initial temperature as a baseline is reasonable, given that the optimal initial temperature possibly requires environment-specific tuning. In contrast, employing ERA completely obviates this issue.

### C.1.9    VALIDATION AGAINST STABLE-BASELINES3 (SB3) IMPLEMENTATIONS

To validate the reliability and generalizability of our experimental findings, we benchmarked the performance of our SAC and PPO baseline implementations against the standard implementations provided by the Stable-Baselines3 (SB3) library.

**SAC Comparison.**    For the Soft Actor-Critic (SAC) agent, we precisely aligned the network architecture and hyperparameter configurations with the SB3 implementation. We then conducted comparative experiments on four tasks from the DeepMind Control (DMC) Suite (specifically, the Dog and Humanoid domains). The results indicate that the SB3 SAC implementation performs slightly better than our JAX RL-based implementation on the Dog tasks, but conversely, underperforms on the Humanoid tasks. Ultimately, the final normalized scores for our baseline and the SB3 baseline were nearly identical. However, the SB3 implementation demonstrated greater stability (i.e., lower variance across seeds). Overall, the SB3 baseline still exhibits suboptimal performance on these complex control tasks, showing a significant performance gap compared to our proposed SAC(ERA) agent.

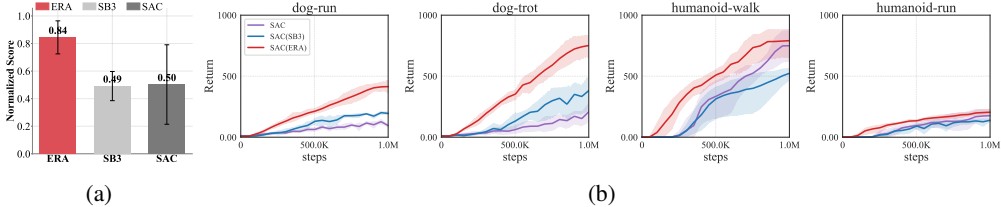

(a)                                                                                          (b)

Figure 22: **Performance comparison of StableBaselines3 (SB3), JAX RL SAC(our baseline), and SAC(ERA) on DMC Dog and Humanoid tasks, averaged over 5 seeds.** (a) Normalized scores across four tasks. SB3 SAC demonstrates greater stability (lower variance) compared to JAX RL SAC, although their average scores are comparable. (b) Learning curves for the four tasks. SB3 SAC excels on the Dog tasks, while JAX RL SAC performs better on the Humanoid tasks. SAC(ERA) consistently outperforms both baselines across all environments, while also exhibiting comparable or superior stability.

**PPO Comparison.** Similarly, for the Proximal Policy Optimization (PPO) agent, we utilized hyper-parameter settings identical to those in our primary experimental setup. The evaluation reveals that the SB3 PPO implementation achieved slightly inferior results compared to our PPO implementation on both the HalfCheetah and Ant environments. Consistent with the SAC results, the SB3 PPO agent again exhibited superior stability.

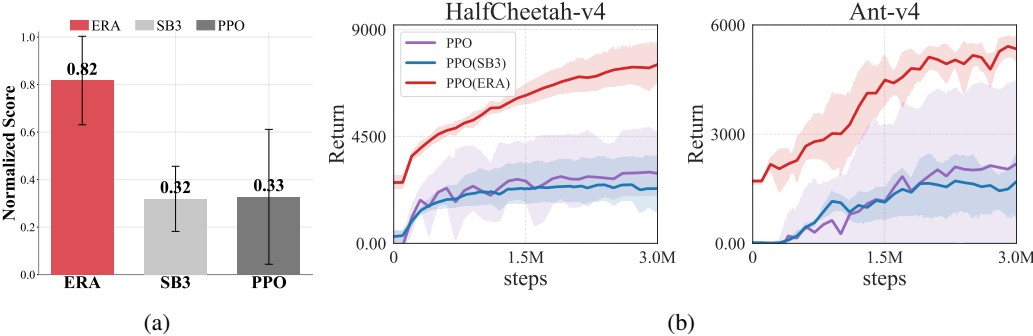

(a)                                                                                          (b)

Figure 23: **Performance comparison of StableBaselines3 (SB3), PPO(our baseline), and PPO(ERA) on HalfCheetah and Ant, averaged over 5 seeds.** (a) Normalized scores across the two tasks. SB3 PPO shows enhanced stability (lower variance) relative to our PPO baseline, though their average scores are similar. (b) Learning curves for the two tasks. Our PPO baseline and SB3 PPO achieve similar performance on both environments. PPO(ERA) consistently surpasses both baselines across all tested environments, while also demonstrating comparable or superior stability.

**Conclusion.** In summary, the baselines used in our study and their SB3 counterparts demonstrate highly comparable performance. This suggests that substituting our baselines with the SB3 implementations would not substantively alter the main conclusions of this work. While the SB3 baselines exhibited greater stability, this difference is not significant enough to affect our conclusions, which are based on aggregates over at least 5 random seeds. Furthermore, it is noteworthy that our ERA-enhanced agent significantly outperforms the SB3 baselines across all tested environments, while also demonstrating comparable or superior stability. This underscores the effectiveness of the ERA method in robustly boosting both agent performance and stability.

### C.1.10 COMPARISON OF ENTROPY DYNAMICS WITH SAC VARIANTS

We conducted a comparative analysis of three methods on the *dog-trot* task: standard SAC (using the Tanh-Gaussian policy with a std range of [-10, 2]), SAC with a truncated normal distribution (SAC-TN), and SAC-ERA (using a truncated normal distribution with an auto-tuning compensation term). For all methods, the target entropy was set to $-\mathcal{A}$. Following the same visualization protocol used in our Scale Std analysis, we plotted both the entropy curves and the log std density heatmaps for all three approaches, as shown in Figure 24.

The results indicate that, given the same entropy target, SAC-ERA maintains the most stable entropy curve. SAC-TN exhibits slightly smaller oscillations than the standard SAC. The log std density heatmaps reveal further distinctions. Both standard SAC and SAC-TN undergo a rapid, abrupt shift in the log std distribution during the early stages of training; this corresponds to the dynamic adjustment of the entropy temperature parameter as it converges to the target. Concurrently, their log std distributions diffuse both faster and more broadly compared to SAC-ERA. In contrast, the log std distribution for SAC-ERA is markedly more stable, exhibiting a gradual and controlled diffusion process over time. This highlights a significant divergence in training dynamics, distinguishing SAC's extrinsic adjustment via an entropy term from ERA's intrinsic regulation via its activation function.

In terms of final performance, SAC-ERA also outperforms both SAC and SAC-TN. The performance of SAC-TN is approximately on par with the standard SAC. This finding suggests that merely replacing the Tanh-Gaussian policy with a truncated normal distribution does not, by itself, yield significant performance gains. Instead, the critical factor appears to be the ERA entropy constraint mechanism, which provides a more stable entropy regulation process and, consequently, more stable training dynamics.

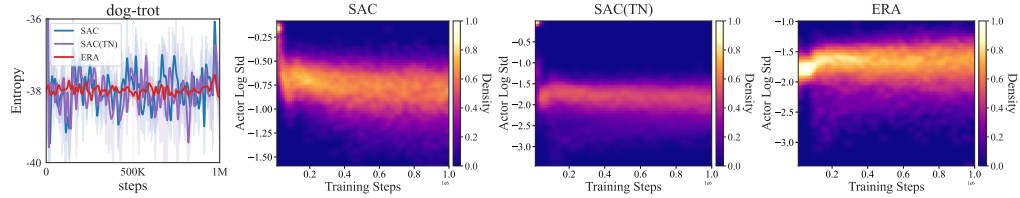

Figure 24: Comparison of entropy curves (left) and log standard deviation heatmaps (middle, right) for standard SAC, SAC-TN, and SAC-ERA on the *dog-trot* task.

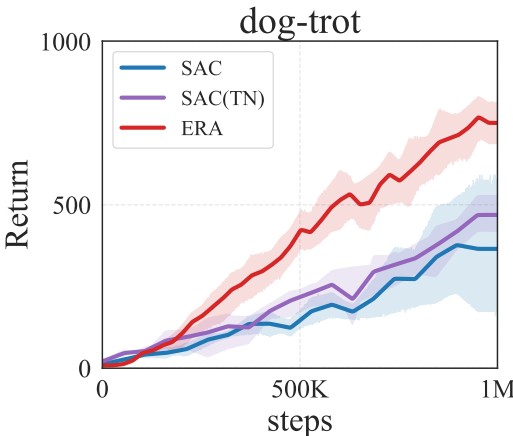

Figure 25: Performance comparison of SAC, SAC-TN, and SAC-ERA on the *dog-trot* task.

### C.1.11 A DEMONSTRATIVE EXPERIMENT ON GRADIENT CONFLICTS IN SAC

We do a simple experiment to demonstrate the gradient conflict between reward maximization and entropy maximization in SAC on DMC humanoid-run task. We compute the gradients of the reward objective and the entropy objective on distribution parameters $\mu, \sigma$ and the final action $a$. We then compute the cosine similarity between the two gradients to measure their alignment. A cosine similarity greater than 0 indicates that the two gradients are aligned(i.e. their angle is less than 90 degrees), while a cosine similarity less than 0 indicates that the two gradients are conflicting(i.e. their angle is greater than 90 degrees). We plot the cosine similarity over training steps in 26. Our results show that for all three parameters, among all 5 seeds tested, the cosine similarity is all negative for the majority of training time, indicating that the reward and entropy objectives, for the most part, has conflicting gradients. This supports our claim that in SAC, the reward maximization and entropy maximization objectives are often at odds, leading to inefficient policy optimization paths. In contrast, with ERA, the mean and standard deviation only receive gradients from the reward objective, while

the entropy constraint is handled internally by the policy itself, allowing for more direct and efficient optimization towards the reward goal.

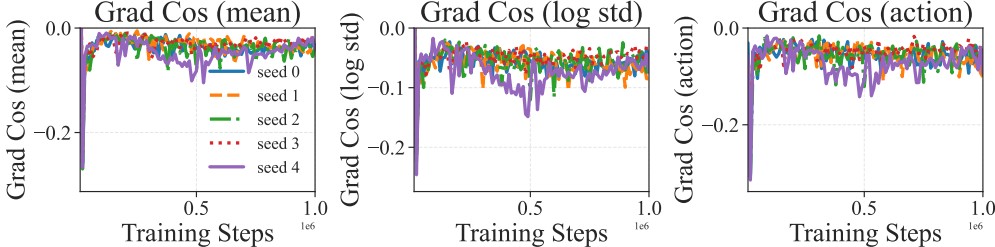

Figure 26: Cosine similarity between reward and entropy gradients on $\mu, \sigma, a$ over training steps in SAC on DMC humanoid-run task. Negative values indicate conflicting gradients.

### C.1.12   TIME COST OF ERA IN CONTINUOUS CONTROL

A potential concern might be the additional time overhead introduced by using ERA. To evaluate this, we recorded the training times of FastTD3 and FastSAC-ERA on HumanoidBench, as shown in Figure 27. It can be observed that using ERA does introduce some time overhead due to the more complex activation function applied to the output. However, this overhead accounts for only about 6% of the total training time on average. Considering the improved exploration performance and higher sample efficiency brought by ERA, we believe this is a worthwhile trade-off.

The scenario for comparing training speed against FastTD3 is particularly stringent. This is because FastSAC-ERA must additionally output per-dimension policy standard deviations, which introduces computational overhead not present in FastTD3. To quantify the specific overhead of our method, we measured the training time of baseline SAC versus SAC-ERA in the *dog-trot* environment. When trained on a single A10 GPU, the additional time cost of SAC-ERA, averaged over five seeds, was approximately 3%.

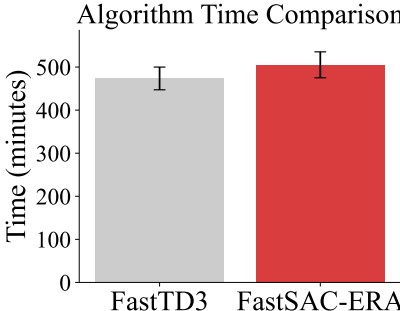

Figure 27: **Time comparison on *h1hand-hurdle-v0*.** We compare the training time of FastTD3 and FastSAC-ERA on HumanoidBench. The results show that using ERAintroduces a modest time overhead, averaging around 6% of the total training time, which is a reasonable trade-off for the improved exploration performance and sample efficiency it provides.

### C.2   ADDITIONAL RESULTS ON IMAGE CLASSIFICATION

### C.2.1   COMPARING ERA WITH COMMON REGULARIZATION TECHNIQUES

A plethora of regularization methods have been proposed and utilized in the field of image classification. To further investigate the comparative effectiveness of ERA against commonly used regularization methods like dropout and label smoothing in the vision domain, we conducted a series of straightforward comparative experiments on the CIFAR-10 dataset. In our main experiment, we adopted the default settings from the `timm` library, which include a label smoothing factor of 0.1 and no dropout. For the sake of comparison, we respectively adjusted the label smoothing factor to 0.2 and 0.3, and the dropout rate to 0.1, 0.2, and 0.3. The results were then compared against the baseline algorithm from our main experiment and ERA.

The experimental results are presented in Figure 28. The findings indicate that increasing the intensity of label smoothing adversely affects model performance, while the improvement from employing dropout is marginal (the top-1 accuracy may decrease, whereas the top-5 accuracy shows an improvement). In contrast, ERA effectively and consistently enhances model performance, with a margin of improvement significantly superior to that of both dropout and label smoothing. This outcome further validates the advantage of ERA over conventional regularization methods. While constraining the model's entropy, ERA permits the model to freely allocate uncertainty among dimensions, thereby better adapting to the intrinsic structure of the data. This enables ERA to more effectively boost the model's generalization capability.

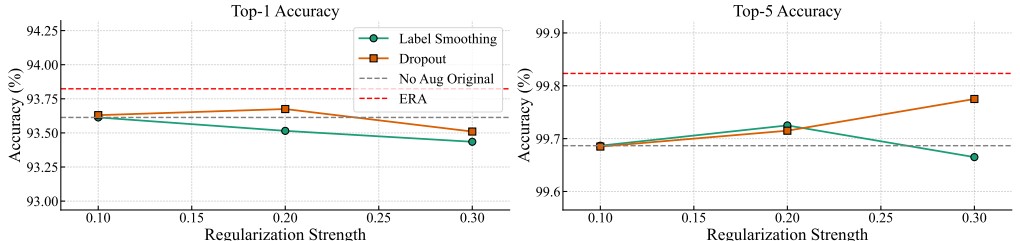

Figure 28: **Comparison of different regularization methods on the CIFAR-10 dataset.** The left subplot shows the Top-1 accuracy, and the right subplot shows the Top-5 accuracy. Our method, ERA, is compared against varying intensities of Label Smoothing and Dropout.

Furthermore, we extended our comparison to two other entropy constraint methods, which are common in reinforcement learning but rare in image classification. The first is the **Entropy Term**, which adds an entropy penalty directly to the loss. The second is the projection-based method from Akrour et al. (2019), which we term **Linear Interpolation**. Similar to its continuous-space counterpart, this method acts when the policy's output entropy falls below a target: it increases the entropy by interpolating the distribution with a uniform distribution.

Analogous to the continuous case, we provide a theoretical analysis of the gradient back-propagation mechanism under the cross-entropy loss to elucidate the fundamental difference between ERA and projection-based methods. Consider the policy output distribution $\pi(a|s)$ with corresponding logits $\vec{l}$. For a target class $k$, the cross-entropy loss is $\mathcal{L} = -\log \pi(a_k|s)$.

In the **Linear Interpolation** method, the adjusted probability is a mixture of the original policy $\pi_{\text{orig}}$ and a uniform distribution, governed by the entropy constraint:

$$\pi(a_k|s) = \underbrace{\frac{\log N - \mathcal{H}_0}{\log N - \mathcal{H}(\pi_{\text{orig}})}}_{\lambda} \pi_{\text{orig}}(a_k|s) + \underbrace{\frac{\mathcal{H}_0 - \mathcal{H}(\pi_{\text{orig}})}{\log N - \mathcal{H}(\pi_{\text{orig}})}}_{1-\lambda} \frac{1}{N} \tag{52}$$

Applying the chain rule, we derive the gradient of the loss with respect to the original probability $\pi_{\text{orig}}(a_k|s)$. Note that the mixing coefficient $\lambda$ depends on the global entropy $\mathcal{H}(\pi_{\text{orig}})$, which in turn depends on $\pi_{\text{orig}}$:

$$\frac{\partial \mathcal{L}}{\partial \pi(a_k|s)} = -\frac{1}{\pi(a_k|s)} \tag{53}$$

$$\frac{\partial \pi(a_k|s)}{\partial \pi_{\text{orig}}(a_k|s)} = \lambda + \pi_{\text{orig}}(a_k|s)\frac{\partial \lambda}{\partial \pi_{\text{orig}}(a_k|s)} + \frac{1}{N}\frac{\partial(1-\lambda)}{\partial \pi_{\text{orig}}(a_k|s)} \tag{54}$$

Substituting the partial derivatives of the entropy term $\frac{\partial \mathcal{H}(\pi_{\text{orig}})}{\partial \pi_{\text{orig}}(a_k|s)} = -\log \pi_{\text{orig}}(a_k|s) - 1$, we obtain the complex sensitivity term:

$$\frac{\partial \pi(a_k|s)}{\partial \pi_{\text{orig}}(a_k|s)} = \lambda + \left(\pi_{\text{orig}}(a_k|s) - \frac{1}{N}\right)\frac{\log N - \mathcal{H}_0}{(\log N - \mathcal{H}(\pi_{\text{orig}}))^2}(\log \pi_{\text{orig}}(a_k|s) + 1) \tag{55}$$

Critically, although mathematically involved, this sensitivity term depends principally on the target class $k$ and the global entropy state. When propagating to the logits $l_i$, the total gradient becomes:

$$\frac{\partial \mathcal{L}}{\partial l_i} = \underbrace{\left[ \frac{\partial \mathcal{L}}{\partial \pi(a_k|s)} \cdot \frac{\partial \pi(a_k|s)}{\partial \pi_{\text{orig}}(a_k|s)} \right]}_{\Psi(\pi,k)} \cdot \frac{\partial \pi_{\text{orig}}(a_k|s)}{\partial l_i} \tag{56}$$

Here, $\frac{\partial \pi_{\text{orig}}(a_k|s)}{\partial l_i}$ is the standard softmax gradient. The term $\Psi(\pi, k)$ acts effectively as a scalar coefficient common to the gradient flow. This indicates that the projection method primarily acts as a **uniform gradient scaler**: it creates a gradient signal that pushes the distribution towards uniformity globally, without providing dimension-specific guidance beyond what the original softmax offers. This limitation stems from its nature as a post-processing step.

In stark contrast, ERA modifies the logits directly within the architecture *before* the softmax. The gradient flow for ERA is defined as:

$$\frac{\partial \mathcal{L}}{\partial l_i} = \frac{\partial \mathcal{L}}{\partial \pi(a_k|s)} \cdot \sum_j \frac{\partial \pi(a_k|s)}{\partial l'_j} \frac{\partial l'_j}{\partial l_i} \tag{57}$$

where $l'_i = h^{-1}(g(l_i))$. approximating $g(l_i)$ as a shifted softmax $g(l_i) \approx a \frac{e^{l_i}}{\sum_j e^{l_j}} + b$, the gradient can be expressed as:

$$\frac{\partial \mathcal{L}}{\partial l_i} \approx (\delta_{ik} - \pi(a_i|s)) \cdot a \cdot \frac{\partial}{\partial l_i}\left( \frac{e^{l_k}}{\sum_j e^{l_j}} \right) \cdot \frac{\partial \mathbf{l'_i}}{\partial \mathbf{g(l_i)}} \tag{58}$$

The crucial differentiator is the term $\frac{\partial \mathbf{l'_i}}{\partial \mathbf{g(l_i)}}$. Since this derivative depends on the value of $g(l_i)$, which varies across dimensions according to their individual contribution to the entropy, it acts as a **dimension-specific scaling factor**. Unlike the post-processing projection which applies a uniform scalar $\Psi$ to all gradients, ERA generates a structured gradient field that adapts individually to each logit $l_i$, enabling the model to learn an optimal entropy allocation strategy.

We tested both methods on CIFAR-10 using the same experimental setup as ERA (without data augmentation, as in our ablation studies). We tested the Entropy Term with coefficients of 1e-4, 1e-3, and 1e-2. For Linear Interpolation, the target entropy was set to 0.6, identical to that used in ERA.

The results are shown in Figure 29. Both of these entropy constraint methods underperform ERA in both top-1 and top-5 accuracy. This suggests that the utility of these RL-centric entropy methods may be limited in image classification, which could explain their infrequent use in the CV domain.

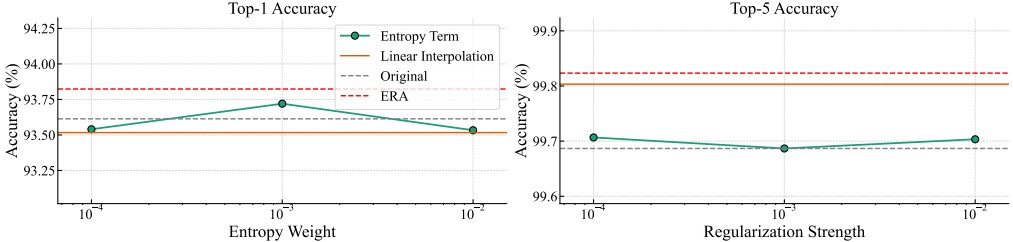

Figure 29: **Comparison of different regularization methods on the CIFAR-10 dataset.** The left subplot shows the Top-1 accuracy, and the right subplot shows the Top-5 accuracy. Our method, ERA, is compared against varying entropy term weights and the Linear Inerpolation method.

Furthermore, we evaluated the efficacy of the SAC-style automatic temperature adjustment mechanism on the CIFAR-10 dataset. **It's worth noting that while ERA and Linear Interpolation regulate the lower bound of entropy in image classification, SAC-style automatic temperature adjustment regulates the expectation of entropy. We have to choose higher target entropy to keep the entropy level aligned with ERA.** We experimented with three target entropy values: 1.2, 1.25, and 1.5. These values were selected based on prior experimental findings, where the final training loss typically converged around 1.21 and increased to approximately 1.23 with the addition of ERA.

Consequently, at a target entropy of 1.2, the entropy constraint term remains largely inactive; at 1.25, the target entropy aligns with the ERA baseline; and at 1.5, the target imposes a higher entropy requirement. We initialized the temperature coefficient at $10^{-6}$ with a learning rate of $10^{-3}$. The results, depicted in Fig. 31, reveal a distinct trade-off between Top-1 and Top-5 accuracy when using the SAC-style adjustment. Specifically, while Top-5 accuracy exhibits a slight improvement as the target entropy increases, Top-1 accuracy declines. Ultimately, this approach fails to achieve the performance levels attained by ERA.

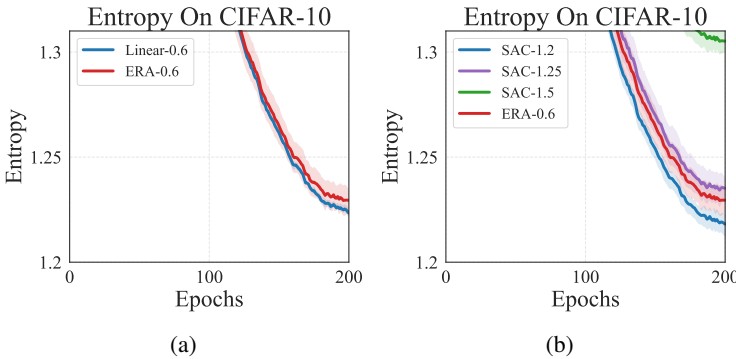

(a)  (b)

Figure 30: **Entropy curves for CIFAR-10 classification.** (a) Entropy curves for ERA and Linear Interpolation with entropy target both set to 0.6. They both regulate the entropy on all samples to be above the target. So entropy curves are quite similar. While in practice they demonstrate different top-1 and top-5 accuracies. (b) Entropy curves for SAC style entropy adjustment and ERA on CIFAR-10 classification. We test three different entropy targets (1.2, 1.25, 1.5) for SAC style adjustment. It's worth noting that SAC style adjustment only regulates the expected entropy to be close to the target, so we must raise the target to achieve similar entropy levels as ERA-0.6. Even when SAC style adjustment achieves similar entropy levels (target=1.25), ERA still outperforms it in terms of top-1 and top-5 accuracies.

In the context of image classification tasks, we further observed that the SAC-style constraint mechanism is largely ineffective. This is primarily because the number of gradient steps is significantly fewer than in control tasks. Moreover, the initial entropy is substantially higher than the loss, causing the entropy term (temperature coefficient) to decrease initially; it only begins to increase gradually once the entropy approaches the threshold. Consequently, it is difficult to effectively satisfy the target entropy constraint within the limited training duration, resulting in final performance that remains close to the baseline.

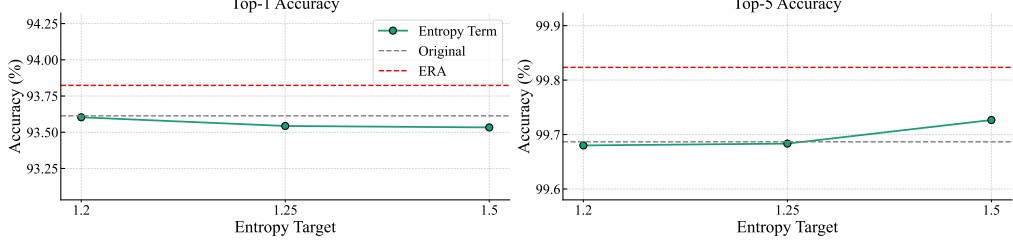

Figure 31: **Comparison of SAC-style automatic temperature adjustment on the CIFAR-10 dataset.** The left subplot shows the Top-1 accuracy, and the right subplot shows the Top-5 accuracy. Our method, ERA, is compared against varying target entropy values using SAC-style temperature adjustment. Here "entropy target" refers to targets of the SAC-style method. A fixed entropy target of 0.6 is used for ERA in this experiment.

### C.2.2 TIME COST OF ERA IN IMAGE CLASSIFICATION

We compared the training time of the ResNet-50 model on the CIFAR-10 dataset, with and without using ERA, under the data augmentation supported by the `timm` library. Consistent with our main results, the experiments were conducted on three machines, each equipped with four NVIDIA A40

GPUs, and we report the average training time. The results are presented in Figure 32. As shown in the figure, since the data is already well-parallelized, there is almost no difference in training time between the algorithm using ERA and the original version.

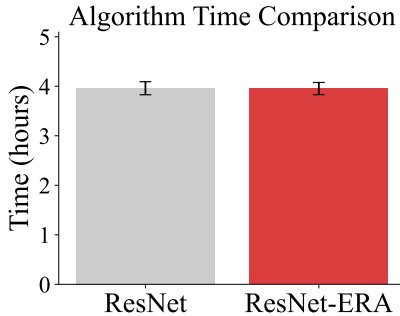

Figure 32: **Time comparison on CIFAR-10.** We compare the training time of ResNet and ResNet-ERAon CIFAR-10. The results show that using ERA introduces almost no time overhead.

### C.3 ADDITIONAL RESULTS ON LLMS

#### C.3.1 ABLATION STUDY ON ADVANTAGE SCALING

In this section, we explore the use of advantage scaling:

$$
A'_t = \begin{cases} \frac{1}{k} A_t & H_{\text{resp}} < \omega_{\text{low}}, A_t > 0, \\ z & (\omega_{\text{low}} \leq H_{\text{resp}} \leq \omega_{\text{high}}, A_t < 0) \text{ or } A_t > 0, \\ k A_t & H_{\text{resp}} > \omega_{\text{high}}, A_t > 0. \end{cases}
$$

For ERA with advantage scaling, we train it for 1400 steps, with hyperparameter $\omega_{\text{low}} = 0.45, \omega_{\text{high}} = 3.0, k = 2$; and for ERA without advantage scaling, we train it in two stages for 1100 steps in total, where the first stage lasts 600 steps with $\omega_{\text{low}} = 0.45, \omega_{\text{high}} = 3.0, k = 2$, and the last 500 steps is trained with $\omega_{\text{low}} = 0.2, \omega_{\text{high}} = +\infty, k = 2$.

As shown in Table 7, both variants—training with or without advantage scaling—achieve substantial improvements over the GRPO baseline. Although adding advantage scaling results in a higher score, the advantage estimates in GRPO are already noisy, so we expect both options to work similarly well and the performance gap to remain relatively small.

Table 7: Ablation study on advantage scaling based on Qwen2.5-Math-7B. For AIME and AMC, the results are avg.@16.

| Model | AIME24 ↑ | AIME25 ↑ | AMC ↑ | MATH500 ↑ | Minerva ↑ | Olympiad ↑ | Avg. ↑ |
|---|---|---|---|---|---|---|---|
| *Base Models* | | | | | | | |
| Qwen2.5-Math Yang et al. (2024a) | 8.6 | 6.3 | 52.2 | 50.8 | 12.1 | 17.2 | 24.5 |
| Qwen2.5-Math-Instruct Yang et al. (2024a) | 13.3 | 10.0 | 57.1 | 81.0 | 32.7 | 38.8 | 38.8 |
| GRPO (Shao et al., 2024) | 34.4 | 12.3 | 69.5 | 80.6 | 36.8 | 40.6 | 45.7 |
| ERA (w/o advantage scaling) | **37.5** | 16.9 | 72.8 | 84.6 | **42.6** | 46.5 | 50.2 |
| ERA (w/ advantage scaling) | 36.0 | **21.0** | **76.6** | **85.4** | 40.1 | **46.8** | **51.0** |

#### C.3.2 DETAILED ENTROPY ANALYSIS

We present the complete entropy curve of our two-stage training without advantage scaling in Figure 33. After decreasing $\omega_{\text{low}}$, the entropy rapidly drops and stabilizes at the second-level entropy lower bound. This confirms that our ERA method successfully enforces a non-trivial entropy floor for the model.

We further analyze the entropy distribution across tokens by plotting the average entropy of the top 20% tokens ($H_{\text{resp}}$) and the bottom 80% tokens in Figure 34. This experiment is carried out with $\omega_{\text{low}} = 0.45, \omega_{\text{high}} = 3.0, k = 2$ without topk. Following Wang et al. (2025), we observe

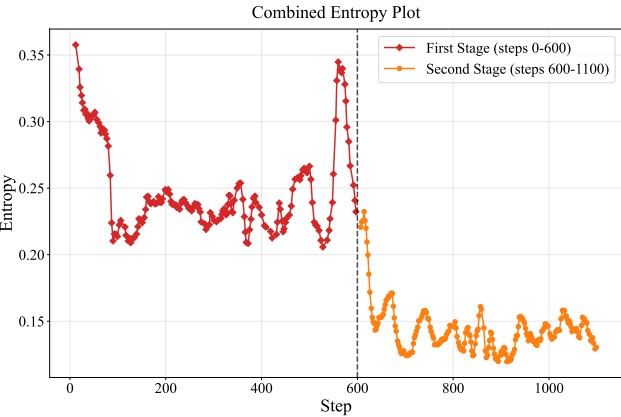

Figure 33: **Entropy curve during two-stage training.** After decreasing $\omega_{\text{low}}$, the entropy rapidly drops and stabilizes at the second-level entropy lower bound, showing that ERA enforces a non-trivial entropy floor.

that the bottom $80\%$ tokens exhibit nearly zero entropy, consistent with our theoretical prediction. Additionally, we plot the proportion of responses with $H_{\text{resp}} < \omega_{\text{low}}$, $H_{\text{resp}} > \omega_{\text{high}}$ in Figure 34. The fraction of responses with $H_{\text{resp}} > \omega_{\text{high}}$ quickly drops to zero, while the fraction with $H_{\text{resp}} < \omega_{\text{low}}$ remains stable at the interval $[0, 0.06]$. This demonstrates that whenever overly low-entropy responses appear, ERA adaptively raises their entropy to a moderate level.

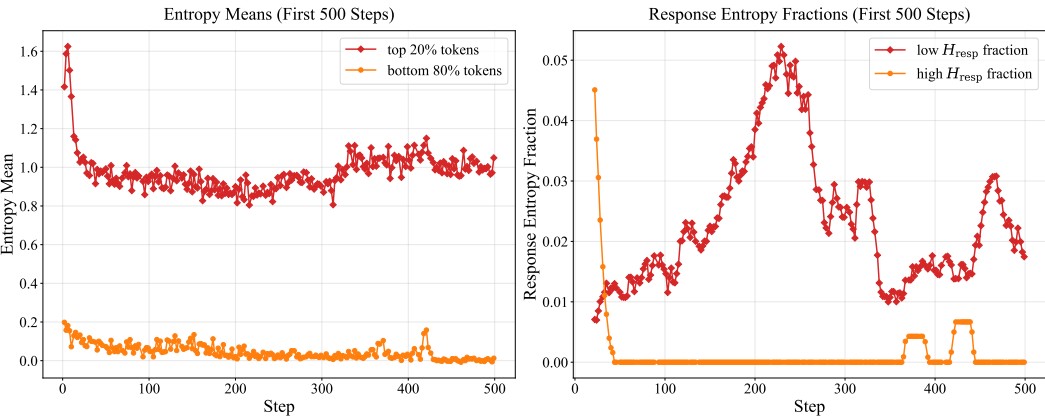

Figure 34: **Detailed entropy analysis.** Left: average entropy of the top $20\%$ tokens ($H_{\text{resp}}$) and the bottom $80\%$ tokens. Right: proportion of responses (running average with window size 20) with $H_{\text{resp}} < \omega_{\text{low}}$ or $H_{\text{resp}} > \omega_{\text{high}}$, demonstrating ERA's ability to prevent both entropy collapse and overly high entropy.

### C.3.3 ABLATION STUDY ON ENTROPY BOUND

Since the purpose of $\omega_{\text{low}}$ is to set a lower bound on entropy, we explore the role of $\omega_{\text{high}}$ in the ERA. As can be seen in Figure 35, without the constraint of $\omega_{\text{high}}$, the model's entropy explodes in a very short time. This indicates that adding an upper bound constraint during training is essential for controlling the entropy of the training process.

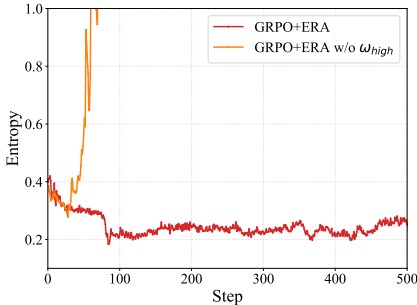

Figure 35: Comparison of ERA with and without $\omega_{\text{high}}$. The entropy of ERA without $\omega_{\text{high}}$ tends to explode within a very short number of steps, leading to the collapse of model training.

### C.3.4 ABLATION STUDY ON THE PROPORTION OF HIGH-ENTROPY TOKENS

In this section, we explore the use of different proportions of tokens to calculate $H_{\text{resp}}$ for rollout samples. We select the top 10% of tokens with the highest entropy from each rollout to represent the entropy $H_{\text{resp}}$ of that sample. For other parameters such as $\omega_{\text{low}}$ and $\omega_{\text{high}}$, we kept them unchanged from the original settings.

As shown in Table 8, modifying the calculation of $H_{\text{resp}}$ still achieves significant improvements compared to GRPO. However, the improvement is smaller compared to ERA. This is because the $H_{\text{resp}}$ calculated from the top 10% tokens is naturally higher than that from the top 20%. As a result, fewer samples meet the condition $H_{\text{resp}} < \omega_{\text{low}}$ compared to the version using 20%. Therefore, the constraining power of entropy is limited, and the results lie between ordinary GRPO and ERA.

Table 8: Ablation study on the proportion of high-entropy tokens based on Qwen2.5-Math-7B. For AIME and AMC, the results are avg.@16.

| Model | AIME24 ↑ | AIME25 ↑ | AMC ↑ | MATH500 ↑ | Minerva ↑ | Olympiad ↑ | Avg. ↑ |
|---|---|---|---|---|---|---|---|
| *Base Models* | | | | | | | |
| Qwen2.5-Math Yang et al. (2024a) | 8.6 | 6.3 | 52.2 | 50.8 | 12.1 | 17.2 | 24.5 |
| Qwen2.5-Math-Instruct Yang et al. (2024a) | 13.3 | 10.0 | 57.1 | 81.0 | 32.7 | 38.8 | 38.8 |
| GRPO (Shao et al., 2024) | 34.4 | 12.3 | 69.5 | 80.6 | 36.8 | 40.6 | 45.7 |
| ERA w/ top 10% tokens | 36.6 | 15.8 | 71.8 | 82.4 | 38.9 | 43.1 | 48.1 |
| ERA | **37.5** | **16.9** | **72.8** | **84.6** | **42.6** | **46.5** | **50.2** |

### C.3.5 TIME COST OF ERA IN LLM

ERA is applied when computing the `log_probs` of tokens in the responses. To evaluate its efficiency, we compare the value of `timing_s/old_log_prob` at the first step in verl's implementation. The experiments were conducted on 32 NVIDIA H20 GPUs, consistent with our main results. The outcomes are shown in Figure 36. As illustrated, since the sampled response is identical in the first step, ERA introduces only about a $5.6\%$ overhead in time cost. When considering an entire training step, the overhead of ERA is even smaller, since its implementation does not affect other components of training (e.g., generation, model update, or advantage calculation).

### C.4 TRAINING CURVES OF CONTINUOUS CONTROL TASKS

## D THE USE OF LARGE LANGUAGE MODELS IN THIS PAPER

In the preparation of this paper, we utilized LLMs as a general-purpose writing assistance tool. Specifically, LLMs were employed for proofreading and polishing the language of certain sections to improve clarity and readability. The final title of this paper was also partially inspired by suggestions from an LLM.

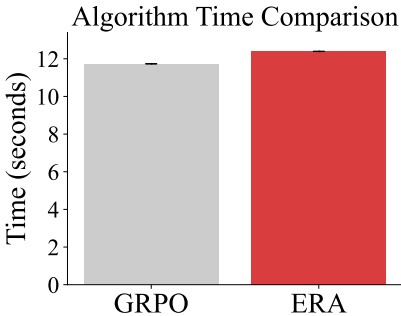

Figure 36: Comparison of computation time between GRPO and ERA, measured by `timing_s/old_log_prob` at the first step. ERA introduces only about a $5.6\%$ overhead.

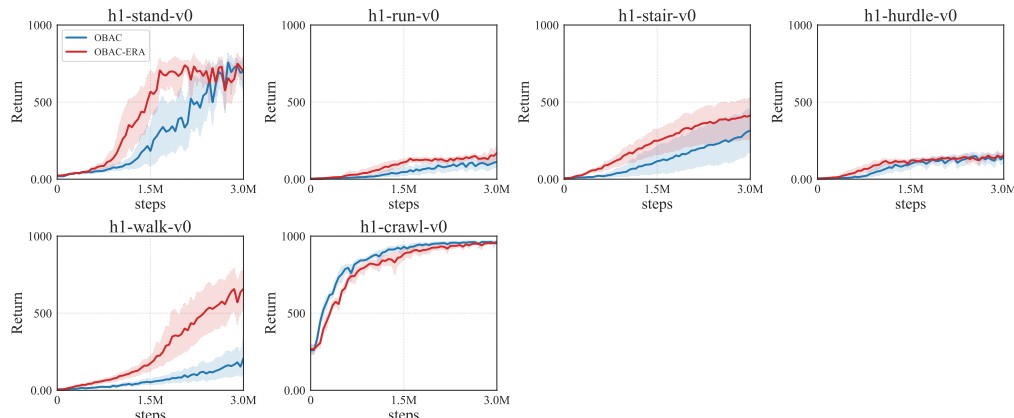

Figure 37: Training curves of OBAC and OBAC-ERA on HumanoidBench environments.

However, we clarify that the core contributions of this work were conceived and developed entirely by the human authors. The design of the methodology, the execution of experiments, and the interpretation of the results did not involve the use of LLMs. All content, including text, figures, and tables, was carefully reviewed, edited, and verified by the authors to ensure scientific accuracy and integrity.

Finally, we would like to express our gratitude for the occasional sparks of inspiration and the assistance in debugging code provided by our LLM friends. Their contribution, while not qualifying for co-authorship, was nonetheless appreciated.

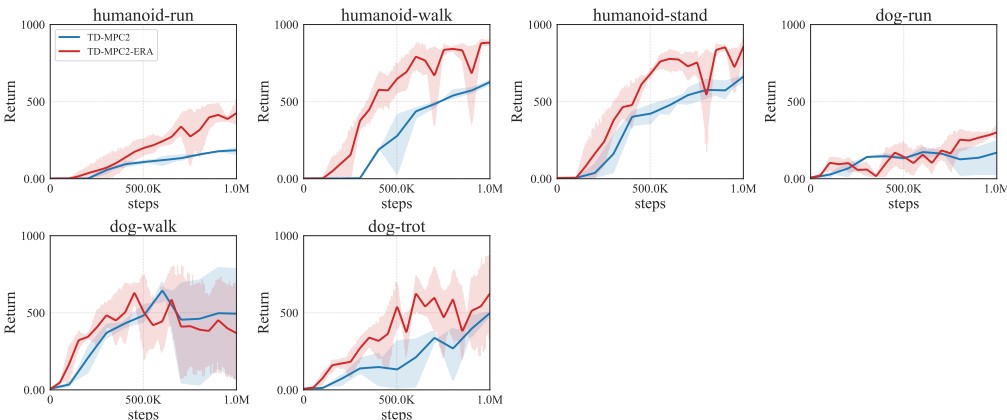

Figure 38: Training curves of TD-MPC2 and TD-MPC2-ERA on DMC environments.

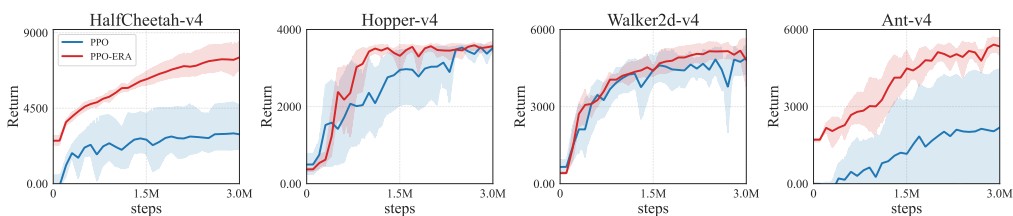

Figure 39: Training curves of PPO and PPO-ERA on Mujoco Gym environments.

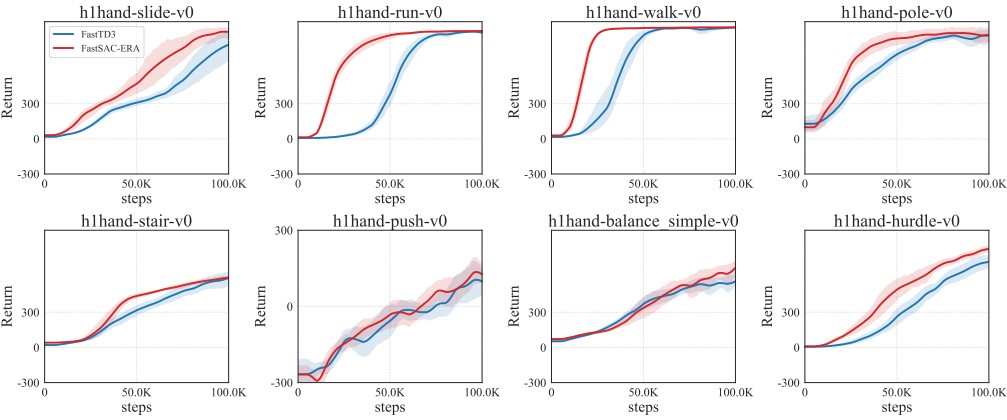

Figure 40: Training curves of FastTD3 and FastSAC-ERA on HumanoidBench environments.

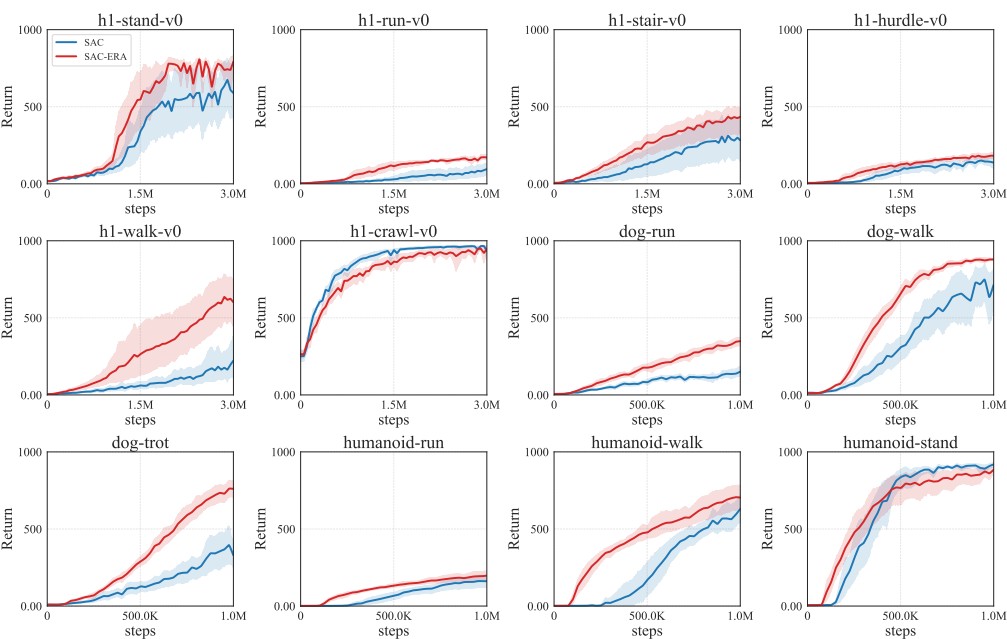

Figure 41: Training curves of SAC and SAC-ERA on HumanoidBench and DMC environments.

