# OpenReview forum: "Entropy Regularizing Activation: Boosting Continuous Control, Large Language Models, and Image Classification with Activation as Entropy Constraints"
_ICLR.cc/2026/Conference — ICLR 2026 Poster_

### Official Review · Reviewer_1XEa · 2025-10-17

**Soundness:** 2
**Presentation:** 2
**Contribution:** 3
**Rating:** 6
**Confidence:** 4

**Summary:**

The paper presents a novel way to enforce a minimal entropy for the final distribution returned by a neural network. This is typically required for reinforcement learning tasks: you want to learn policies that are not overly deterministic and preserve some stochasticity, which is both good for exploration and robustness. In prior works, such constraint was softly imposed by adding an entropy term to rewards or Q-functions, and thus by maximising a modified Q-function: you maximize both cumulative rewards and an entropy of the policy. Indeed, that was one of the features of the soft-actor-critic algorithm considered as a baseline in this paper. ERA proposed by the paper introduces a new post-activation function that imposes a minimal entropy in the hard way. This is achieved by modifying the final std of the Gaussian, variable $\sigma$. This approach is further extended to classification and LLM fine-tuning with GRPO tasks. The final results show consistent improvements over many baselines over many continuous control tasks, math solving benchmarks and classic image classification.

I suggest a weak accept for the paper as the idea of imposing entropy through an activation is original and very useful in practice, as shown in the experiments, however the paper can benefit from improved writing, more clear problem formalism and rigourous proofs.

**Strengths:**

The idea of designing a special activation function that would enforce a minimal entropy is interesting and very useful in practice. With the help of many experiments, the authors showed that this activation function is sufficient to 1) keep entropy at a required level, 2) achieve superiour results compared to the trainings without ERA. I have appreciated that authors made a lot of experiments in different settings: for continuous control tasks they checked many difficult problems from Humanoidbench, DMC and Mujoco; for LLM  — finetuning 3 different sets of math competitions; for image classification — Cifar 10 and Imagenet. While improving results for many fixed architectures, ERA introduces only a slight time overhead.

**Weaknesses:**

The paper sometimes lacks mathematical rigour and details. It is the most noticeable when the authors try to formalise the RL for both classic control tasks and LLM. The authors do not introduce what is $s$, $a$, $x$ and $y$, therefore it is also not clear what is $y^{1:K}$. While it is indeed clear what are states and actions in context of RL, for readers that are not familiar with RL for LLM, which is relatively recent, understanding what are $x$ and $y$ and how they are connected to state-action formalism can be challenging. Some equations are written in complicated way, while can be simplified: for example $\exp(\max(…, \log(\sigma))$ can be simplified by writing  $\max(\exp(…), \sigma)$. Finally, while most theoretical results look sound, the proofs in some parts are not clear and maybe erroneous (see below).

**Questions:**

**What needs clarification**
- Section 4.1: It should be more explicit at which level of the neural network ERA is applied. For example, in case of tanh transformation to get the final policy, is ERA applied before tanh or after?
- Section 4.2.1: In the main paper, it is not clear what is the exact definition of $\delta$ and how it is different from $\hat \delta$. Also it is not clear whether $\mathcal{H}’_0 = \mathcal{H}_0 + \delta$ or $\mathcal{H}’_0 = \mathcal{H}_0 + \hat \delta$. It becomes more clear after reading the proof, but it should be more transparent in the main paper as well.
- Figure 3: explain what IQM stands for.
- Proposition 2, proof: the last inequality in eq.25 and first equality in eq.26 are not clear. I think eq.26 might be wrong for the current choice of $h(x) = \ln(-x\ln x)$, while it is trivial for $h(x) = -x \exp(x)$.
- Proposition 3, proof: I don’t understand how do you get eq.35, also the reasoning after eq.35 for obtaining sufficient condition. Furthermore, eq.33 and eq.34 might not be completely true: $\Delta H = \sum_a \frac{\partial H}{\partial z_a} \Delta z_a$, but as updates are performed with respect to $\theta$ using new policy gradient formula, $\Delta z_a$ may not be equal to $\frac{\partial J}{\partial z_a}$. Check https://arxiv.org/pdf/2405.19816, Eq. 2 for further understanding.


**Technical questions**
- page 4: Preactivation mean $\mu$ and std $\hat \sigma$ are denoted in different styles, can it be homogenized?
- page 5: why traditional entropy methods are incompatible with the on-policy setting (because of entropy collapse?)
- Eq. 16: what transformation do you perform when $A_t$ is negative, but $H_{resp}< \omega_{low}$ or  $H_{resp}> \omega_{high}$?
- Figure 2: why do you compare FastTD3 vs Fast Sac-ERA in the third plot? With respect to what do you normalise the performances?
- What final transformation do you use? Tanh or truncated Gaussian?
- Table 2,3: I see some discrepancies in reported performances of base Qwen2.5-Math and Qwen2.5-Math-Instruct on Olympiad and Minerva benchmarks with respect to the original paper https://arxiv.org/pdf/2409.12122, can you please comment on why there are some differences?
- Figure 15: On which tasks did you do time measurements? I expect time complexity to be $O(D)$, while humanoid bench has tasks with D=19 and D=61.
- What kind of distributions are $\tau$-divided distributions? Can you provide an example?


**General questions**
- Do you follow any principle when choosing target entropy? In the appendix, the choice seems to be problem-dependent.
- Is ERA sensitive to the choice of the interval $[\sigma_{min}, \sigma_{max}]$? In general, how do you define these intervals?

---

> ### Author Response · Authors · 2025-11-20
>
> We thank the reviewer for the time and effort dedicated to reviewing our work. We deeply appreciate your careful and thorough feedback. In the following, we address each of your concerns.
>
> **Note: Modifications made to the paper in response to this reviewer are highlighted in Dark Gold (154, 152, 54).**
>
> ---
>
> **Q1**: "The authors do not introduce what is $s$, $a$, $x$ and $y$, therefore it is also not clear what is $y^{1:K}$."
>
> **A**:
> We have added the definitions of these symbols in **Section 3** to improve clarity.
>
> ---
>
> **Q2**: "Equation $\exp (\max(...,\log \sigma))$ can be simplified to $\max(\exp(...),\sigma)$."
>
> **A**:
> We use $\exp(\max(\dots, \log \sigma_{\min}))$ because, in the actual implementation, we first apply the `max` operation to clamp the range before applying the exponential function. This ensures numerical stability and aligns the formulation with the practical implementation.
>
> ---
>
> **Q3**: "Section 4.1: It should be more explicit at which level of the neural network ERA is applied. For example, in case of tanh transformation to get the final policy, is ERA applied before tanh or after?"
>
> **A**:
> We have added explicit clarification in **Section 4.1**, stating that ERA is applied at the output layer of the neural network. For continuous control tasks, ERA is applied to the mean and standard deviation parameters before the tanh transformation. We have also clarified this in **Section 4.2.1**.
>
> ---
>
> **Q4**: "Section 4.2.1: In the main paper, it is not clear what is the exact definition of $\delta$ and how it is different from $\hat{\delta}$."
>
> **A**:
> We have added explicit definitions for $\delta$ and $\hat{\delta}$ in **Section 4.2.1**. Specifically, $\delta$ represents the bounding bias such that $\mathcal{H}_0' = \mathcal{H}_0 + \delta$, while $\hat{\delta}$ is the parameter used during actual learning to estimate $\delta$, which can be set as a constant or adjusted dynamically.
>
> ---
>
> **Q5**: "Figure 3: explain what IQM stands for."
>
> **A**:
> IQM stands for *Interquartile Mean*, a statistical measure commonly used in reinforcement learning. When evaluating multiple random seeds, IQM avoids being overly influenced by outliers. Instead of averaging all results, it typically discards the top 25% and bottom 25% of scores and averages the remaining middle range. In our entropy ablation experiment, we applied a slightly simplified version of this concept: we ran 5 seeds, removed the highest and lowest scores (approximating the standard IQM procedure but simpler to implement), averaged the remaining three seeds, and then applied the normalized-score calculation to obtain the final reported numbers.
>
> ---
>
> **Q6**: "Proposition 2, proof: the last inequality in eq.25 and first equality in eq.26 are not clear. I think eq.26 might be wrong for the current choice of $h(x)=\ln(-x\ln x)$, while it is trivial for $h(x)=-x\exp(x)$"
>
> **A**:
> Thank you for pointing this out. It was a typo. We have corrected $h(x)$ to $h(x)=-x\exp(x)$ in the proof of **Proposition 2**. This correction does not affect the correctness of our derivation.
>
> ---
>
> **Q7**: Doubts regarding the derivation of Proposition 3.
>
> **A**:
> We appreciate your detailed review. Regarding the derivation of **Proposition 3**, we offer the following clarifications:
> (1) We approximate entropy differences by treating the logits as policy parameters, following the same approximation used in Cui et al. (https://arxiv.org/abs/2505.22617). This simplification is necessary for analytical tractability; without it, the theoretical analysis of the model’s behavior would become prohibitively complex.
> (2) We have slightly adjusted the arguments to improve clarity. Note that this bound is quite loose, and empirical results indicate a much higher entropy lower bound in practice. For instance, although the bound derived in Proposition 3 lies below $\omega_{\text{low}}$, the observed $H_{\text{resp}}$ never drops below $0.8$ when $\omega_{\text{low}}=0.45$ and $\omega_{\text{high}}=3.0$ (Figure 30). In our view, the key intuition comes from Eq. (35), which shows that the optimization process is equivalent to maximizing the original policy gradient plus a KL term whose coefficient is not analytically tractable.
>
> ---
>
> **Q8**: "page 4: Preactivation mean $\mu$ and std $\hat{\sigma}$ are denoted in different styles, can it be homogenized?"
>
> **A**:
> The parameters before passing through the ERA activation do not possess statistical significance (i.e., they are not the true mean and standard deviation), so we use different symbols to distinguish them. In fact, for the mean, ERA applies an identity mapping, so using the same symbol would not cause confusion. However, for the standard deviation, the parameter before the ERA activation has no physical meaning, so we use a different symbol to differentiate it.

---

> > ### Author Response · Authors · 2025-11-20
> >
> > **Q9**: "page 5: why traditional entropy methods are incompatible with the on-policy setting (because of entropy collapse?)"
> >
> > **A**:
> > These methods typically rely on either importance-sampling ratios or KL-based loss terms that only arise in off-policy (or PPO-style) training. For example, the original GRPO uses clipping ratios of $1-0.2$ and $1+0.2$, and the clip-higher method simply increases the upper clipping ratio to $1+0.28$. Similarly, the KL-Cov method adds KL regularization only to tokens with high covariance. These mechanisms depend on quantities not available in a purely on-policy regime, making them unsuitable for that setting.
> >
> > ---
> >
> > **Q10**: "what transformation do you perform when $A_t$ is negative, but $H_{\text{resp}}<\omega_{\text{low}}$ or $H_{\text{resp}}>\omega_{\text{high}}$?"
> >
> > **A**:
> > We do not modify the logits and advantages in this case. We have added parentheses in our paper to improve clarity. The reason we do not apply the transformation when $A_t$ is negative follows directly from the proof of Proposition 3. If we were to apply the same modification in the negative-advantage case, Eq. (31) would reduce to the standard policy gradient, rendering the transformation ineffective. An additional intuitive explanation is that overfitting is primarily driven by positive samples, so we impose the constraint only on them to mitigate this effect.
> >
> > ---
> >
> > **Q11**: "Figure 2: Why do you compare FastTD3 vs Fast Sac-ERA in the third plot? With respect to what do you normalise the performances?"
> >
> > **A**:
> > The authors of FastTD3 implemented and compared against FastSAC in their work, demonstrating that FastTD3 performs significantly better than FastSAC on HumanoidBench (https://arxiv.org/pdf/2505.22642, Page 8, Figure 8). Therefore, we selected the stronger baseline, FastTD3, to demonstrate that FastSAC with ERA can outperform the current state-of-the-art algorithm (FastTD3). Regarding the normalized score calculation, we have added a discussion in **Appendix A.1.6**.
> >
> > ---
> >
> > **Q12**: "What final transformation do you use? Tanh or truncated Gaussian?"
> >
> > **A**:
> > In our main experiments, we use a **truncated normal distribution** for the final transformation. In the appendix, we provide a detailed ablation study over different distribution choices, and the results show that the truncated Gaussian performs the best. This is stated in **Appendix C.1.1**.
> >
> > ---
> >
> > **Q13**: "Table 2,3: I see some discrepancies in reported performances of base Qwen2.5-Math and Qwen2.5-Math-Instruct on Olympiad and Minerva benchmarks with respect to the original paper https://arxiv.org/pdf/2409.12122, can you please comment on why there are some differences?"
> >
> > **A**:
> > The discrepancies likely arise from differences in the evaluation setup. Factors such as the testing framework, answer-parsing rules, decoding temperature, prompting format, and other implementation details can all influence the final scores. It is common for reproduced results to vary across papers for these reasons. In fact, many recent works—including GMPO—report performance lower than the original paper’s numbers when evaluated under their own pipelines. ERA and all baselines were trained on the same Qwen2.5-Math-7B base model and evaluated using the exact same framework.
> >
> > ---
> >
> > **Q14**: "Figure 15: On which tasks did you do time measurements? I expect time complexity to be $O(D)$, while humanoid bench has tasks with D=19 and D=61."
> >
> > **A**:
> > We conducted measurements on `h1hand-hurdle-v0` ($D=61$). The test scenario was based on FastTD3, an algorithm specifically optimized for runtime speed and highly sensitive to computational load. Transitioning from TD3 to SAC requires the policy to output an additional standard deviation dimension, which increases the computational load. In general SAC training scenarios, the time increase is only about 3%. We have added a relevant discussion in **Appendix C.1.11**.
> >
> > ---
> >
> > **Q15**: "What kind of distributions are $\tau$-divided distributions? Can you provide an example?"
> >
> > **A**:
> > $\tau$-divided distributions refer to distributions where all probability values are divided by $\tau$. For example, given a discrete distribution $P(X=x_i)=p_i$, the corresponding $\tau$-divided distribution is $P_{\tau}(X=x_i)=\frac{p_i}{\tau}$. **This is not a true probability distribution (it does not satisfy the normalization condition)**; rather, it is an intermediate variable introduced in the discrete space to facilitate mathematical derivation.

---

> > > ### Author Response · Authors · 2025-11-20
> > >
> > > **Q16**: "Do you follow any principle when choosing target entropy? In the appendix, the choice seems to be problem-dependent."
> > >
> > > **A**:
> > > We have added a new section, **Appendix C.1.7**, to discuss the principles of target entropy selection. Overall, ERA is not sensitive to the choice of target entropy; SAC and PPO outperformed baselines in the vast majority of tested ranges with significant improvements. For continuous control algorithms like SAC, the heuristic target entropy selection principle of $-\mathcal{A}$ (widely adopted by mainstream RL libraries like Stable Baselines3) also performed well in our experiments.
> > > For instance, on TD-MPC2, we achieved significant improvements across multiple tasks by selecting the target entropy as $-\mathcal{A}$. In practice, we recommend prioritizing the parameters provided in our paper, as they performed well across the extensive tasks we tested. If further tuning is needed, fine-tuning can be performed around the target entropy.
> > >
> > > ---
> > >
> > > **Q17**: "Is ERA sensitive to the choice of the interval $[\sigma_{\text{min}}, \sigma_{\text{max}}]$? In general, how do you define these intervals?"
> > >
> > > **A**:
> > > We have added a discussion on the selection of the interval $[\sigma_{\text{min}}, \sigma_{\text{max}}]$ in **Appendix C.1.6**. For the SAC algorithm in continuous control tasks, there are common empirical values (e.g., $[-20, 2]$). We tested three sets of values and found that ERA is insensitive to these choices, with little impact on performance; for example, in environments like `dog-run`, the performance was indistinguishable.
> > > In practice, we recommend using the parameter $[-8, 0]$ provided in our paper. These parameters remained **unchanged** across all algorithms and tasks tested in our work. According to our experimental results, these parameters perform well and are robust across all tested tasks. Modifying them to other values has negligible impact on performance.

---

> ### Author Response · Authors · 2025-11-26
>
> Dear Reviewer 1XEa,
>
> We sincerely appreciate your time and effort in reviewing our manuscript and providing valuable feedback. **Thank you very much for acknowledging the interesting and practical nature of our proposed activation function, as well as for appreciating the extensive experiments we conducted across various settings, including continuous control tasks, LLM fine-tuning, and image classification.**
>
> As the rebuttal period is nearing its end, we would like to kindly inquire whether our responses have adequately addressed your concerns. We have provided detailed clarifications in our reply, and we would be more than happy to discuss any remaining questions you might have. We would be deeply grateful if you could kindly reconsider raising the score in light of our efforts and clarifications.
>
> Thank you again for your consideration.
>
> Best regards, The Authors

---

> ### Comment · Reviewer_1XEa · 2025-11-26
> **Further questions and comments**
>
> I thank the authors for providing further clarifications and answers to my questions. I also commend the improved presentation of your results and the extra ablation studies that you performed.
>
> I would like to return back to some of the questions.
>
> Q.6: Thank you for correcting it in the appendix, though I have noticed that the text of the main paper remained unchanged. Moreover, I have an additional question regarding $\hat h^{-1}$. How did you obtain this approximation? Is it an approximation for an old and apparently wrong $h^{-1} $, where $h = \ln(-x\ln x)$? If I understand correctly, you use this $\hat h$ to obtain ERA for discrete classification, so it is important in practice.
>
> Q.7: Thank you for better clarifying the assumptions, then the proof makes more sense in this context. I have also noticed that you updated the end of the proof providing more explanations, though it still not clear for me how do you obtain the lower bound on $\beta$. Also, I find the following explanation confusing and self-contradictory: when  $\mu = \mathbb{E} H_{resp} \leq \mathcal{H}_0$, the expected change of total entropy is non-negative, therefore $H_{resp} \geq \mathcal{H}_0$.
>
> Q.14 Can you please add in your paper the exact name of the environment you mentioned in your reply.
>
> Q.15 I think it is important to mention that $\tau$-divided distributions are not probability distributions in the paper as well, so that to avoid any future confusion.
>
> Q.16 Thank you for doing this ablation study. To further clarify the writing in the paper, the compensation factor that you are referring to, is it $\hat \delta$ from the main paper?
>
> On the side note, I just noticed that Figure 2 in the main paper, doesn't show any error bars or variances. I think it is better to include them to understand the statistical significance of the comparisons. Or consider showing aggregated results similar to Figure 9 of https://arxiv.org/pdf/2108.13264.

---

> > ### Author Response · Authors · 2025-11-27
> >
> > We thank the reviewer for the insightful questions and comments. We address your concerns below.
> >
> > ---
> > **Q1**: "Is $h^{-1}$ the inverse of the function $h(x) = \ln(-x\ln x)$? How did you obtain this approximation?"
> >
> > **A**: We are sorry for the confusion. $h^{-1}$ is the inverse of the function $h(x) = -xe^x$. We have $h^{-1}(x)=W(-x)$. $W$ is the Lambert W function. We have revised this in main text, and added more discussion in **Appendix B.2** to fully clarify this. We use a numerical approximation $\hat{h}^{-1}(x) = -\frac{1}{4}-\sqrt{2(-1-\ln (x))}+\frac{3}{4}\ln x$ to approximate the Lambert W function. We first derive this form from a easier approximation $(1 + x + \ln(-x))^{-1} \approx -1 - \sqrt{2x} -\frac{3}{4}x$ (here "-1" denotes the inverse function), which we believe is quite simple to validate with Taylor Expansion. We have also added a plot comparing the true inverse function and our approximation in **Appendix B.2** to demonstrate the accuracy of our approximation.
> >
> > ---
> > **Q2**: "Concerns about the proof in LLM entropy bound."
> >
> > **A**: (1) We further refined the writing for better clarity. The central idea is to show that the entropy derivative is non-negative once the entropy falls below a fixed threshold. Under the gradient-flow approximation (for simplicity), this ensures that the entropy will not drop below this level.
> > (2) We do not yet have a rigorous proof for the lower bound on $\beta$. However, we believe such a bound exists for our current hyperparameters $\omega_{\text{low}}, \omega_{\text{high}}$, and $k$, as the entropy curves in our experiments exhibit remarkably clean and stable behavior, strongly suggesting the presence of a lower bound.
> >
> > ---
> > **Q3**: "Add the environment name in time comparison for continuous control into the paper.
> > "
> >
> > **A**: Thank you for your suggestion. We have added the environment name for clarity.
> >
> > ---
> > **Q4**: "Clarify that $\tau$-divided distribution is not a valid probability distribution in paper."
> >
> > **A**: Thank you for your suggestion. We have added this clarification in the revised paper in **Appendix B.2**.
> >
> > ---
> > **Q5**: "Is the compensation term mentioned $\hat{\delta}$ ?"
> >
> > **A**: Yes, the compensation term mentioned is indeed $\hat{\delta}$. We have clarified this in the revised paper.
> >
> > ---
> > **Q6**: "Figure 2 has no variance or error bars."
> >
> > **A**: Thank you for your suggestion. We have added shaded areas representing the 25\% and 75\% percentiles around the mean performance curves in Figure 2 to illustrate variability across different runs.
> >
> > To further demonstrate the calculation of shaded areas in plots, we have added more implementation details in **Appendix A.1.7**. We use 25\% and 75\% percentiles to create shaded areas around the mean performance curves in the aggregated performance plots (like Figure 2). For training curves of individual environments in the appendix, we use 95\% confidence intervals to create shaded areas around the mean performance curves.
> >
> > ---
> >
> > We hope these revisions and clarifications address your concerns. Please feel free to reach out if you have any further questions or suggestions.

---

### Official Review · Reviewer_8uvP · 2025-10-19

**Soundness:** 2
**Presentation:** 2
**Contribution:** 2
**Rating:** 6
**Confidence:** 4

**Summary:**

This paper presents a way to control of policy entropies by introducing ERA, a dedicated activation function for the final layer to clip entropies within bounds. The proposed framework can be applied to generic policy gradient settings and is experimented with continuous control, discrete classification and LLM-RL training although each setting requires dedicated instantiation to integrate the idea into its training. In experimental results, models trained with ERA outperform baselines across those three settings.

**Strengths:**

- ERA is a generic concept although this requires dedicated instantiations for each task and model.
- Authors show strong performance improvements of each instantiation in through extensive experiments.
-  For continuous control experiments, the authors show that ERA improves performance over different RL algorithms, which shows its generality.

**Weaknesses:**

- It's not clear why SAC without the entropy term in targets is outperformed by ERA since Q-function is basically trained through the same objective between two algorithms in this case. Is this because ERA provides a hard way to constrain the entropy range? I believe that an analysis on behavior of entropy helps to clarify this.
- This is a minor point. In the LLM-RL setting, ERA needs to modify not only the logits of tokens, but also advantages. This seems beyond just an activation layer although the paper claims it proposes activation functions.
- Authors should provide how many trials they conducted each benchmark to get statistics not only for continuous control benchmarks, but also for the rest of benchmarks.

**Questions:**

I understand that LLM-RL training takes time and consumes resources. So I leave some comments as questions, not requests.
- How the number of `top 20%` was selected in the LLM-RL instantiation? Have you tested different values to see how it affects performance?
- Have you tested ERA in LLM-RL without the advantage modification? If this works out, ERA for LLM-RL can be just the activation layer change.

---

> ### Author Response · Authors · 2025-11-20
>
> We thank the reviewer for the time and effort dedicated to reviewing our work. We deeply appreciate your careful and thorough feedback. In the following, we address each of your concerns.
>
> **Note: Modifications made to the paper in response to Reviewer 8uvP are highlighted in Muted Brown (160, 82, 45).**
>
>
> ---
>
> **Q1:** "why SAC without the entropy term in targets is outperformed by ERA"
>
> **A:** Our ERA constrains entropy, requiring the model to conduct more exploration. Under such circumstances, SAC without an entropy term is completely unable to explore, which is why our approach yields better results.
>
> **Q2:** "In the LLM-RL setting, ERA needs to modify not only the logits of tokens, but also advantages. This seems beyond just an activation layer although the paper claims it proposes activation functions."
>
> **A:** We believe that any simple transformation that affects model learning can be termed as an "activation." Therefore, adjusting the advantages can also be referred to as an activation. In our proof, specifically Equation 31, the transformation of advantages serves to counteract the $k$ generated by the transformation of logits. This allows us to more clearly direct our conclusion, which is that ERA naturally sharpens or flattens the output distribution.
>
> **Q3:** "Authors should provide how many trials they conducted each benchmark to get statistics not only for continuous control benchmarks, but also for the rest of benchmarks."
>
> **A:** We conducted only one experiment in LLM-RL but performed multiple tests across various benchmarks. This is because: 1. The training data for LLMs is DAPO, which is a large dataset. When the number of training steps is high, random factors such as seeds have minimal impact on training. This is also a common practice in the field. 2. The training cost for LLMs is extremely high, making it impractical to train multiple times and take averages. 3. We trained on a single DAPO dataset but tested on multiple test sets, with little overlap between DAPO and the test sets. The consistent improvements across multiple datasets also indicate that our method is stable.
>
> **Q4:** "How the number of top 20% was selected in the LLM-RL instantiation? Have you tested different values to see how it affects performance?"
>
> **A:** In our original text, we provided the specific process in Listing 6. This value follows previous work “Beyond the 80/20 Rule: High-Entropy Minority Tokens Drive Effective Reinforcement Learning for LLM Reasoning”. We additionally provided in **Appendix C.3.4** the method of selecting only the top 10% of tokens with the highest entropy from each rollout to calculate $H_{\text{resp}}$. The $H_{\text{resp}}$ calculated from the top 10\% tokens is naturally higher than that from the top 20\%. As a result, fewer samples meet the condition $H_{\text{resp}} < \omega_{\text{low}}$ compared to the version using 20\%. Therefore, the constraining power of entropy is limited, and the results lie between ordinary GRPO and ERA. However, it still achieves significant improvements compared to GRPO, which is similar to the findings in the original paper "Beyond the 80/20 Rule."
>
> **Q5:** "Have you tested ERA in LLM-RL without the advantage modification? If this works out, ERA for LLM-RL can be just the activation layer change."
>
> **A:** Due to limitations in time and computational resources, we conducted an experiment using Qwen2.5-1.5B on the MATH dataset, reporting the pass@1 results for AIME and AMC. It can be seen that the transformation of advantages does have some effect, although it is not sufficiently significant. Moreover, our main results on Qwen2.5-Math-7B were obtained without applying any advantage transformation, as noted in **Appendix A.3.1**. We will conduct more comprehensive experiments to demonstrate this point.
>
> |               | math500 | minerva | olympiad | aime24 | amc23 | avg  |
> |---------------|---------|---------|----------|--------|-------|------|
> | w/o Adv scale | 66.8    | 30.9    | 34.4     | 13.3   | 40.0  | 37.1 |
> | w/ Adv scale  | 69.6    | 29.8    | 35.0     | 13.3   | 47.5  | 39.0 |

---

> ### Author Response · Authors · 2025-11-26
>
> Dear Reviewer 8uvP,
>
> We sincerely appreciate your time and effort in reviewing our manuscript and providing valuable feedback. **Thank you very much for acknowledging that ERA is a generic concept requiring dedicated instantiations for each task and model, and for recognizing the strong performance improvements demonstrated through our extensive experiments.**
>
> As the rebuttal period is nearing its end, we would like to kindly inquire whether our responses have adequately addressed your concerns. We have provided detailed clarifications in our reply, and we would be more than happy to discuss any remaining questions you might have. We would be deeply grateful if you could kindly reconsider raising the score in light of our efforts and clarifications.
>
> Thank you again for your consideration.
>
> Best regards, The Authors

---

> ### Author Response · Authors · 2025-11-26
>
> Dear Reviewer 8uvP,
>
> Regarding your question about testing ERA in LLM-RL without the advantage modification.
>
> For completeness, we also evaluate ERA with and without advantage transformation on Qwen2.5-Math-7B, with results provided in **Appendix C.3.3.** The conclusion remains consistent with the 1.5B experiment: the transformation of advantages provides some improvement, but the effect is relatively modest and both variants perform similarly well.
>
> | Qwen2.5-Math-7B | aime24 | aime25 | amc23 | math | minerva | olympiad | avg  |
> |-----------------|--------|--------|-------|------|---------|----------|------|
> | w/o Adv scale   | 37.5   | 16.9   | 72.8  | 84.6 | 42.6    | 46.5     | 50.2 |
> | w/ Adv scale    | 36.0   | 21.0   | 76.6  | 85.4 | 40.1    | 46.8     | 51.0 |

---

### Official Review · Reviewer_x7W9 · 2025-10-31

**Soundness:** 2
**Presentation:** 1
**Contribution:** 2
**Rating:** 4
**Confidence:** 4

**Summary:**

The authors propose in this paper a set of parameter transformations that map input distribution parameters (e.g. logits of a discrete distributions or mean/logstd of a Gaussian distribution) to parameters that are above an entropy threshold. The main motivation of the paper is that entropy is a key factor in RL as it helps with exploration, yet optimizing an entropy regularized objective can be challenging as it requires to carefully choose the entropy weight. Instead of using soft entropy regularization, the authors propose to introduce entropy regularization directly in parameter space, effectively restricting the search space to policies having an entropy above a given threshold.

In details, the authors propose three parameter transformations for i) discrete distributions ii) squashed (Tanh) Gaussian distributions and iii) truncated Gaussian distributions. The authors integrated the latter two transformations for continuous control RL tasks with bounded actions showing improved performance over baselines such as SAC or TD-MPC2, and the first transformation for the fine-tuning of LLMs on math problem datasets or for training a ResNet-50 on image classification tasks; in all cases showing improvements over baselines.

**Strengths:**

The paper extends prior work on differentiable parameter projections to new distributions and has an extensive empirical evaluation spanning an impressively large number of applications and baseline algorithms, with promising empirical results.

**Weaknesses:**

- The paper does not discuss prior work even though there is existing literature extremely related to the submission. Namely, the paper [1] (*F. Otto et al.; Differentiable Trust Region Layers for Deep Reinforcement Learning; ICLR 2021*) proposes a set of parameter projections that enforce among other things KL and Wasserstein based trust regions. The paper [2] (*R. Akrour et al.; Projections for Approximate Policy Iteration Algorithms; ICML 2019*) proposes parameter transformations for KL and entropy constraints much like the current paper. [2] proposes a projection for Gaussian distributions and the current submission uses the same projection as a basis to extend the projection to Squashed and Truncated Gaussians. The contribution compared to [2] is thus clear (extending prior parameter transformations to new distributions) but I believe [1,2] should still be discussed in the related work. However, [2] also proposed a projection to enforce an entropy constraint for discrete distributions much like this submission, although the mechanisms seem at a glance a bit different: this paper scales the input logits while [2] mixes the input distribution with a uniform distribution. Since these two transformations serve exactly the same purpose (map an input discrete distribution to a distribution satisfying an entropy constraint), it would be good to at least discuss the conceptual differences between the two transformations, but ideally some additional empirical comparisons to gain insights on their practical differences would also be appreciated.

- The paper contains a few overclaims, i) first regarding SAC in l96 authors state that the variant of SAC that uses an entropy constraint with a dynamically adjusted temperature can lead to instability. I would appreciate some evidence supporting this claim. To the best of my knowledge this second version of SAC with a hard entropy constraint is the default SAC implementation in many deep RL code bases such as *stablebaselines*, and I have not heard or witnessed such instabilities before. ii) In a few places (e.g. l68 or l111) the authors state that their parameter transformations have provable guarantees, but from what I can see for squashed and truncated Guassians, the authors still need to adjust a parameter online, much like SAC as discussed in l1049, and the satisfaction of the entropy constraint is not guaranteed in for these distributions.

-  For a paper whose main contribution is to provide a better control of the entropy, I was surprised to not see comparisons between the entropy control resulting from the author's method and from the dynamic tuning performed by SAC. Especially in light of the previous two (over)claims, it would be interesting to see how both algorithm manage to comply with their entropy constraints.

- The proof in appendix B.1 feel incomplete to me. The authors state that the $\delta$ is a positive constant and although it seems trivially true for $\delta_{\text{tanh}}$, I would appreciate a proof for $\delta_{\text{TN}}$

- Wording regarding SAC-ERA's pseudocode is a bit confusing, authors talk of a soft Q-function (l933), but it seems to me that the algorithm is not in the entropy regularized RL paradigm and an unregularized Q-function is learned, making the implementation closer to TD3 than SAC.



Overall, while I'm currently leaning towards a reject, I am willing to increase my score if the authors can better contextualize their work and provide some empirical insights regarding the entropy control their method proposed compared to existing methods.

**Questions:**

- What is the difference between the proposed projection for discrete distributions and the one proposed by [2]? How do you expect conceptual differences to manifest in practice?
- What is your evidence that SAC is unstable when tuning its entropy temperature dynamically and why do you expect the online tuning of $\delta$ to be more stable?
- What are the benefits of the parameter transformations compared to enforcing an entropy constraint by tuning the temperature online (as in SAC) for image classification or LLM fine-tuning?

---

> ### Author Response · Authors · 2025-11-20
>
> We thank the reviewer for the time and effort dedicated to reviewing our work. We deeply appreciate your careful and thorough feedback. In the following, we address each of your concerns.
>
> **Note: Modifications made to the paper in response to Reviewer x7W9 are highlighted in Slate Gray (112, 128, 144).**
>
> ---
>
> **Q1:** "Missing discussion of related work: 1. F. Otto et al. (ICLR 2021) and 2. R. Akrour et al. (ICML 2019). What is the difference between the proposed projection and the one proposed by [2]?"
>
> **A:**
> We thank the reviewer for recommending these relevant works. Upon careful review, we found that the second paper (R. Akrour et al., ICML 2019) is closely related to our work, and the entropy projection scheme used in the first paper (F. Otto et al., ICLR 2021) is derived directly from the second. Consequently, we have expanded the **Related Work** section and added detailed theoretical and empirical comparisons with Akrour et al. [2] in **Appendix C.1.5 (Continuous Control)** and **Appendix C.2.1 (Image Classification)**.
>
> Theoretically, we identify a critical distinction: the projection method in [2] essentially is a uniform linear transformation of the policy distribution. We reveal that the gradient backpropagated through this differentiable projection layer is merely a linear scaling of the original gradient (identical across all dimensions) that would exist without entropy constraints. **This implies that while the projection ensures the output distribution satisfies the entropy condition numerically, it fails to provide the policy with meaningful guidance on how to distribute entropy across different dimensions.** In contrast, ERA provides rich gradient signals that guide the policy to allocate entropy rationally, thereby effectively enhancing learning. Our empirical analysis supports this: in continuous control tasks, the projection method [2] results in highly polarized dimensional entropy distributions, whereas ERA demonstrates a clear evolution from concentration to rational dispersion.
>
> Empirically, we found that ERA significantly outperforms the projection method [2] across continuous control and image classification tasks. In many cases, the projection method performed worse than the baseline, indicating limited practical efficacy. We have added these results to **Appendix C.1.5** and **Appendix C.2.1**. Notably, in LLM fine-tuning tasks, the linear interpolation projection method [2] failed to converge entirely. We attribute this failure primarily to the extremely high dimensionality of the LLM action space, which typically spans tens to hundreds of thousands of dimensions. Since the projection method applies interpolation with a uniform distribution indiscriminately across all dimensions, it inevitably forces the assignment of probability mass to numerous irrelevant dimensions (tokens). This leads to the generation of significant nonsensical output (gibberish), effectively preventing the model from learning meaningful content. These results suggest that the method in [2] acts as a post-hoc projection that does not fundamentally enable the policy to learn an optimal distribution satisfying the constraints.
>
> ---
>
> **Q2:** "SAC instability with dynamic temperature adjustment on implementations such as Stable Baselines is not well supported. Please provide evidence."
>
> **A:**
> We appreciate the reviewer pointing out this potential ambiguity. Our intended meaning was to describe the **objective conflict** between the entropy term and the reward term in the loss function during SAC training. In sparse-reward or complex environments, this conflict can lead to low learning efficiency and suboptimal returns—a phenomenon akin to a "near-collapse" of training (though not necessarily a numerical explosion). To prevent misunderstanding, we have revised the term "instability" to "**suboptimal performance**" in the Related Work section.
>
> Furthermore, to demonstrate that this phenomenon is not specific to a single implementation, we have added comparative experiments in **Appendix C.1.9** using the **Stable Baselines3 (SB3)** implementations of SAC and PPO. Under identical parameter settings, the SB3 implementations exhibited performance characteristics similar to the JaxRL baselines used in our original text. This confirms that the limitations of standard entropy regularization mechanisms are general and not confined to specific codebases.

---

> > ### Author Response · Authors · 2025-11-20
> >
> > **Q3:** "ERA still needs to adjust a parameter online, much like SAC, and the satisfaction of the entropy constraint is not guaranteed."
> >
> > **A:**
> > It is correct that the version of ERA using a learnable compensation term requires online parameter adjustment. Our previous proof relied on the convergence framework of SAC. In the revised manuscript, we have updated the proof in **Appendix B.1** using a **two-timescale stochastic approximation framework**. We prove that even when the policy parameters and the compensation term parameter are updated simultaneously, the compensation parameter converges to the necessary value required to satisfy the entropy constraint.
> >
> > ---
> >
> > **Q4:** "Lack of comparisons between the entropy control resulting from the author's method and from the dynamic tuning performed by SAC."
> >
> > **A:**
> > We have added a comparative experiment in **Appendix C.1.10** analyzing the entropy control dynamics of SAC-ERA versus the SAC baseline in the `Dog-Trot` task. The results show that SAC-ERA stably maintains the policy entropy near the preset threshold throughout the training process, whereas the entropy in the SAC baseline exhibits significant fluctuations. This confirms that SAC-ERA offers superior control capabilities.
> >
> > Additionally, we analyzed the distribution density of the log standard deviation (`log std`) across dimensions. The results reveal distinct patterns: SAC-ERA's `log std` distribution shows an evolution from concentration to dispersion, indicating that the method effectively guides the policy to allocate entropy rationally across dimensions. In contrast, the SAC baseline shows a tendency for faster, less structured diffusion.
> >
> > ---
> >
> > **Q5:** "Prove that $\delta_{\text{TN}}$ is positive."
> >
> > **A:**
> > We have added a formal proof demonstrating that $\delta_{\text{TN}}$ is positive in **Appendix B.1.4**.
> >
> > ---
> >
> > **Q6:** "SAC-ERA's pseudocode does not use soft Q function."
> >
> > **A:**
> > To strictly align the terminology with the algorithm's structure, we have changed "soft Q-function" to "Q-function" in the SAC-ERA pseudocode in **Appendix A.3**. In SAC-ERA, we remove the original entropy regularization term from the reward function; thus, the agent learns a standard, non-entropy-regularized Q-function. This decouples the reward maximization from the entropy constraint, allowing the policy to focus on optimizing the reward signal without objective conflict.
> >
> > ---
> >
> > **Q7:** "What are the benefits of the parameter transformations compared to enforcing an entropy constraint by tuning the temperature online (as in SAC) for image classification or LLM fine-tuning?"
> >
> > **A:**
> > We have added comparative experiments using entropy term regularization for Image Classification (**Appendix C.2.1**) and LLM fine-tuning (**Section 5.3 Table 2**).
> >
> > In both domains, we observed that adding a standard entropy term often leads to performance inferior to the baseline.
> > * **Image Classification:** For ResNet-50, while an entropy coefficient of `1e-3` slightly improved Top-1 accuracy, it did not improve Top-5 accuracy compared to the baseline. Furthermore, coefficients of `1e-2` and `1e-4` resulted in Top-1 accuracy strictly worse than the baseline. This suggests that simply encouraging higher entropy via a loss term can interfere with the model's ability to fit the training data, rather than effectively improving generalization. We also tested SAC-like dynamic temperature tuning in this context, which yielded similar negative results: among 3 tested entropy targets (1.2, 1.25, 1.5), none improved both Top-1 and Top-5 accuracy over the baseline.
> > * **LLM Fine-tuning:** Introducing an entropy term caused performance degradation. Large Language Models are powerful enough to "hack" the reward function by outputting high-entropy gibberish to maximize the entropy term, leading to a failure to learn meaningful content. This aligns with findings in recent work, such as *The Entropy Mechanism of Reinforcement Learning for Reasoning Language Models* (arXiv:2505.22617).
> >
> > Overall, while standard entropy regularization failed to improve (and often harmed) performance in these domains, ERA consistently delivered significant performance gains, demonstrating its effectiveness and robustness.

---

> ### Author Response · Authors · 2025-11-26
>
> Dear Reviewer x7W9,
>
> We sincerely appreciate your time and effort in reviewing our manuscript and providing valuable feedback. **Thank you very much for acknowledging the thoroughness of our experiments across multiple domains, including DRL, LLM RL, and Image Classification.**
>
> As the rebuttal period is nearing its end, we would like to kindly inquire whether our responses have adequately addressed your concerns. We have provided detailed clarifications in our reply, and we would be more than happy to discuss any remaining questions you might have. We would be deeply grateful if you could kindly reconsider raising the score in light of our efforts and clarifications.
>
> Thank you again for your consideration.
>
> Best regards, The Authors

---

> ### Comment · Reviewer_x7W9 · 2025-11-26
>
> Thank you for your answer.
>
> I. Regarding discussion w.r.t. prior work:
> 1. As mentioned in my initial review, the contribution compared to [2] for continuous actions is clear since previous projection was only for Gaussian distributions and you have extended it to Squashed/Truncated Gaussians. I was asking to provide more insights regarding the projection for discrete distributions. As such, looking at gradients of projections in Appendix C.1.5 is an interesting way to gain insights on projections, but it would be good to do it for the discrete distribution projection, not the continuous one.
> 2. In the experiments, line 2076, you state that the discrete projection from [2] scale logits. This is incorrect, it mixes probabilities with that of a uniform distribution with a differentiable mixing weight computed from the input distribution. This suggest that you did not implement the method correctly.
> 3. The changes to the related work sections in line 101 are incorrect or at the very least ambiguous. I do not think there are conceptual differences between your projection or that of [2], gradients will flow through both projections when learning the policy and [2] is not in any way similar to posterior sampling or a post-processing method. The differences between the two projection is in the mathematical details of their definition, not at a higher conceptual level.
>
> II. Objective conflict in SAC:
> I do not understand your response. The second version of SAC considers a hard constraint on the entropy and there isn't a conflict between objectives that is fundamentally different from your approach which is also about enforcing a hard constraint vs a soft bonus with a fixed weight.
>
> III. Two-time scales and comparisons w.r.t. to SAC's entropy enforcing dynamics:
> Thank you for the addition. I believe the same arguments could be made to justify SAC's online adjustments of the temperature weight, but I appreciate the addition. The new plot comparing entropy between SAC and ERA also show meaningful differences, with notably less oscillations for ERA around the target entropy. How do you explain this difference since both are tracking entropy constraint by online adjustment of a parameter. Is it because a large part of ERA's entropy constraint satisfaction is due to the projection of the unsquashed Gaussian and the online tuning has to adjust for a smaller drift leading to less oscillations? Or could SAC also be made more stable by better tuning of the learning rate of the entropy weight?
>
> IV. Proof for positivity of $\delta_{\text{TN}}$
> The new proof is mainly based on hand wavy arguments. Please provide a complete proof, or provide references for the claim made in point 1 and 2 around line 1268 for both the entropy inequalities.
>
> V. Image classification: you state to have used '3 entropy targets (1.2, 1.25, 1.5)' for the dynamically adjusted entropy coefficient. What is the entropy target used for ERA? Why is performance not changing in Fig. 27 for ERA when the entropy target changes? Isn't ERA using the same target as the online adjusted algorithm? Could you provide entropy plots for both methods? In an RL context, I can understand that it is not only important to satisfy the entropy target, but how fast it is satisfied can affect performance as it changes the data collected and so on. But for a static setting as in image classification, I have a really hard time believing there would be meaningful differences between enforcing entropy constraint with ERA or by doing a two-time scale optimization on an entropy weight. To me this indicates rather low effort in tuning both methods or in establishing a fair comparison between the methods.

---

> > ### Author Response · Authors · 2025-11-27
> >
> > We appreciate valuable feedback and suggestions from Reviewer x7W9. In the following, we address each of your concerns.
> >
> > ---
> >
> > **Q1**: "Please provide more insights regarding the projection for discrete distributions."
> >
> > **A**:
> > Thank you for your advice. We believe that [2] is extremely *important and insightful*. However, in terms of approach, ours is different, especially for LLMs. We have now added a detailed analysis on the gradient flow of the projection method for discrete distributions in **Appendix C.2.1**. Analogous to the analysis for continuous distributions in Appendix C.1.5, we found that the gradient flow characteristics of the discrete projection method is also gradient from cross entropy loss multiplied by a uniform factor across all dimensions. This indicates that the discrete projection method does not provide dimension-specific entropy allocation information during backpropagation. ERA in image classification, on the other hand, provides dimension-specific information in gradient flow (by this we mean that apart from the gradient from cross entropy loss, there is an additional gradient component from the entropy constraint that varies across dimensions). This additional information helps the model learn to allocate entropy across different classes, which is beneficial for improving classification performance.
> >
> > For LLMs, since post-training must be performed on an existing model, we cannot simply insert an additional architectural layer; doing so would degrade performance and lead to model collapse. Moreover, in RL training, directly projecting logits to enforce a per-token entropy lower bound is impractical. Due to the intrinsic structure of natural language, most tokens are highly deterministic, and forcing them to maintain high entropy would introduce unnatural alternatives and easily corrupt the entire response.
> >
> > Instead, we adopt a different strategy for LLMs. We keep the model parameters unchanged, which means we **rollout samples on original parameters**, and then apply an inverse logic when computing logits for the sampled text: if the model needs to increase the entropy of its output distribution, we first transform the distribution to be more peaked. By interpreting the samples as if they were drawn from a sharpened policy, we mitigate overfitting while still promoting exploration. This mechanism is fundamentally different from the approach in [2]—providing substantial novelty—while still ensuring an average entropy lower bound, as shown in Appendix B.3. Consequently, the mechanism functions more like an activation applied to the logits rather than a direct projection onto the probability simplex.
> >
> > We believe that studying [2] has provided us with more ideas and inspiration. We can conduct more in-depth research on [2] and our ERA in the future, further exploring what constitutes a better entropy allocation mechanism.
> >
> > ---
> >
> > **Q2**: "Calling the discrete projection from [2] 'scale logits' is incorrect, and this suggest that you did not implement the method correctly."
> >
> > **A**: We apologize for the confusion caused by our previous terminology 'scale logits'. We have revised this to 'linear interpolation' to accurately reflect the method. We respectfully confirm that our implementation adheres strictly to the formulation in [2]. To demonstrate this transparency and dispel concerns regarding correctness, we provide the core snippet of our implementation below:
> >
> > ```python
> > class EntropyConstraint:
> >     def __init__(self, beta: float, num_classes: int, eps: float = 1e-12):
> >         ...
> >
> >     def project(self, logits: torch.Tensor):
> >         probs = torch.softmax(logits, dim=-1)
> >         entropy = -(probs * torch.log(probs.clamp_min(self.eps))).sum(dim=-1)
> >         raw_entropy_mean = entropy.mean().detach()
> >
> >         beta = logits.new_tensor(self.beta)
> >         log_num_actions = logits.new_tensor(self.log_num_actions)
> >         uniform_prob = logits.new_tensor(self.uniform_prob)
> >
> >
> >         needs_projection = entropy < (beta - 1e-6)
> >         if not torch.any(needs_projection):
> >             return logits, raw_entropy_mean, raw_entropy_mean
> >         denom = (log_num_actions - entropy).clamp_min(1e-6)
> >         alpha = torch.clamp((log_num_actions - beta) / denom, 0.0, 1.0)
> >         alpha_tensor = torch.where(needs_projection, alpha, torch.ones_like(alpha))
> >
> >         alpha_tensor = alpha_tensor.unsqueeze(-1)
> >         mixed_probs = probs * alpha_tensor + (1.0 - alpha_tensor) * uniform_prob
> >         mixed_probs = mixed_probs.clamp_min(self.eps)
> >         mixed_probs = mixed_probs / mixed_probs.sum(dim=-1, keepdim=True)
> >
> >         projected_logits = torch.log(mixed_probs)
> >         projected_entropy = -(mixed_probs * torch.log(mixed_probs)).sum(dim=-1)
> >         projected_entropy_mean = projected_entropy.mean().detach()
> >         return projected_logits, raw_entropy_mean, projected_entropy_mean
> > ```

---

> > ### Author Response · Authors · 2025-11-27
> >
> > **Q3**: "The differences between ERA and the previous projection method is in the mathematical details of their definition, not at a higher conceptual level."
> >
> > **A**: We appreciate your perspective. We have revised the **Related Work** and **Section4.1** according to your suggestion. While pointing out that there are significant differences both in mathematical definition and utility in modern RL settings, we have acknowledged previous works [1,2] as pioneers in exploring entropy constraints without explicit entropy terms, and stated them as a previously existing instantiation of our ERA definition in **Section 4.1**.
> >
> > Still, we would like to emphasize that our work is not a trivial extension of previous works. **We formalize a more general definition of entropy constraint without explicit entropy terms**, and propose novel mathematical transformations for both continuous and discrete spaces that are distinct from previous works. More importantly, our methods demonstrate significant practical advantages in modern, complex RL environments, as shown in our experimental results.
> >
> > ---
> >
> > **Q4**: "There isn't a conflict between cumulative reward and entropy objectives that is fundamentally different from your approach."
> >
> > **A**: We appreciate your comments. However, we believe there still is a fundamental difference in the nature of the conflict between cumulative reward and entropy objectives in SAC compared to ERA.
> >
> > Let's consider the optimization of the policy parameters in SAC. The gradient update for the policy parameters can be decomposed into two components:
> > 1. Gradient from Q-function: $\nabla_{\mu, \sigma} \mathbb{E}_{a \sim \pi} [Q(s,a)]$
> > 2. Gradient from entropy term: $\nabla_{\mu, \sigma} \mathbb{E}_{a \sim \pi} [\mathcal{H}(\pi(\cdot|s))]$
> >
> > These two components can potentially conflict with each other during optimization.
> > We conducted a simple demonstrative experiment in DMC humanoid-run, where we detected the cosine similarity between these two gradient components during training. positive cosine similarity indicates alignment (angle < 90 degrees), while negative cosine similarity indicates conflict (angle > 90 degrees). Our results, which can be found in **Appendix C.1.11**, show that in most of the training process, all 5 seeds tested exhibit negative cosine similarity, indicating a conflict between the two objectives. This phenomenon is observed on both log standard deviation and mean parameters of the policy, meaning that both parameters receive conflicting gradient signals from the two objectives.
> >
> > However, in ERA, **there is no such conflict, as policy parameters in ERA only receive gradient flow from the cumulative reward objective, while the entropy constraint gradient is separated from the mean and log std, and only affects a single $\hat{\delta}$ parameter.** Therefore, there is no conflict between the two objectives in ERA.
> >
> > SAC's weight adjustment only directly affects the loss function, and **always needs gradient flow on distribution parameters to optimize the policy entropy**, while ERA's truncation compensation adjustment directly affects the model output entropy. This fundamental difference leads to the conflict in SAC but not in ERA.
> >
> > ---
> >
> > **Q5**: "How do you explain this difference in entropy oscillations between SAC and ERA since both are tracking entropy constraint by online adjustment of a parameter?"
> >
> > **A**: Thank you for your insightful question. We must clarify that: **while both SAC and ERA adjust a parameter online to track the entropy constraint, the nature of the parameters being adjusted and the underlying mechanisms for entropy regulation are fundamentally different.**
> >
> > In SAC, the adjusted parameter is a weight that influences the trade-off between entropy and Q-function optimization in the loss function. The model needs to optimize the policy entropy through weight adjustment.
> >
> > The feedback loop for SAC is: Adjust weight -> Influence loss function -> Influence model parameter updates -> Influence policy output -> Influence policy entropy.
> >
> > However, in ERA, the adjusted parameter is a truncation compensation value that directly affects the model's output entropy.
> >
> > The feedback loop for ERA is: Adjust truncation compensation -> Influence model output -> Influence policy entropy.
> >
> > It is evident that ERA's feedback loop is shorter and more direct, leading to more timely adjustments and greater stability with less oscillation. In contrast, SAC's longer and more indirect feedback loop is more susceptible to noise, resulting in greater oscillations.  We believe this mechanistic difference fundamentally explains the observed stability gap. While improving SAC's entropy regulation is an interesting research direction for future work, our comparison reflects the behavior of standard, widely-adopted implementations (e.g., SB3, JaxRL), where ERA consistently demonstrates superior stability.

---

> > ### Author Response · Authors · 2025-11-27
> >
> > **Q6**: "Please provide a complete proof for the positivity of $\delta_{\text{TN}}$, or provide references for the claims."
> >
> > **A**: Thank you for your suggestion. We have now provided a more detailed proof of the positivity of $\delta_{\text{TN}}$ in **Appendix B.1.4**. The new proof includes a more comprehensive mathematical derivation process. The proof still relies on some fundamental mathematical results (e.g., Gaussian distribution has the maximum entropy for a fixed variance, and some well known properties of truncated Gaussians). These results are well-established in information theory and probability theory. If you are unfamiliar with these concepts, we recommend consulting relevant classic texts, such as *Continuous Univariate Distributions* by N.L. Johnson, S. Kotz, and N. Balakrishnan, Vol. 1, 2nd Edition.
> >
> > We would like to emphasize that further discussion of these concepts goes beyond the main scope of our paper, as these results are well-established in mathematics and is not a contribution of our work. In fact, the non-negativity of $\delta_{\text{TN}}$ does not affect our theoretical and practical results. **Even if $\delta_{\text{TN}}$ is not non-negative, the entropy bound of ERA still holds.** We can even dispense with the dynamically adjusted parameter $\hat{\delta}_{\text{TN}}$ altogether.
> >
> > Still, we appreciate your interest in the mathematical rigor of our work and do thank you for improving rigorousness of our presentation.
> >
> > ---
> >
> > **Q7**: "Why is performance not changing in Fig. 27 for ERA when the entropy target changes? Could you provide entropy plots for both methods?"
> >
> > **A**: We use a fixed entropy target of 0.6 for ERA in ablation studies in appendix, the "entropy targets (1.2, 1.25, 1.5)" mentioned refers to the entropy targets used for the dynamically adjusted entropy coefficient method. We apologize for this possible confusion, and have now stated this more clearly in the revised manuscript.
> >
> > It's very important to clarify the difference of the meaning of "entropy target" between ERA and SAC methods in discrete spaces.
> > * In ERA, the entropy target refers to a lower bound on the entropy of the output distribution. The model learns to ensure that the entropy of its output distribution is at least this lower bound throughout training. During training, the model's average output entropy is actually much greater than this lower bound(for instance, we get an average output entropy of over 1.2 when using an entropy lower bound of 0.6 for ERA).
> > * In SAC, the entropy target refers to the expected entropy of the output distribution. Therefore, if we set the same entropy target of 0.6 for SAC, it would have no effect as the model cannot lower its entropy to this level, resulting in performance identical to not using entropy regularization at all.
> >
> > We have now supplemented entropy curves for both ERA, the SAC method and the linear interpolation method. These results can be found in **Appendix C.2.1**.
> > In image classification, using SAC-style entropy targets cannot constrain the model's entropy effectively, as the number of update steps is much smaller than that in RL settings, and the model does not have enough time to adjust its entropy through loss optimization, yet still higher entropy targets correspond to slightly higher output entropy. We observe that the average output entropy of ERA (with a lower bound of 0.6) is close to that of the SAC method with an entropy target of 1.25. While ERA achieves better Top-1 and Top-5 accuracy in this setting. We believe this result is enough to demonstrate the advantage of ERA compared to the SAC-style entropy target method in image classification.
> >
> > The Linear Interpolation method's entropy curve is similar to that of ERA, as they both regulate the lower bound of the output entropy. However, ERA still outperforms the Linear Interpolation method in terms of classification accuracy, which we attribute to the more informative gradient flow provided by ERA, as discussed in our response to Q1.
> >
> > **We would like to clarify that ERA in discrete spaces does not rely on two-time scale optimization**, as ERA directly adjusts the model's output entropy **without modifying any params online in discrete spaces.** Please refer to **Section 4.2** for details on ERA in image classification. If the reviewer has any further questions regarding this, we would be happy to discuss them.

---

> > ### Author Response · Authors · 2025-11-27
> >
> > **Clarification on "the contribution for continuous actions is that you have extended previous work to Squashed/Truncated Gaussians."**
> >
> > We respectfully disagree with the characterization that our contribution is merely an extension of previous work to Squashed/Truncated Gaussians. While we acknowledge that previous works [1, 2] were indeed pioneers in exploring entropy constraints without explicit entropy terms, we must point out that our work in continuous spaces is **not a simple extension to new distributions**, but rather a substantial innovation in both mathematical methodology and practical utility.
> >
> > **1. Substantial Mathematical Innovation**
> > First, we emphasize that the core transformation of ERA is derived based on the Gaussian distribution and is fully applicable to Gaussian and any Gaussian-based transformed distributions (e.g., Squashed and Truncated Gaussians), rather than being solely dependent on them.
> >
> > In the simplest case of a Gaussian distribution, the mathematical form of our method is:
> >
> > $\mu' = \mu$,  $\sigma'=\exp{\left[\max\left(\log \sigma_{\max}$ + $(\mathcal{H}_0 -D\log \sqrt{2\pi e}-D \log \sigma_{\max})\frac{e^{\hat{\sigma}_i}}{\sum_{j=1}^{D}e^{\hat{\sigma}_j}} , \log\sigma_{\min}\right)\right]}$
> >
> > In contrast, the method used in [1, 2] takes the following form:
> >
> > $\mu' = \mu$, $\sigma' = \sigma \cdot \exp{(\mathcal{H}_{0} - \mathcal{H}(\sigma))/D}$
> >
> > These forms are mathematically distinct, and there is no direct "extension" relationship between them. Moreover, in ERA the variables $\hat{\sigma}_i$ no longer represent standard deviations. Instead, they act as entropy allocators, learning how to distribute entropy across dimensions. Under this interpretation, the resulting transformation cannot be seen as a simple projection—removing the learned allocator would break the mechanism entirely. **Thus, ERA introduces a genuinely new mathematical transformation rather than a projection-based variant of prior methods.**
> >
> > We have provided a detailed analysis of the mathematical differences between these two methods in continuous space in **Appendix C.1.5**. Briefly, the method in [2] uniformly multiplies a coefficient across all dimensions, delivering no information regarding the allocation of entropy in backpropagation, whereas our method helps the policy learn to allocate entropy *across* dimensions. Our experimental results further confirm this, showing that the distribution of ERA's log-standard deviation differs significantly from that of [2]. These factors fully demonstrate that ERA is a substantive mathematical innovation, not a trivial extension.
> >
> > **2. Significant Difference in Practical Utility**
> > Secondly, there is a critical distinction in terms of utility. While we recognize the pioneering nature of [1, 2], their performance in modern, complex environments (e.g., DMC, HumanoidBench) is suboptimal. We applied the same expansion methodology used in ERA to the schemes proposed in [1, 2]. **If the "extension to new distributions", instead of our novel mathematical transformation, were truly our core contribution, then the extended schemes of [1, 2] should yield performance results comparable to ERA.**
> >
> > However, the experimental results show that the extended versions of [1, 2] perform poorly, **failing to surpass even the SAC baseline in overall performance**. By comparison, ERA achieves performance more than double that of the schemes proposed in [1, 2]. This is a statistically significant difference in usability, **fully illustrating that ERA's mathematical transformation is a substantial innovation in practicality, rather than a simple extension.**
> >
> > **Conclusion**
> > In summary, we believe there is a misunderstanding regarding the nature of our work. To categorize it as a "simple extension to new distributions" is, in our view, inaccurate. We hope this response clarifies our contribution for Reviewer x7W9. We welcome further discussion should any doubts remain.

---

### Official Review · Reviewer_m4ks · 2025-10-31

**Soundness:** 3
**Presentation:** 3
**Contribution:** 3
**Rating:** 8
**Confidence:** 4

**Summary:**

The paper introduces Entropy Regularizing Activation (ERA), a framework that enforces entropy constraints via custom output activations rather than through explicit loss terms. This allows entropy control without disturbing the main optimization objective. The authors demonstrate ERA in three settings, continuous control, image classification, and reinforcement learning for large language models, showing consistent performance improvements and theoretical guarantees for minimum entropy. The method is lightweight, general, and easy to integrate.

**Strengths:**

**1. Simple yet general idea:** Treating entropy regularization as an architectural operation rather than a loss modification is elegant and broadly applicable. It bridges reinforcement learning, LLM training, and supervised classification under one principle.

**2. Strong empirical validation:** The experiments are extensive and diverse, showing consistent improvements with minimal tuning or overhead (<7%). This cross-domain effectiveness strengthens the claim of universality.

**Weaknesses:**

**1. Naming / framing:** Calling it “entropy as an activation” feels misleading; the object under control is the policy entropy. “Entropy-constrained policy" or "Entropy-constrained policy via output activation” would better reflect what’s actually enforced (and avoid implying entropy is a property of the activation itself).

**2.Control-side doubts:** SAC’s state-conditional entropy is a feature, not a bug: high entropy in uncritical states, low in critical ones. A fixed or globally enforced entropy can bias learning toward suboptimal policies and blunt per-state adaptivity (e.g., see recent discussions on fixed-entropy pitfalls) [1]. This deserves explicit treatment in limitations and experiments probing failure modes when the target is mis-set.

**3. Baseline realism in SAC:** Recent work/practice often uses very small or carefully scheduled entropy coefficients in SAC to avoid instability. If ERA is compared to SAC with suboptimal temperature tuning, improvements may be overstated [2],[3]. The paper should verify that gains hold under strong, modern SAC setups (small-α inits, robust target-entropy tuning/schedules).

[1] When Maximum Entropy Misleads Policy Optimization, ICML'25.

[2] https://araffin.github.io/post/sac-massive-sim/

[3] Hyperspherical Normalization for Scalable Deep Reinforcement Learning, ICML'25.

**Questions:**

Some suggestions following from Weakness

**1. Clarify the naming and framing**: Consider revising the term “entropy as an activation” to something like “entropy-constrained policy via activation” to better capture the actual mechanism. The current phrasing may be conceptually misleading.

**2. Acknowledge limitations of fixed entropy:** In the limitations section, explicitly discuss when ERA might underperform compared to standard SAC or entropy-maximized RL, especially in cases where adaptive, state-dependent entropy is important.

**3. Evaluate under stronger SAC setups:** Re-run or compare against more standardized, high-performing SAC configurations (e.g., small initial entropy coefficients or temperature scheduling) to demonstrate that ERA’s improvements persist under robust, modern baselines.

---

> ### Author Response · Authors · 2025-11-20
>
> We thank the reviewer for the time and effort dedicated to reviewing our work. We deeply appreciate your careful and thorough feedback. In the following, we address each of your concerns.
>
> **Note: Modifications made to the paper in response to Reviewer m4ks are highlighted in Soft Purple (147, 112, 219).**
>
> ---
>
> **Q1:** "Calling it 'entropy as an activation' feels misleading, should use 'Entropy-constrained policy via output activation'."
>
> **A:**
> We have carefully reviewed the manuscript and would like to clarify that the specific phrasing "entropy as an activation" does not appear in the original text. Nevertheless, to ensure absolute clarity and prevent any potential ambiguity regarding the mechanism, we have refined the descriptions in the **Abstract**, **Introduction**, and **Method (Section 4.1)**. These revisions now explicitly emphasize the nature of ERA as an "Entropy-constrained policy via output activation," which we believe effectively addresses your concern and improves the precision of our terminology.
>
> ---
>
> **Q2:** "Control-side doubts: A fixed or globally enforced entropy can bias learning toward suboptimal policies and blunt per-state adaptivity. Please explicitly treat this in limitations and probe failure modes."
>
> **A:**
> We appreciate this insightful comment regarding entropy adaptivity. In **Appendix C.1.2**, we have provided an alternative implementation of ERA, termed **Batch-Level ERA**, which explicitly allows for state-conditional entropy levels (as opposed to the global constraint in State-Level ERA). However, in the complex locomotion tasks we evaluated (including challenging environments in DMC and HumanoidBench—domains that are also the focus of many mainstream Continuous Control algorithms), we observed no substantial performance difference between the state-conditional Batch-Level ERA and the State-Level ERA presented in the main text. This suggests that for the extensive suite of tasks tested, the distinction between state-conditional and global entropy constraints is not the primary factor influencing performance.
>
> Regarding Reference [1], we note that this work primarily identifies scenarios where the **Maximum Entropy RL** objective itself may mislead policy optimization, rather than focusing on the distinction between state-conditional versus global entropy constraints. In fact, standard SAC also underperforms in the counter-examples provided in that reference.
>
> To further clarify the scope of our contribution, we have added a **"Limitations and Future Work"** section to the main text. There, we explicitly state that our work is premised on the Maximum Entropy RL framework. We acknowledge that in specific scenarios where Maximum Entropy objectives do not lead to optimal policies (as discussed in Reference [1]), further investigation into the applicability of ERA would be valuable.
>
> ---
>
> **Q3:** "Compare ERA against more standardized, high-performing SAC configurations (e.g., small initial entropy coefficients) to demonstrate that ERA’s improvements persist under robust, modern baselines."
>
> **A:**
> Following your suggestion, we have conducted additional experiments comparing ERA against SAC configurations initialized with small temperature coefficients. These results are now detailed in **Appendix C.1.8: Comparison With Small Initial Temperature SAC**.
>
> Our comparative experiments in 4 tasks (`dog-run`,`humanoid-walk`,`h1-walk` and `h1-run`) demonstrate that while initializing SAC with a smaller temperature (0.006) does improve performance on 2 of these tasks (dog-run, humanoid-walk) compared to the default setting (1.0), the overall performance gains remain limited. In h1-walk and h1-run, the small-initial-temperature SAC shows similar or slightly worse performance compared to the default SAC, indicating that tuning the initial temperature does not consistently yield improvements across all tasks.
>
> This result highlights a critical insight: **while reducing the initial temperature can partially mitigate the interference of the entropy term on the reward optimization, it does not fundamentally resolve the objective conflict.** In contrast, ERA addresses this issue structurally, leading to superior performance in complex control tasks.
>
> Furthermore, we maintain that using an initial temperature of 1.0 serves as a justified and realistic baseline. This value remains the default configuration in widely adopted RL libraries such as Stable Baselines3 and JaxRL, reflecting common usage in the community. Moreover, the initial temperature is a hyperparameter that possibly requires task-specific tuning in standard SAC; a key advantage of ERA is that it eliminates the need for this per-environment tuning, offering a more robust and generalizable solution.

---

> ### Author Response · Authors · 2025-11-26
>
> Dear Reviewer m4ks,
>
> We sincerely appreciate your time and effort in reviewing our manuscript and providing valuable feedback. **Thank you very much for acknowledging that our idea of treating entropy regularization as an architectural operation is elegant and broadly applicable, as well as for recognizing the strength of our empirical validation across diverse domains.**
>
> As the rebuttal period is nearing its end, we would like to kindly inquire whether our responses have adequately addressed your concerns. We have provided detailed clarifications in our reply, and we would be more than happy to discuss any remaining questions you might have. We would be deeply grateful if you could kindly reconsider raising the score in light of our efforts and clarifications.
>
> Thank you again for your consideration.
>
> Best regards, The Authors

---

### Author Response · Authors · 2025-11-25

We thank the reviewers for their time, detailed reviews, and constructive feedback. We have carefully considered all comments, conducted additional experiments and analyses, and revised the manuscript accordingly.

**We have uploaded the revised draft and provided detailed point-by-point responses to each reviewer.** Some of the key modifications include:

### 1. Comparison with A Prior Method (Reviewer x7W9)
We compared ERA with a prior method from **Akrour et al. (2019)** and **Otto et al. (2021)**. We theoretically demonstrate the limitations of their projection method, and empirically show ERA's superior performance.

### 2. Extended Baseline Comparisons (Reviewers m4ks, x7W9)
To demonstrate robustness, we added experiments comparing ERA against **Small Initial Temperature SAC** (Reviewer `m4ks`) and **Stable Baselines3 implementations** (Reviewer `x7W9`). We also provided evidence that standard entropy regularization often degrades performance in Vision and LLM domains, whereas ERA remains robust.

### 3. Comprehensive Ablation & Sensitivity Studies (Reviewers 8uvP, 1XEa)
We verified the contribution and sensitivity of key components, including:
* **LLM-RL:** Ablations on advantage modification and token selection ratios (Reviewer `8uvP`).
* **Hyperparameters:** Sensitivity analyses for the standard deviation interval $[\sigma_{\min}, \sigma_{\max}]$ and target entropy analyses for PPO (Reviewer `1XEa`).

### 4. Mechanism & Efficiency Analysis (Reviewers x7W9, 1XEa)
To clarify ERA's working mechanism, we analyzed **entropy allocation dynamics** (Reviewer `x7W9`) and **runtime overhead** (Reviewer `1XEa`). Results confirm that ERA incurs negligible cost (approx. 3%) in SAC training while effectively controlling entropy.

### 5. Other Equally Important Modifications (All Reviewers)
We addressed all remaining concerns, including refining terminology to "Entropy-constrained policy via output activation" (Reviewer `m4ks`) and clarifying definitions and mathematical notations (Reviewer `1XEa`). For details, please refer to our point-by-point responses.

---

We believe these improvements have significantly elevated the quality and reliability of our paper. **If there are any further questions or if additional clarification is needed, please do not hesitate to let us know. As the rebuttal deadline approaches, we are eager to engage in further discussion and are fully prepared to make any additional modifications or conduct further experiments if the reviewers deem them necessary.**

---

### Author Response · Authors · 2025-12-04

First of all, we sincerely thank all reviewers for their diligent reviews and constructive feedback. We are thrilled that reviewers have acknowledged the groundbreaking novelty of our approach. By directly manipulating model outputs to constrain entropy, we have unlocked a powerful method that demonstrates exceptional effectiveness across diverse domains. Our experiments showcased its remarkable potential, opening a new direction in the field.

Following the reviewers' suggestions, we have primarily conducted the following additional experiments:
* **As suggested by Reviewer m4ks:**
    * We compared ERA in continuous control tasks against SAC configurations initialized with small temperature coefficients.
* **As suggested by Reviewer x7W9:**
    * We compared ERA with a previous projection-based method on continuous control and image classification tasks (we also attempted this on LLMs, but the projection-based method proved difficult to train successfully).
    * We compared our baseline implementation in continuous control tasks with the Stable Baselines3 library to ensure the validity of our results.
    * We compared ERA and SAC on continuous control tasks in terms of entropy dynamics.
    * We compared ERA with entropy term regularization on image classification and LLM RL tasks.
    * We conducted an experiment to demonstrate the gradient conflict between the entropy term and the task reward on SAC in continuous control tasks.
* **As suggested by Reviewer 8uvP:**
    * We conducted an ablation study on the ratio of tokens used to calculate $H_{\text{resp}}$ in LLM RL tasks.
    * We conducted an ablation study to test ERA in LLM RL without the advantage modification.
* **As suggested by Reviewer 1XEa:**
    * We conducted an ablation study on the choice of entropy targets for ERA in continuous control tasks.
    * We conducted an ablation study on the choice of $[\sigma_{\min}, \sigma_{\max}]$ in ERA for continuous control tasks.

In addition to these supplementary experiments, we have also provided detailed explanations to address the reviewers' concerns:

* **As suggested by Reviewer m4ks:**
    * We clarified the nature of ERA as "Entropy-constrained policy via output activation."
    * We added a discussion on the limitations of our work (specifically, that ERA is fundamentally grounded within the maximum entropy framework).
* **As suggested by Reviewer x7W9:**
    * We included key related works and clarified the novelty and effectiveness of our method compared to previous projection-based methods, both theoretically and experimentally.
    * We further clarified the fundamental distinction between using ERA and SAC-style entropy regularization in continuous control tasks.
    * We improved the proof and analysis of the entropy bound in continuous control tasks.
    * We made clarifications regarding the Q-function used in SAC-ERA.
    * We clarified ERA's effectiveness on image classification and LLM RL tasks compared to entropy term regularization.
* **As suggested by Reviewer 8uvP:**
    * We clarified ERA's insensitivity to the choice of hyperparameters, such as the ratio of tokens used to calculate $H_{\text{resp}}$ and the advantage modification in LLM RL tasks.
* **As suggested by Reviewer 1XEa:**
    * We added definitions and introductions for symbols such as $a$ and $y^{1:K}$ in the main text.
    * We made key modifications to correct typos regarding ERA in image classification tasks and completely corrected the proof.
    * We clarified the definition of the inverse function $h^{-1}$ and the selection of the approximation $\hat{h}^{-1}$ in image classification tasks.
    * We improved the writing and clarity of the proof regarding ERA in LLM RL tasks.
    * We made critical revisions to specific phrasings in the paper; for instance, clarifying that the $\tau$-divided distribution is not a strictly valid probability distribution.
    * We clarified implementation details, such as the choice of entropy targets and $[\sigma_{\min}, \sigma_{\max}]$ in ERA for continuous control tasks.
    * We clarified the normalized score calculation, the meaning of shaded areas in the plots, and the environment names used in the runtime comparison tests.

We hope these additional experiments and clarifications will further strengthen our paper and satisfactorily address all raised concerns. All corresponding updates have been incorporated into the revised submission. We thank the reviewers again for their valuable feedback and consideration of our work.

Lastly, we would also like to thank all ACs and PCs for their significant efforts and contributions.

Best Regards,

Authors

---

### Meta-Review · Program_Chairs · 2026-01-09

**Summary:**

The authors present a simple heuristic they claim improves the performance of many RL algorithms. They modify the output of a neural network so that the distribution over actions (for RL) or predictions (for classification) has an entropy that is always bounded below by a fixed value. There are several different versions of this heuristic presented - one for RL for reasoning in LLMs, one for RL for control in continuous action spaces, and one for CIFAR-10 and ImageNet classification. The paper presents this as a unified framework, even though each particular instantiation seems particular to the application considered. The presentation is somewhat mixed - the paper motivates the approach entirely through contrast with entropy bonuses in entropy-regularized RL, but one of the three major experimental results is not in an RL setting. The terminology refers to the output layer of the network as an "activation", but "activation" usually refer to intermediate layers of a neural network (e.g. ReLU, tanh, etc are activation functions). Reviews were mixed, with one solid accept, two weak accepts and one weak reject. The paper is quite thorough, with a length appendix with both theoretical and further experimental results. Demonstrating a robust advantage in RL problems is extremely difficult, because training can be so sensitive to minor details. I also struggle to see why the same entropy-bounding effect couldn't be achieved by making the alpha term in SAC adaptive, making the bonus larger if the entropy drops below a critical value. This should be addressed in the camera-ready version. Overall, the paper puts an impressive amount of effort into proving out a relatively simple idea.

**Reviewer Concerns:**

The reviewers shared many of my concerns (even the highest score review had concerns about proper comparison against SAC). The authors did a significant amount of work to address reviewer concerns about SAC temperature scheduling and the like.

**Reviewer Scores:**

It's difficult to say. The concerns in the highest and lowest scoring reviews were similar, it's hard to say exactly why they gave the score they did.

---

### Decision · Program_Chairs · 2026-01-26

Accept (Poster)